# C-reactive protein is a broad-spectrum capsule-binding receptor for hepatic capture of blood-borne bacteria

Danyu Chen[1,7], Jiao Hu[1,7], Mengran Zhu[1,7], Yufeng Xie [2,7], Hantian Yao[1,7], Haoran An [3], Yumin Meng[2,4], Juanjuan Wang [1], Xueting Huang[1], Yanni Liu[1,5], Zhujun Shao[6], Ye Xiang [1,5], Jianxun Qi [2,4✉], George Fu Gao [2,4✉] & Jing-Ren Zhang [1✉]

## Abstract

**Plasma C-reactive protein (CRP) is widely used as a biomarker for bacterial infections due to its massive induction during infections. However, the biological function of CRP remains largely undefined. Here we show that CRP enables liver resident macrophages (Kupffer cells) to capture and eliminate a wide range of invasive bacteria from the bloodstream of mice, and thereby provides rapid and sterilizing immunity. Mechanistically, CRP binds to at least 20 capsule types of Gram-positive and -negative pathogens, and shuffles the encapsulated bacteria to Kupffer cells embedded in the lining of the liver sinusoidal vasculatures by the complement-dependent and -independent pathways. The complement-dependent mode involves the activation of complement C3 at the bacterial surface, and the capture of the C3-opsonized bacteria by the CRIg and CR3 complement receptors on Kupffer cells. Cryo-electron microscopy analysis revealed a flexible structural framework for CRP's recognition of structurally diverse capsular polysaccharides. Because human CRP also possesses the broad capsule-binding activities, our findings provide a biological reason for the massive rise of plasma CRP during bacterial infections.**

**Keywords** C-reactive Protein; Encapsulated Bacteria; Capsule; Kupffer Cell and Complement System
**Subject Categories** Immunology; Microbiology, Virology & Host Pathogen Interaction

## Introduction

C-reactive protein (CRP) is an acute phase plasma protein named after its binding to the cell wall C-polysaccharide or phosphocholine (PC) of *Streptococcus pneumoniae* (pneumococcus) (Tillett and Francis, 1930; Volanakis and Kaplan, 1971). PC was later found as a constituent of *Haemophilus influenzae* lipopolysaccharide (LPS) and other bacterial polysaccharides (Kolberg et al, 1997; Mold et al, 1982; Volanakis and Kaplan, 1971; Weiser et al, 1998). Human CRP is present at a trace level in healthy individuals, but rapidly rises by as much as 1000 folds during severe bacterial infections and other inflammatory conditions, making it a common biomarker for clinical diagnosis of systemic infections and inflammation (Plebani, 2023). Surprisingly, the biological function of CRP has remained largely speculative for nearly a century since its discovery (Ji et al, 2023). Many single nucleotide polymorphisms (SNPs) have been identified in the human CRP gene locus, but none of these change the amino acid sequence of CRP (Carlson et al, 2005). A polymorphisms in the CRP promoter region is associated with the increased mortality in pneumococcal bacteremia patients (Eklund et al, 2006).

CRP is evolutionarily conserved from anthropoids to humans (Pathak and Agrawal, 2019). Mouse CRP shares 71% amino acid homology with the human ortholog, but is stably expressed at low levels even under infection conditions (Sproston and Ashworth, 2018). Passive administration or transgenic expression of human CRP in mice has been reported to protect mice from septic infection of *S. pneumoniae* (Mold et al, 1981; Ngwa et al, 2020a; Ngwa et al, 2020b; Szalai et al, 1995; Yother et al, 1982), but the combined data demonstrate a modest immunity. Consistently, CRP binding to the pneumococcal cell wall does not stimulate effective phagocytic killing in vitro (Holzer et al, 1984).

As a member of the C-type lectin protein family, CRP is a disc-shaped homo pentameric protein, in which each protomer contains two calcium ions (Shrive et al, 1996). PC is the best-defined

[1]Center for Infection Biology, School of Basic Medical Sciences, Tsinghua University, Beijing 100084, China. [2]CAS Key Laboratory of Pathogen Microbiology and Immunology, Institute of Microbiology, Chinese Academy of Sciences, Beijing 100101, China. [3]Institute of Medical Technology, Peking University Health Science Center, Beijing 100191, China. [4]University of Chinese Academy of Sciences, Beijing 100049, China. [5]Tsinghua-Peking Joint Center for Life Sciences, Tsinghua University, Beijing 100084, China. [6]National Key Laboratory of Intelligent Tracking and Forecasting for Infectious Diseases, National Institute for Communicable Disease Control and Prevention, Chinese Center for Disease Control and Prevention, Beijing 102206, China. [7]These authors contributed equally: Danyu Chen, Jiao Hu, Mengran Zhu, Yufeng Xie, Hantian Yao. ✉E-mail: jxqi@im.ac.cn; gaof@im.ac.cn; zhanglab@tsinghua.edu.cn

microbial ligand of CRP, but many autologous PC-free targets of CRP have been described, such as complement component 1q (C1q) and Fcγ receptors (Bharadwaj et al, 1999; Crowell et al, 1991; Kaplan and Volanakis, 1974; Marnell et al, 1995). CRP forms two functional faces: the activating face (A face) and the ligand binding face (B face) (Guillon et al, 2014; Noone et al, 2021). Under the in vitro conditions, CRP interaction with PC at the B face leads to the subsequent C1q binding at the A face and activation of the complement system through the classical pathway (Kaplan and Volanakis, 1974; Mortensen et al, 1976). Human CRP has been shown to activate the mouse complement system, which is attributed to human CRP-mediated protection in mouse models of pneumococcal bacteremia (Mold et al, 2002; Szalai et al, 1996).

Capsule-coated or encapsulated bacteria cause many invasive diseases in humans and domestic animals, such as pneumonia, sepsis and meningitis (An et al, 2024; Naghavi and Collaborators, 2022). Capsules are known for their special physical and antiphagocytic properties (Brown and Gresham, 2012; Nahm and Katz, 2012; Taylor and Roberts, 2002). Nearly all the capsules consist of capsular polysaccharides (CPSs) (An et al, 2024; Whitfield et al, 2020). Many pathogens produce large numbers of capsule types that differ in structure and antigenicity (An et al, 2024). More than 100 capsule serotypes have been documented for the human pathogens *S. pneumoniae* and *Klebsiella pneumoniae* (Manna et al, 2024; Wyres et al, 2016). Capsule types are clinically associated with the disease potentials of encapsulated bacteria. The low-numbered pneumococcal serotypes/serogroups were dominantly identified in fatal pneumonia cases in the pre-antibiotic era (White, 1938); *H. influenzae* type b is responsible for the vast majority of the childhood infections among the six serotypes (Retchless et al, 2024).

Kupffer cells (KCs), the liver resident macrophages embedded in the liver sinusoids, represent approximately 90% of the total tissue macrophages in the body (Bilzer et al, 2006). KCs are highly capable of clearing blood-borne bacteria (Broadley et al, 2016; Gola et al, 2021; Helmy et al, 2006; Kolaczkowska et al, 2015; Lee et al, 2010; Surewaard et al, 2016; Wong et al, 2013; Zeng et al, 2016). Our recent studies have uncovered that capsules shield encapsulated bacteria from KC capture in the liver sinusoids (An et al, 2022; Huang et al, 2022). Consistent with the clinical association between capsule type and disease potential, certain capsule types are more capable of circumventing KC capture than the others in a mouse sepsis model (An et al, 2022; Huang et al, 2022). The high-virulence (HV) types escape KC capture and thereby replicate in the blood, resulting in severe bacteremia and septic death. To a less extent, the low-virulence (LV) types partially escape the hepatic interception and only cause fatal infection when KC-mediated antibacterial "firewall" is compromised or overwhelmed (An et al, 2022; Huang et al, 2022). The capture of the LV capsule types by KCs indicates the existence of host receptors for these capsules. In sharp contrast to the massive numbers of capsule variants, the asialoglycoprotein receptor (ASGR) is the only known capsule receptor on KCs (recognizing serotype-7F and -14 capsules of *S. pneumoniae*) (An et al, 2022). Our recent study has identified plasma natural antibodies as the binding receptor for serotypes-10A and -39 capsules of *S. pneumoniae* and serotype-K50 capsule of *K. pneumoniae* (Tian et al, 2025). However, it remains completely unclear how KCs recognize many other capsules.

This study sought to identify new capsule-binding receptors that enable KCs to capture the LV bacteria in the liver. CRP was found to broadly recognize a wide range of capsular serotypes of Gram-positive and -negative bacteria. The receptor-ligand interactions enable liver KCs to capture and kill blood-borne bacteria, providing a potent immunity against otherwise fatal infections.

## Results

### CRP binds to capsular polysaccharide of serotype-23F *S. pneumoniae*

To understand how the liver executes the clearance of blood-borne bacteria, we sought to identify the receptor(s) recognizing the capsule of serotype-23F *S. pneumoniae* (*Sp*23F), one of the most predominant serotypes causing the childhood invasive infection (Silva-Costa et al, 2024). Our previous study has shown that *Sp*23F is rapidly captured by KCs into the liver of mice (An et al, 2022). Our initial experiment showed that the hepatic immunity is blocked by free capsular polysaccharide of *Sp*23F (CPS23F) (Fig. 1A). Consistently, the freshly isolated KCs from mouse liver showed significant adherence to LV *Sp*23F with normal mouse serum, but were poorly adherent to HV *Sp*8 (An et al, 2022). We enriched CPS23F-binding protein(s) by affinity pulldown after incubation of CPS23F-coated beads with the membrane protein-enriched fraction of mouse liver nonparenchymal cells (NPCs) in the presence of 10% mouse serum (Fig. 1B). CPS of the HV serotype-8 *S. pneumoniae* (CPS8) was used as a negative control due to its poor binding to KCs (An et al, 2022). Serum was included since circulating bacteria are soaked in the blood before being captured by KCs. Linear regression comparison of CPS23F- and CPS8-enriched proteins showed a low correlation of $R^2 = 0.1094$ between the two sets of proteins, and identified a total of 255 significantly enriched proteins by CPS23F. C-reactive protein (CRP) was revealed as the top hit followed by Mettl3, Slc14a2, C1sb, C1ra, Lyz1, Rps26, Tpm4, Ep4112 and Oas1g (Fig. 1C; Appendix Tables S1 and S2).

We verified the interaction between CPS23F and CRP using the recombinant mouse CRP (r-mCRP) (Fig. EV1A). r-mCRP showed a dose-dependent binding to immobilized CPS23F, but not CPS8 (Fig. 1D), in which a saturating concentration of CPS8 and CP23F was used to coat 96-well plates (10 μg/ml) on the basis of the preliminary trial (Fig. EV1B). Since C-polysaccharide is a common contaminant of pneumococcal capsular polysaccharide preparations (Paliwal et al, 2024), we tested its impact on CRP-capsule interaction using anti-PC monoclonal antibodies. The antibody detected a substantial level of PC in CPS23F, which was significantly removed by the combination of anti-PC antibodies and protein-G beads (Fig. EV1C). However, the same PC depletion did not obviously impact the level of CRP-capsule binding (Fig. 1E). This finding showed that the *Sp*23F capsule is a specific ligand for mouse CRP.

We further characterized specific binding of CRP to the capsule of the intact bacteria by flow cytometry. The result revealed a negligible r-mCRP binding to serotype-8 bacteria (1.5%), but the protein was found to bind to virtually all serotype-23F pneumococci (98.5%) (Fig. 1F). In contrast, both the serotypes showed a

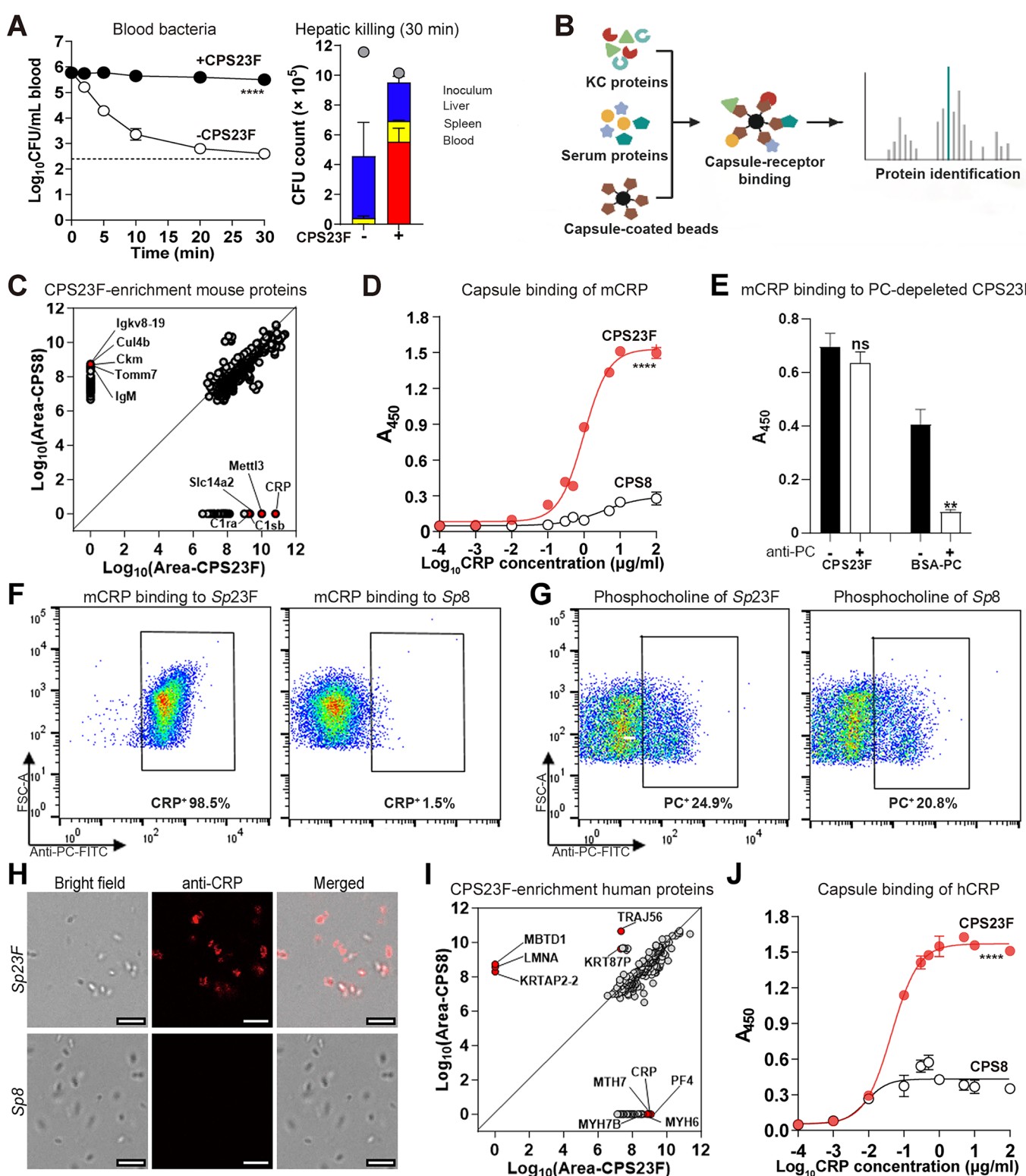

comparable level of binding to anti-PC antibodies (Fig. 1G; Appendix Fig. S1). Likewise, fluorescence microscopy showed that r-mCRP bound to the live Sp23F, but not Sp8 (Fig. 1H). In agreement with the surface localization and overwhelming abundance of the capsular polysaccharide as compared with the

C-polysaccharide, these results demonstrate that capsule is the dominant target of CRP on the cell envelope of Sp23F.

Since human and mouse CRPs have a 70.9% amino acid sequence identity (Fig. EV2A), we determined if human CRP is an also a dominant capsule-binding protein for serotype-23F

**Figure 1. Identification of CRP as a CPS23F-binding protein.**

(A) Inhibition of free CPS23F against hepatic clearance of serotype-23F *S. pneumoniae* (*Sp*23F). Mice were i.v. inoculated with 400 µg free CPS23F prior to i.v. infection with $10^6$ CFU of *Sp*23F to assess bacterial clearance (left) and hepatic killing (right). Dotted line, detection limit. The inoculum of each group is indicated with a filled circle. The blood and organ CFU in individual mice were presented as mean ± SEM of three biological replicates ($n = 3$). $P$ value was calculated by Two-way ANOVA (****$P < 0.0001$). $P$ value:8.61e−019. (B) Illustration of the affinity pulldown procedure. CPS of *Sp*23F (CPS23F) was coated to latex beads to identify mouse capsule-binding proteins by mass spectrometry. (C) CPS23F-enriched mouse proteins identified by mass spectrometry are plotted against the levels of the same proteins in the CPS8 group; top 5 hits in both the groups specified. (D) Specific binding of recombinant mouse CRP (r-mCRP) to immobilized CPS23F was detected by ELISA using CPS8 as a negative control. $A_{450 nm}$ were presented as mean ± SEM of three biological replicates ($n = 3$). $P$ value was calculated by Two-way ANOVA (****$P < 0.0001$). $P$ value: 1.99e−039. (E) Negligible impact of phosphorylcholine (PC) on the CRP-CPS23F binding. CPS23F and PC-conjugated bovine serum albumin (BSA-PC) were sequentially treated with an anti-PC antibody and protein G beads. The PC-depleted CPS23F and BSA-PC were coated to 96-well plates along with the untreated controls, and reacted with r-mCRP. Bound mCRP were detected using an anti-CRP antibody by ELISA. $A_{450 nm}$ were presented as mean ± SEM of three biological replicates ($n = 3$). $P$ value was calculated by Two-way ANOVA with Sidak's multiple comparisons (**$P < 0.01$; ns $P > 0.05$, no significance). $P$ values: BSA-PC vs BSA-PC + anti-PC, 0.0015; CPS23F vs CPS23F + anti-PC, 0.5836. (F) mCRP deposition to the intact *Sp*23F. Serotype-23F and -8 (negative control) pneumococci were sequentially treated with r-mCRP, anti-CRP antibody and FITC-conjugated secondary antibody before being analyzed by flow cytometry. The percentage of CRP-positive (CRP$^+$) cells is indicated for each serotype. (G) Phosphocholine on the intact pneumococci. *Sp*23F and *Sp*8 were sequentially stained with anti-PC antibody and FITC-conjugated secondary antibody. PC-positive (PC$^+$) cells were identified by flow cytometry and indicated at the bottom of each panel. (H) CRP binding to *Sp*23F was detected by immunofluorescence microscopy with r-mCRP and anti-mCRP antibody; CRP-bound bacteria were visualized with AF647-conjugated secondary antibody (red). Scale bar, 5 µm. (I) CPS23F-enriched human proteins were enriched and identified as in (C). (J) CPS23F binding of human CRP (hCRP) was detected as in (D). $A_{450 nm}$ were presented as mean ± SEM of three biological replicates ($n = 3$). $P$ value was calculated by Two-way ANOVA (****$P < 0.0001$). $P$ value: 1.82e−033. Source data are available online for this figure.

pneumococci by affinity pulldown of human serum proteins with CPS23F-conjugated beads. As compared with CPS8-coated beads, CPS23F-conjugated beads yielded 109 proteins with significant enrichment (Appendix Table S3). The top 10 hits include PF4, CRP, MYH7, MYH6, MYH7B, MYL2, MYL4, MYCL, PLXNA1, TNNT1 (Fig. 1I). Based on the specific binding of mouse CRP to CPS23F, we selectively tested potential binding of human CRP to the capsule. The ELISA result verified specific binding of recombinant hCRP (r-hCRP) to CPS23F (Fig. 1J).

## CRP enables Kupffer cells to capture blood-borne *Sp*23F in the liver

We next determined the functional contribution of CRP against *Sp*23F infection by comparing the early clearance of *Sp*23F between wild-type (WT) and CRP-deficient ($Crp^{-/-}$) mice in the first 30 min post i.v. inoculation of $10^6$ colony forming unit (CFU) bacteria. While WT mice rapidly cleared *Sp*23F with a 50% clearance time ($CT_{50}$) of 1.3 min, $Crp^{-/-}$ mice fully retained the inoculum in the circulation (Fig. 2A). By comparison, $Crp^{-/-}$ and WT mice were equally able to clear serotype-14 pneumococci (*Sp*14), which is known to be recognized by ASGR on KCs (An et al, 2022). Since the liver is the major organ to trap blood-borne *Sp*23F and other LV pneumococci (An et al, 2022), we tested the impact of CRP deficiency on hepatic capture of *Sp*23F. The livers of WT mice contained 72.9% of the inoculum at 5 min post infection and the infected WT mice have only a residual level of bacteremia, whereas only 3.0% of the inoculated bacteria accumulated in the livers of $Crp^{-/-}$ mice, with the vast majority of the bacteria in the blood (Fig. 2B). By comparison, both $Crp^{-/-}$ and WT mice effectively trapped *Sp*14 bacteria in the liver at this time point. Further analysis revealed a dramatic reduction of bacteria in the blood, spleen, and liver of WT mice at 30 min post infection, but $Crp^{-/-}$ mice failed to clear *Sp*23F, resulting in a substantial increase in the total bacteria. This CRP-dependent immunity is serotype specific since $Crp^{-/-}$ mice did not show any impairment in clearing *Sp*14 bacteria, a CRP-insensitive LV serotype. To ascertain the impact of the well-known CRP binding to the C-polysaccharide on the host-pathogen interaction, we tested the clearance of an isogenic

acapsular strain, which maximally expose the C-polysaccharide at the surface. The Δ*cps* mutant was effectively cleared from the circulation of WT and $Crp^{-/-}$ mice in a similar manner (Fig. 2C). This result further verified the capsule as the dominant target of CRP for the hepatic clearance of *Sp*23F.

It has been documented that CRP is upregulated by bacterial infections in human being but not in mouse and many mammals (Pathak and Agrawal, 2019), which may affect the infection outcome. We thus assessed the plasma level of CRP in normal mice at 6, 12, 24, 36, 48, 60, and 72 h (hr) post i.v. infection with $10^6$ CFU of *Sp*23F. The ELISA results revealed that serum CRP remained stable in the course of pneumococcal infection (Fig. EV2B).

Based on the dominant role of KCs in hepatic capture of the LV pneumococci (An et al, 2022), we tested the contribution of CRP to KC capture by using intravital microscopy (IVM) imaging. *Sp*23F pneumococci were rapidly tethered to KCs upon entering the vasculatures of the liver sinusoids of WT mice, but smoothly passed KCs in $Crp^{-/-}$ mice (Fig. 2D and Movie EV1). Consistent with the serotype-specific binding of CRP to pneumococcal capsules, *Sp*14 bacteria were similarly captured by KCs of WT and $Crp^{-/-}$ mice (Fig. 2D and Movie EV2). This KC-enabling immunity of CRP was verified under the in vitro conditions. Primary mouse KCs abundantly bound to *Sp*23F in the presence of serum from WT mice but not that from $Crp^{-/-}$ mice (Fig. 2E). In contrast, the macrophages similarly attracted KCs *Sp*14 regardless of the host source of serum. These lines of evidence demonstrated that CRP enables KCs to capture circulating *Sp*23F in the liver sinusoids.

The significance of CRP in host defense against invasive pneumococcal infection was evaluated by two approaches. We first tested the therapeutic impact of recombinant CRP on the hepatic clearance of blood-borne bacteria. *Sp*23F were pre-incubated with r-mCRP before i.v. inoculation into $Crp^{-/-}$ mice. The pre-opsonization with CRP enabled $Crp^{-/-}$ mice to shuffle *Sp*23F from the circulation to the liver (Fig. 2F). This result demonstrated that CRP is a potent immune molecule that drive hepatic capture of *Sp*23F. The CRP-mediated immunity was further characterized by monitoring the survival of $Crp^{-/-}$ mice after i.v. inoculation with $10^6$ CFU of *Sp*23F. As compared with the full survival of WT mice,

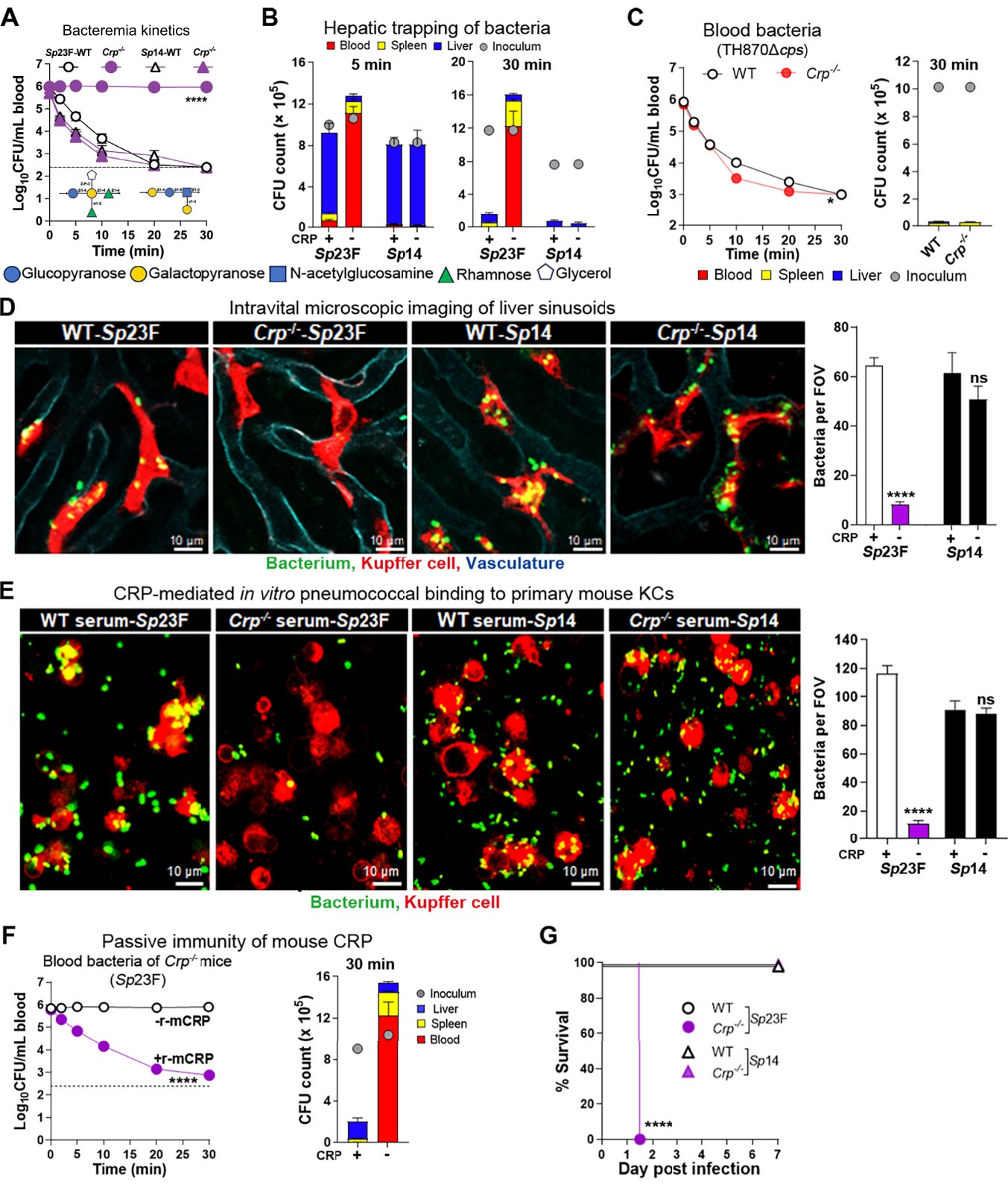

◄ **Figure 2. CRP-activated hepatic capture and killing of serotype-23F *S. pneumoniae*.**

(A) The importance of CRP in clearing *Sp*23F was determined by blood CFU plating of *Crp*$^{-/-}$ and WT mice at various time points post i.v. infection with 10$^6$ CFU of *Sp*23F. The capsule repeat units of serotypes 23F and 14 were presented. Dotted line, detection limit. The blood CFU in individual mice was presented as mean ± SEM of five to six biological replicates (*n* = 5–6). *P* value was calculated by Two-way ANOVA with Tukey's multiple comparisons (23F *Crp*$^{-/-}$ vs 23F WT) (****P* < 0.0001). *P* value: 1.30e −014. (B) The contribution of CRP to hepatic trapping of *Sp*23F was characterized by CFU plating of the blood, liver and spleen of mice at 5 and 30 min post infection as in (A). The inoculum of each group is indicated with a filled circle. The organ CFU in individual mice was presented as mean ± SEM of five to six biological replicates (*n* = 5–6). (C) CRP-independent clearance of capsule-deficient pneumococci. (TH870Δcps) in WT and CRP-deficient mice. WT and CRP knockout mice were i.v. infected with 10$^6$ CFU of TH870Δcps strain. Bacterial loads in organs were quantified 30 min post infection. The blood and organ CFU in individual mice were presented as mean ± SEM of three biological replicates (*n* = 3). *P* value was calculated by Two-way ANOVA (**P* < 0.05). *P* value: 0.0113. (D) Capture of blood bacteria (green) by KCs (red) was visualized by IVM in the context of liver vasculatures (cyan) of *Crp*$^{-/-}$ and WT mice post i.v. infection. Scale bar, 10 μm. KC-associated bacteria per field of view (FOV) were presented as mean ± SEM of five to eight images. *P* value was calculated by Two-way ANOVA with Sidak's multiple comparisons (****P* < 0.0001; ns *P* > 0.05, no significance). *P* values: *Sp*23F CRP+ vs *Sp*23F CRP-, 4.17e−009; *Sp*14 CRP+ vs *Sp*14 CRP-, 0.2486. (E) CRP-mediated KC binding to *Sp*23F was measured in vitro by infecting primary mouse KCs (red) with *Sp*23F or *Sp*14 (green) in the presence of 10% mouse serum from *Crp*$^{-/-}$ or WT mice, and visualized by immunofluorescence microscopy 30 min later. Scale bar, 10 μm. KC-associated bacteria per field of view (FOV) were presented as mean ± SEM of ten images. *P* value was calculated by Two-way ANOVA with Sidak's multiple comparisons (****P* < 0.0001; ns *P* > 0.05, no significance). *P* values: *Sp*23F CRP+ vs *Sp*23F CRP-, 7.88e−017; *Sp*14 CRP+ vs *Sp*14 CRP-, 0.8975. (F) *Sp*23F pneumococci (10$^6$ CFU) were pre-treated with 100 μl r-mCRP (10 μg/ml) for 5 min at room temperature before i.v. inoculation to test bacterial kinetics (left) and organ burden (right). The blood and organ CFU in individual mice were presented as mean ± SEM of six biological replicates (*n* = 6). *P* value was calculated by Two-way ANOVA (****P* < 0.0001). *P* value: 1.93e−037. (G) Protective immunity of CRP against septic infection of *Sp*23F were monitored for 7 days post i.v. infection with 10$^6$ CFU. Survival rates were calculated from six biological replicates (*n* = 6). *P* value was calculated by log-rank test (H) (*Sp*23F WT vs *Sp*23F *Crp*$^{-/-}$) (****P* < 0.0001). *P* value: 4.04e−005. Source data are available online for this figure.

all *Crp*$^{-/-}$ mice succumbed to the infection within 36 hr (Fig. 2G). By comparison, *Crp*$^{-/-}$ mice were fully competent in the defense against *Sp*14. The hyper-susceptibility of *Crp*$^{-/-}$ mice to *Sp*23F was also reflected by significantly higher bacteremia levels in *Crp*$^{-/-}$ mice in the first 24 hr post infection and reduced survival time (Fig. EV3A,B). Importantly, CRP deficiency results in at least an 800-fold decrease in immunity in mice (Appendix Fig. S2). Together, this set of experiments have demonstrated that CRP confers a potent capsule type-specific immunity against systemic infection of serotype-23F *S. pneumoniae*.

## CRP broadly recognizes many capsule types of *S. pneumoniae*

We further tested CRP binding to CPSs of 24 additional pneumococcal serotypes in our collection, which showed the LV phenotype in our previous work (An et al, 2022). ELISA revealed broad binding interactions of mouse CRP with the capsular polysaccharides of many serotypes. r-mCRP bound to free CPSs of 16 additional serotypes: 11A, 15B, 15C, 16F, 17F, 20B, 21, 23A, 23B, 27, 33A, 35A, 35B, 35C, 37 and 41A, but not those of the other eight serotypes (9N, 9V, 10A, 14, 19A, 19F, 34 and 48) (Fig. EV3C). Except for 23B, the early clearance of all the other 15 serotypes was significantly impaired in *Crp*$^{-/-}$ mice although the degrees of the immune deficiency varied among these serotypes, in terms of the CT$_{50}$ value (Figs. 3A and EV3D). For the sake of simplicity, we will refer the serotypes with significant clearance retardation in *Crp*$^{-/-}$ mice to as "CRP-sensitive" bacteria hereafter. Likewise, enumeration of organ bacteria at 5 min post i.v. inoculation revealed significantly decreased capture of the CRP-sensitive serotypes in the livers of *Crp*$^{-/-}$ mice as compared with WT mice (Fig. 3B). Consistently, the vast majority of viable bacteria for the 15 CRP-sensitive serotypes were eliminated to residual levels in WT mice but still retained in *Crp*$^{-/-}$ mice at 30 min post infection (Figs. 3B and EV3E). This result showed that CRP-mediated hepatic capture of these serotypes leads to rapid bacterial killing. IVM examination of the liver sinusoids revealed potent capture of the CRP-sensitive serotypes by KCs of WT mice, but not *Crp*$^{-/-}$

mice (Fig. 3C and Movies EV3–7). Notably, *Crp*$^{-/-}$ mice were equally able to capture and kill serotype-23B pneumococci as WT mice, indicating that this serotype is also recognized by another uncharacterized receptor(s).

Finally, the CRP-sensitive serotypes showed significantly enhanced virulence in *Crp*$^{-/-}$ mice, which is reflected by the reduced survival (Figs. 3D and EV3B) and persistent bacteremia (Fig. EV3A) of *Crp*$^{-/-}$ mice post i.v. inoculation with 10$^6$ CFU of the bacteria. Together, these findings unequivocally demonstrated the great importance of CRP in broad but serotype-specific protection against septic infection of *S. pneumoniae* by enabling KCs to rapidly capture these bacteria in the liver.

## CRP also recognizes the capsules of major Gram-negative pathogens

To understand if CRP recognizes any capsules of Gram-negative bacteria, we first tested the impact of CRP deficiency on the early clearance of *H. influenzae*, an important etiological agent of childhood pneumonia, septicemia and meningitis (Retchless et al, 2024). Among the six *H. influenzae* serotypes, five were rapidly eliminated from the bloodstream of WT mice post i.v. infection (Appendix Fig. S3A). The CT$_{50}$ of serotype-a strain (*Hia*) was moderately elongated from 0.5 min in WT mice to 1.7 min in *Crp*$^{-/-}$ mice (Fig. 4A; Appendix Fig. S3A), which was accompanied by substantial reduction in hepatic trapping of *Hia* at 5 min post infection (Fig. 4A, right panel). *Hia* bacteria were significantly cleared in *Crp*$^{-/-}$ mice at 30 min, indicating the redundant clearance mechanisms against the bacterium in mouse. In contrast, infection of *Crp*$^{-/-}$ mice with *Hib* yielded a much more dramatic phenotype in early clearance. The bacteria were fully maintained in the circulation of *Crp*$^{-/-}$ mice in the first 30 min post infection (Fig. 4B, left panel), and marginally sequestered in the livers at 5 and 30 min (Fig. 4B, right panel). These results demonstrated CRP as an essential host factor in hepatic trapping and clearance at the early infection stage of blood-borne *Hib*, the most dominant serotype of *H. influenzae* in causing invasive infections in young children (Retchless et al, 2024).

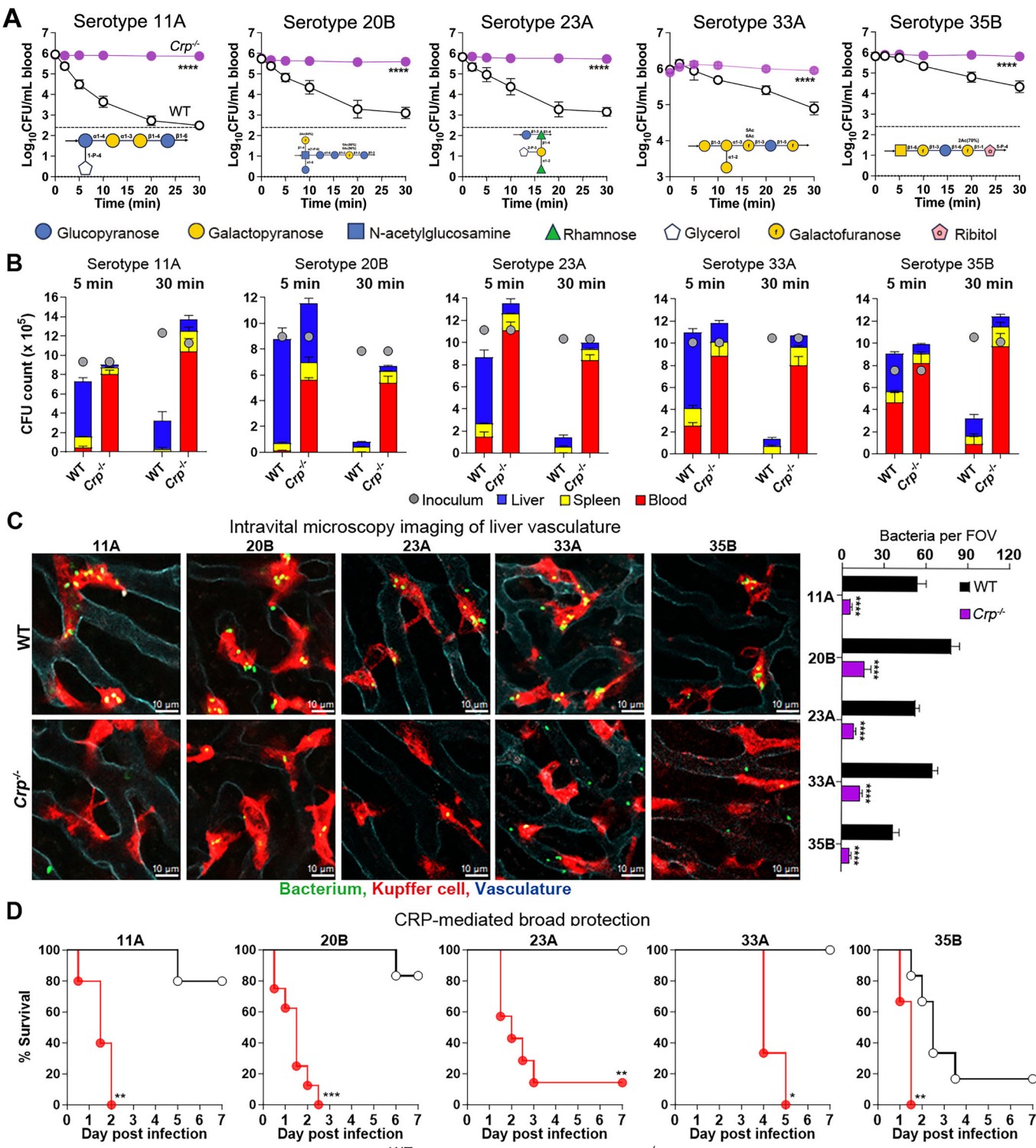

**Figure A–D.** Intravital microscopy imaging of liver vasculature and CRP-mediated broad protection across serotypes 11A, 20B, 23A, 33A, and 35B.

Legend A: Glucopyranose, Galactopyranose, N-acetylglucosamine, Rhamnose, Glycerol, Galactofuranose, Ribitol

Legend B: Inoculum, Liver, Spleen, Blood

Legend C: Bacterium, Kupffer cell, Vasculature — Bacteria per FOV — WT, Crp⁻/⁻

Legend D: WT, Crp⁻/⁻

We further tested the impact of CRP deficiency on the clearance of *K. pneumoniae*, an important Gram-negative pathogen of hospital-acquired pneumonia and sepsis with over 100 capsule types (Paczosa and Mecsas, 2016). Our recent study showed that the LV rather than the HV *K. pneumoniae* strains can be quickly cleared by liver KC, but no capsule receptor is known for any

*K. pneumoniae* serotypes (Huang et al, 2022). The 11 serotypes tested in this trial were rapidly cleared from the bloodstream of WT mice. Besides, the clearance of serotypes K21 (Fig. 4C) and K64 (Fig. 4D) was significantly retarded in *Crp⁻/⁻* mice (Appendix Fig. S3B). This finding revealed that CRP recognizes multiple capsule variants of *K. pneumoniae*.

◀

**Figure 3. CRP-mediated broad protection against many pneumococcal serotypes.**

(A) CRP-based immunity against multiple *S. pneumoniae* serotypes were assessed in WT and $Crp^{-/-}$ mice by blood CFU plating at different time points following i.v. infection with $10^6$ CFU of serotypes 11A, 20B, 23A, 33A, and 35B. The capsule repeat unit of each serotype is displayed. The blood CFU in individual mice was presented as mean ± SEM of three to seven biological replicates ($n$ = 3–7). *P* value was calculated by Two-way ANOVA (****$P$ < 0.0001). *P* value: 11A, 1.25e−034; 20B, 9.82e−013; 23A, 1.21e−014; 33A, 2.58e−009; and 35B, 2.15e−008. (B) Hepatic trapping/killing of CRP-sensitive serotypes. Bacterial burden in the blood, liver and spleen of WT and $Crp^{-/-}$ mice at 5 and 30 min post infection was quantified. The organ CFU in individual mice was presented as mean ± SEM of three to seven biological replicates ($n$ = 3–7). (C) IVM imaging of CRP-mediated KC capture of 5 representative CRP-sensitive serotypes of *S. pneumoniae* in WT and $Crp^{-/-}$ mice as in Fig. 2D. KC-associated bacteria per field of view (FOV) were presented as mean ± SEM of six to eight images. *P* value was calculated by Two-way ANOVA with Sidak's multiple comparisons ($Crp^{-/-}$ vs WT) (****$P$ < 0.0001). *P* values: 11A, 2.25e−011; 20B, 4.91e−016; 23A, 1.43e−010; 33A, 2.23e−013; and 35B, 4.46e−007. (D) The CRP-mediated protection against septic infections of multiple serotypes. WT and $Crp^{-/-}$ mice were i.v. infected with $10^6$ CFU of each serotype, except for serotype 33A ($10^7$ CFU). Survival rates were calculated of three to eight biological replicates ($n$ = 3–8). *P* value was calculated by log-rank test (H) (***$P$ < 0.001, **$P$ < 0.01 and *$P$ < 0.05). *P* values: 11A, 0.0026; 20B, 0.0003; 23A, 0.0026; 33A, 0.0224; and 35B, 0.0046. Source data are available online for this figure.

Gram-negative bacteria captured in the liver sinusoids were further confirmed by IVM imaging. The KC-immobilized *Hib* bacteria were significantly lower in $Crp^{-/-}$ mice than in WT mice (Fig. 4E and Movie EV8). In a similar fashion, $Crp^{-/-}$ mice showed a significantly reduced level of KC-associated *K. pneumoniae* K21 (Fig. 4E and Movie EV9). Specific interactions of CRP with Gram-negative bacteria were confirmed with primary mouse KCs (Fig. 4F). The bacteria of *H. influenzae* type b and *K. pneumoniae* K21 abundantly attached to KCs in the presence of normal serum, but the binding interactions were barely detectable with $Crp^{-/-}$ serum. These imaging analyses showed that CRP promotes serotype-specific KC capture of *H. influenzae* and *K. pneumoniae*.

We further verified the direct binding of CRP to *H. influenzae* and *K. pneumoniae* capsules by ELISA. The experiment revealed that r-mCRP bound to the free CPSs of *H. influenzae* serotypes a and b (Appendix Fig. S3C). In a similar manner, there was serotype-specific binding of r-mCRP to purified CPSs of serotypes-K21 and -K64 *K. pneumoniae* (Appendix Fig. S3D). Since phosphorylcholine on *H. influenzae* LPS is shown to interact with CRP (Weiser et al, 1998), we assessed potential impact of phosphorylcholine on CRP binding to *H. influenzae* capsule by flow cytometry. In contrast to 51.8% CRP-binding bacteria, less than 1% of the cells showed detectable binding to anti-PC antibody (Appendix Figs. S1 and S4). This result showed that the *Hib* strain used in this work did not possess substantial level of phosphorylcholine. Taken together, these experiments uncovered that CRP is a plasma receptor for the capsules of Gram-negative pathogens.

## CRP-opsonized bacteria are delivered to Kupffer cells via multiple pathways

Since CRP is a plasma protein that is not physically associated with KCs, we reasoned that it must indirectly engages KCs for bacterial capture. Previous in vitro studies have shown that human CRP binding to phosphocholine (PC) promotes phagocytosis by binding to complement protein C1q and activating the complement C3 via the classic complement pathway (Haapasalo and Meri, 2019; Kaplan and Volanakis, 1974). Consistently, we detected C3 deposition on CPSs after incubation with WT serum, but not $Crp^{-/-}$ serum (Fig. EV4A). We further determined bacterial clearance in mice lacking C3 ($C3^{-/-}$), a core protein in the complement system (Densen and Ram, 2015). The $C3^{-/-}$ mice showed significant retardation in clearing Sp23F (Fig. 5A, left panel). In a similar pattern, $C3^{-/-}$ mice trapped relatively fewer bacteria in the liver than WT controls at 5 min post infection

(Fig. 5A, right panel). However, the phenotype of $C3^{-/-}$ mice ($CT_{50}$ = 2.07 min) was much milder than $Crp^{-/-}$ mice ($CT_{50}$ > 30 min) (Fig. 2), indicating that C3-independent mechanism functionally bridges CRP and the liver macrophages. To assess the relative contributions of C3-dependent and -independent mechanisms to the CRP-driven immunity, we tested bacterial clearance of $C3^{-/-}$ mice with an elevated infection dose of $5 \times 10^7$ CFU, which WT mice were able to survive (Appendix Fig. S2A). $C3^{-/-}$ mice completely failed to clear Sp23F from the bloodstream (Fig. 5B). Consistently, the absence of C3 led to the loss of hepatic capture of the bacteria. These data revealed that the C3-dependent and -independent mechanisms of CRP function are both capable of eliminating relatively milder form of blood infection, but CRP requires the complement system to combat more severe infections.

C3 is known to promote microbial phagocytosis by engaging complement receptors (Densen and Ram, 2015). Among the four major C3 receptors, mouse KCs express CR3 and CRIg (An et al, 2022), both of which have been shown to promote macrophage uptake of C3-opsonized microbes (Helmy et al, 2006; Walbaum et al, 2021). CR3- and CRIg-deficient mice showed marginally impaired clearance of blood-borne bacteria as compared with WT when infected with $10^6$ CFU bacteria, although CR3 and CRIg double KO mice showed a greater immune deficiency ($CT_{50}$ = 2.09 min) (Fig. 5C). However, the double KO mice were completely unable to clear bacteria at the high infection inoculum (Fig. 5D), which was comparable to $C3^{-/-}$ mice. At the cellular level, IVM imaging also confirmed the critical role of C3 and the C3 receptors in CRP-based KC capture of Sp23F at the high infection dose. The liver sinusoids $C3^{-/-}$ and $CR3/CRIg^{-/-}$ mice showed dramatic reduction in the KC-immobilized bacteria (Fig. 5E and Movie EV10). These observations indicated that KCs employ both CR3 and CRIg to capture of the C3-opsonized Sp23F pneumococci.

The previous studies have shown that human CRP-PC complex activates the complement classical pathway via binding to C1q, the first protein in the cascade of the classical complement pathway (Claus et al, 1977; Kaplan and Volanakis, 1974). We thus tested the phenotype of $C1q^{-/-}$ mice post i.v. infection with $5 \times 10^7$ CFU of Sp23F. In contrast to the dramatic loss of CRP-mediated bacterial clearance in $C3^{-/-}$ mice (Fig. 5B), $C1q^{-/-}$ mice behaved similarly to WT mice (Fig. 5F), demonstrating that CRP activates C3 in a C1q-independent manner in mice. Two articles have reported that CRP interacts with plasma ficolin to generate bactericidal complement proteins via the lectin and classical pathways (Ng et al, 2007; Zhang et al, 2009). We thus tested the contribution of ficolin and the lectin complement activation pathway to the CRP-mediated immunity

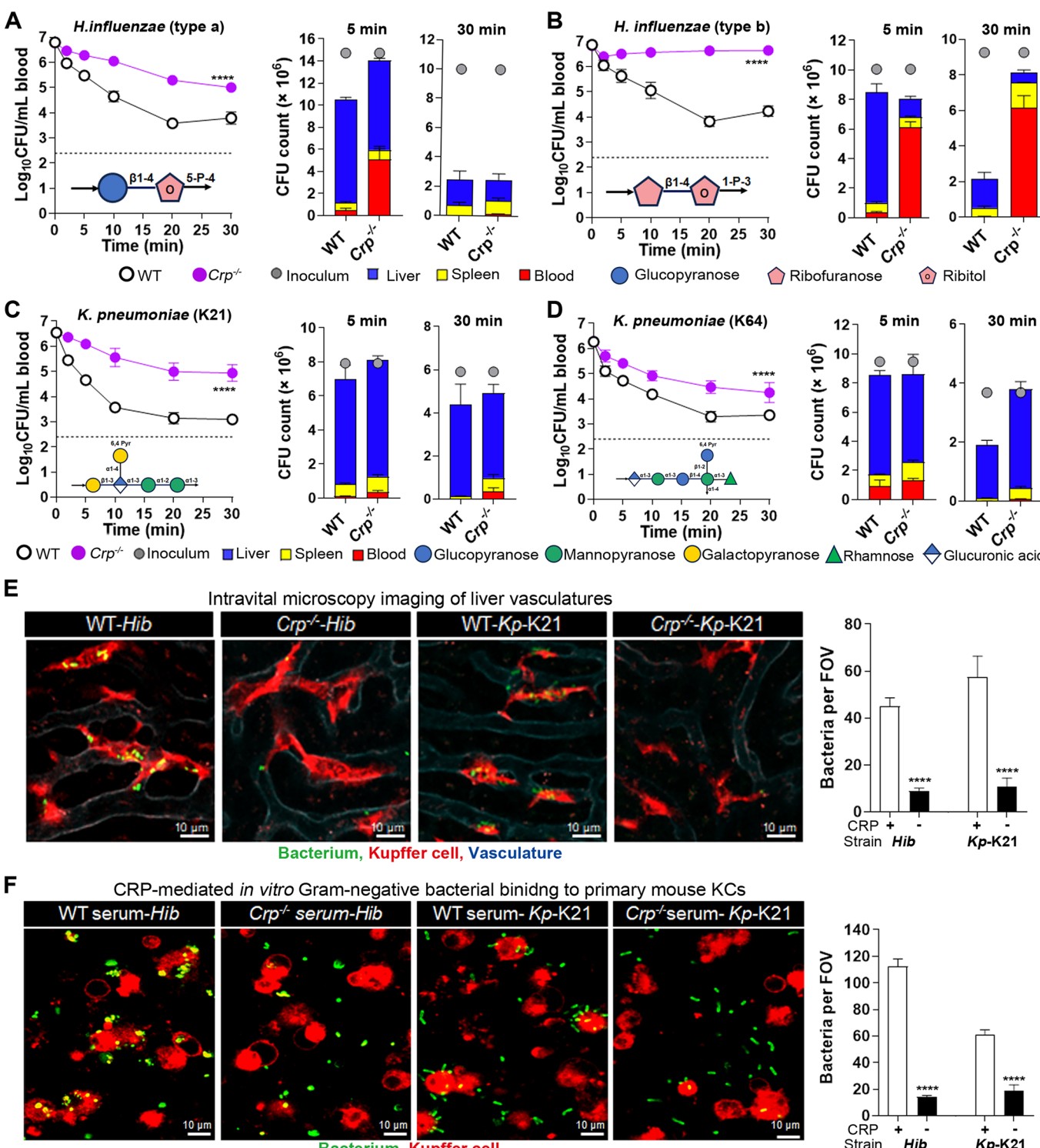

using mice lacking ficolin A (the major ficolin protein in mouse serum), MASP2 (a key component of the lectin pathway) or C4 (an essential factor in both the classical and lectin pathways). CRP-mediated bacterial clearance remained intact in all of the three mouse lines (Fig. EV4B,C). These data showed that ficolin A and the classical/lectin pathways are not involved in the activation of C3 for hepatic capture of CRP-opsonized bacteria in mice.

We finally tested potential involvement of Fcγ receptors (FcγRs) in mediating C3-independent hepatic capture of CRP-opsonized bacteria using mice lacking all the four phagocytosis-associated FcγRs (FcγRI, FcγRIIB, FcγRIII and FcγRIV), based on the literature that CRP binds to FcγRI and FcγRIIA on phagocytes (Bharadwaj et al, 1999; Crowell et al, 1991; Marnell et al, 1995). The FcγR-deficient mice showed normal bacterial clearance when

**Figure 4. CRP-activated hepatic clearance of *H. influenzae* and *K. pneumoniae*.**

(A) CRP-mediated clearance and hepatic trapping (5 min)/killing (30 min) of serotype-a *H. influenzae* were assessed by i.v. infection with $10^7$ CFU as in Figs. 2A and 2B, respectively. The capsule repeat unit is diagrammatically presented below the dash line. The blood and organ CFU in individual mice were presented as mean ± SEM of three biological replicates ($n = 3$). $P$ value was calculated by Two-way ANOVA (****$P < 0.0001$). $P$ value:6.55e−012. (B) CRP-mediated clearance and hepatic trapping (5 min)/killing (30 min) of serotype-b *H. influenzae*. The data were obtained and presented as in (A). The blood and organ CFU in individual mice were presented as mean ± SEM of three biological replicates ($n = 3$). $P$ value was calculated by Two-way ANOVA (****$P < 0.0001$). $P$ value:1.66e−012. (C) CRP-mediated clearance and hepatic trapping (5 min)/killing (30 min) of serotype-K21 *K. pneumoniae*. The data were obtained and presented as in (A). The blood and organ CFU in individual mice were presented as mean ± SEM of three to five biological replicates ($n = 3$–5). $P$ value was calculated by Two-way ANOVA (****$P < 0.0001$). $P$ value:1.39e−014. (D) CRP-mediated clearance and hepatic trapping (5 min)/killing (30 min) of serotype-K64 *K. pneumoniae*. The data were obtained and presented as in (A). The blood and organ CFU in individual mice were presented as mean ± SEM of three to five biological replicates ($n = 3$–5). $P$ value was calculated by Two-way ANOVA (****$P < 0.0001$). $P$ value: 1.15e−006. (E) IVM imaging of CRP-mediated capture of Gram-negative pathogens by KCs. The capture of serotype-b *H. influenzae* (left panel) and serotype-K21 *K. pneumoniae* (right panel) by KCs in the liver sinusoids of $Crp^{-/-}$ and WT mice were visualized and quantified by IVM as in Fig. 2D. KC-associated bacteria per field of view (FOV) were presented as mean ± SEM of six to seven images. $P$ value was calculated by Two-way ANOVA with Sidak's multiple comparisons (****$P < 0.0001$). $P$ values: Hib CRP+ vs Hib CRP-, 5.63e−005; Kp-K21 CRP+ vs Kp-K21 CRP-, 2.50e−006. (F) CRP-mediated in vitro KC binding to serotype-b *H. influenzae* and serotype-K21 *K. pneumoniae* was assessed as in Fig. 2E. KC-associated bacteria per field of view (FOV) were presented as mean ± SEM of nine to ten images. $P$ value was calculated by Two-way ANOVA with Sidak's multiple comparisons (****$P < 0.0001$). $P$ values: Hib CRP+ vs Hib CRP-, 1.62e−020; Kp-K21 CRP+ vs Kp-K21 CRP-, 1.15e−010. Source data are available online for this figure.

infected with $10^6$ CFU (Fig. 5G) or $5 \times 10^7$ CFU (Fig. 5H) of *Sp*23F. These data showed that mouse Fcγ receptors are not involved in the C3-independent mechanism of CRP-activated bacterial clearance.

## CRP recognizes diverse polysaccharide structures via distinct binding modes

The CRP-binding capsules greatly differ in sugar composition, length and branch of repeating unit (Fig. 6A; Appendix Fig. S5). Previous studies have revealed that CRP binds to phosphocholine (PC) and phosphoglycerol (PG) groups in a pocket around the $Ca^{2+}$ binding sites (Goda and Miyahara, 2017; Shrive et al, 1996; Thompson et al, 1999). We determined the binding affinity between CRP and the capsules with phosphocholine (PC) (CPS27), phosphoglycerol (PG) (CPS23F) or none of the either group (CPS33A) by isothermal titration calorimetry (ITC). CPS14 was used as a negative control. The results showed that mCRP exhibited the highest affinity to CPS23F with a dissociation constant $K_D$ of $2.46 \times 10^{-7}$ M (Fig. 6B; Appendix Fig. S6). Besides, the binding stoichiometry of N = 4.71 suggested a molecular ratio of 1:5 in the complex between mCRP and CPS23F ligands. A relatively lower binding affinity was obtained with CPS27 ($K_D = 1.1 \times 10^{-5}$ M) and CPS33A ($K_D = 4.9 \times 10^{-4}$ M). These findings suggest that the presence of PC is not sufficient to confer high affinity binding. In contrast, CRP preferentially binds to capsules containing PG, suggesting that PG plays a more critical role in mediating high-affinity interactions with CRP.

To further define the capsule-binding modes of CRP, we tested the capacity of free PC and PG in competitively inhibiting the binding interactions between CRP and capsules with PC (CPS27), PG (CPS23F) or neither group (CPS33A). Competitive ELISA showed that free PC prevented mCRP from binding to CPS27 in a dose-dependent manner, suggesting that CRP recognizes PC and the PC-containing capsule in a similar mechanism (Fig. 6C). In contrast, free PC showed much weaker inhibition against mCRP binding to the PC-free CPS23F and CPS33A. The median effective concentration of inhibition ($EC_{50}$) for CPS23F indicates that a very high concentration is required to achieve effective inhibition. The $EC_{50}$ for CPS27 was 4.62 folds lower than that those for CPS33A. Surprisingly, free PG was more potent in inhibiting mCRP binding

to CPS27 ($EC_{50} = 0.07$ mM) than PC ($EC_{50} = 0.52$ mM) (Fig. 6D). Likewise, PG also showed more effectively blockage against PC-free CPS23F and CPS33A than PC. In particular, PG prevented mCRP from binding to the PG-containing (CPS23F) and PG-free (CPS33A) capsules in a similar manner. These results suggest that PG plays a more prominent role than PC in CRP binding, with CRP employing different binding strategies depending on the bacterial capsule component.

To characterize the molecular basis of the differences in mCRP binding to capsular polysaccharides, we constructed mCRP point mutants in the phenylalanine residue 66, glycine residue 76 and glutamic acid residue 81 based on their location at the PC-binding surface of human CRP (Ramadan et al, 2002; Thompson et al, 1999). The ELISA result showed the G76Y mutant showed a similar binding to CPS23F, CPS27 and CPS33A as the WT form (Fig. 6E). Besides, the F66A and E81A mutation had significant impairment in mCRP binding to all the three capsules, but its impact on the binding activity of the PC-containing CPS27 and PC/PG-free CPS33A was far more severe than on that of the PG-containing capsule (CPS23F) (Fig. 6E). Together, these data have revealed that mCRP recognizes the diverse structures of bacterial capsules through related but distinct binding modes, with critical binding sites for PC playing different roles in the recognition of various types of capsules.

## Human CRP mediates broad serotype-specific recognition of capsules

To understand the functional significance of human CRP in enabling hepatic clearance of encapsulated bacteria, we determined if hCRP shares the capsule-binding features of mCRP. ELISA results showed significant binding of hCRP to all 17 mCRP-binding capsules of *S. pneumoniae* (Fig. 7A), which highlights a shared characteristic of human and mouse CRPs in their role in anti-pneumococcal immunity. Surprisingly, no significant binding was detected between hCRP and the four mCRP-binding capsules of Gram-negative bacteria (Fig. 7A; Appendix Fig. S7). This result revealed species-specific features of CRP orthologs in the pathogen recognition. We further verified the ability of human CRP in promoting hepatic bacterial capture using primary human KCs. As compared with the poor binding of serotype-8 pneumococci to

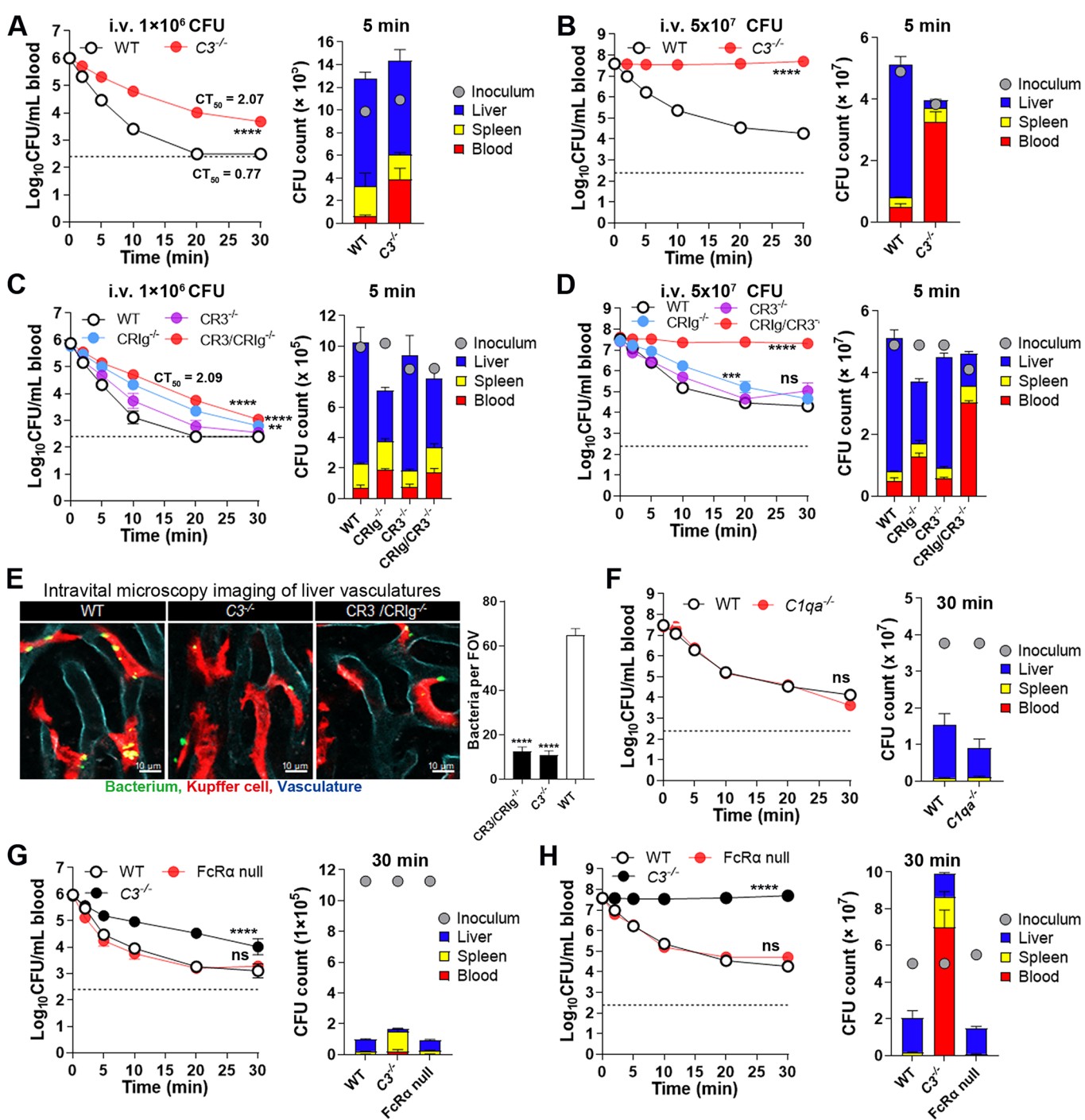

human KCs, *Sp*23F bacteria showed a low but significant level of adhesion to human primary KCs in the presence of normal human serum (Fig. 7B). This activity was further augmented by r-hCRP. Meanwhile, *S. pneumoniae* serotype 23F specifically activated complement in normal human serum (NHS), with 93.4% C3 deposition, whereas serotype 8 exhibited minimal deposition (35.1%) (Fig. 7C). Meanwhile, the addition of exogenous CRP does not further increase C3 deposition due to the already high level of C3 activation and the CRP-CPS binding is inhibited by CPS23F, but not by CPS8 (Appendix Fig. S8). Therefore, these results strongly suggested that human CRP also acts as a plasma

receptor for structurally diverse capsules to promote hepatic capture of blood-borne bacteria.

We further determined the binding affinity of hCRP to five pneumococcal capsule types (Appendix Fig. S6). hCRP showed the strongest binding to the PC-containing CPS27 with a dissociation constant of $K_D$, 1.7 μM, which is comparable to the reported affinity between hCRP and the well-characterized ligand PC ($K_D$, 5 μM) (Christopeit et al, 2009). However, except for CPS27, the binding affinities of hCRP to the other four capsules are generally lower than those of mCRP, except CPS33A. As an example, the CPS11A-binding affinity of mCRP ($K_D$, 0.29 μM) is 100 folds of that of hCRP

◄ **Figure 5.  The role of C3 in the CRP-based immunity.**

(A, B) The contribution of C3 to CRP-mediated immunity was evaluated by CFU plating of the blood (left) and liver (right) in $C3^{-/-}$ mice infected with a low (A) or high (B) dose of *Sp*23F as in Figs. 2A and 2B. The blood and organ CFU in individual mice were presented as mean ± SEM of three to six biological replicates ($n = 3$–6). $P$ value was calculated by Two-way ANOVA (****$P < 0.0001$). $P$ value: $10^6$, 8.70e−023; $5 \times 10^7$, 1.46e−024. (C, D) The role of C3 receptors CRs and CRIg in CRP-mediated immunity was assessed by CFU plating of the blood (left) and liver (right) in $CR3^{-/-}$ and/or $CRIg^{-/-}$ mice infected with $10^6$ and $5 \times 10^7$ CFU of *Sp*23F, as in (A) and (B), respectively. The blood and organ CFU in individual mice were presented as mean ± SEM of three biological replicates ($n = 3$–9). $P$ value was calculated by Two-way ANOVA with Tukey's multiple comparisons (****$P < 0.0001$, ***$P < 0.001$, **$P < 0.01$ and ns $P > 0.05$, no significance). $P$ value in (C): $CRIg^{-/-}$ vs WT, 1.35e−014 $CR3^{-/-}$ vs WT, 0.0038; $CR3/CRIg^{-/-}$ vs WT, 1.0e−015; $P$ value in (D): $CRIg^{-/-}$ vs WT, 0.0002; $CR3^{-/-}$ vs WT, 0.3631; $CR3/CRIg^{-/-}$ vs WT, 9.95e−012. (E) Visualization of CRP-mediated pathogen capture by KCs was performed in mice lacking C3 or C3 receptors as in Fig. 2D. KC-associated bacteria per field of view (FOV) were presented as mean ± SEM of five to eight images. $P$ value was calculated by One-way ANOVA with Sidak's multiple comparisons (****$P < 0.0001$). $P$ values: $CR3/CRIg^{-/-}$ vs WT, 4.44e−010; $C3^{-/-}$ vs WT, 6.15e−010. (F) The role of C1q in CRP-mediated immunity were tested in $C1qa^{-/-}$ mice as in (B). The blood and organ CFU in individual mice were presented as mean ± SEM of three biological replicates ($n = 3$). $P$ value was calculated by Two-way ANOVA (ns $P > 0.05$, no significance). $P$ value, 0.4685. (G, H) The roles of Fcγ receptors in CRP-mediated immunity were tested in FcRα null mice with the low- (G) or high-(H) inoculum of *Sp*23F as in (A) and (B), respectively. The blood and organ CFU in individual mice were presented as mean ± SEM of three biological replicates ($n = 3$). $P$ value was calculated by Two-way ANOVA with Tukey's multiple comparisons (****$P < 0.0001$, ns $P > 0.05$, no significance). $P$ value in (G): $C3^{-/-}$ vs WT, 3.82e−008 Fcαn-null vs WT, 0.4042; $P$ value in (H): $C3^{-/-}$ vs WT, 1.0e−015, Fcαn-null vs WT, 0.6525. Source data are available online for this figure.

($K_D$, 27.90 μM), which may contribute to human tropism of *S. pneumoniae*. To define the molecular details of hCRP binding to capsular polysaccharides, we incubated hCRP and segmented CSP23F in detergent containing buffer to avoid preferential orientation. The complex structure of hCRP and CPS23F by cryo-electron microscopy was resolved at a resolution of ~2.78 Å (Appendix Fig. S9; Appendix Table S4). The EM structure revealed the five well-defined and identical non-covalently bound subunits of hCRP, showing clear structural features of CRP (Appendix Fig. S10). While C5 symmetry was applied during reconstruction, only partial modeling of a single CSP23F repeat unit was achievable, primarily due to ligand flexibility. The density map clearly resolved two sugar moieties (β-L-Rhap and β-D-Galp) and the PG group, while the remaining two sugar rings (α-L-Rhap and β-D-Glcp) exhibited no interpretable density (Figs. 7D and EV5). Notably, the PG group maintained consistent structural features across all five protomers (Appendix Fig. S10). In principle, the primary interactions between hCRP and CPS23F are similar to what is described for the hCRP-PC binding (Lee et al, 2002; Thompson et al, 1999). The CPS binding site is formed by residues D60, N61, E138, Q139, D140, E147, and Q150 of hCRP, which are coordinated by two $Ca^{2+}$ atoms (Fig. 7E).

To define the precise roles of individual amino acids of hCRP in capsule recognition, we tested the impact of F66A, T76Y, and E81A mutation. While F66 and G76 were not essential for capsule recognition of mouse CRP (Fig. 6E), the F66A mutant completely lost the binding to CPS23F, CPS27 and CPS33A (Appendix Fig. S11B), indicating that these residues are placed in more crucial positions in hCRP. hCRP E81A variant showed capsule type-dependent variations as observed with the mCRP counterpart. The E81A mutants of both human and mouse CRPs showed much more severe impairment in binding to CPS33A than to CPS27 and CPS23F. In the context of undetectable binding of human CRP to the four Gram-negative capsules that are recognized by mCRP (Appendix Fig. S7), these mutagenesis results revealed functional commonality and differences between human and mouse CRPs in capsule recognition.

To determine the molecular basis of the selective capsule binding of human and mouse CRPs, we predicted the structure of mCRP and mCRP-PC/PG complex by Alphafold3 (Fig. 7F–H). Comparative analyses suggested major differences in the pocket that accommodates the capsular PC and PG groups. The pocket is overall based on residues 61–91, which form three antiparallel

helices and two short loops (loops 69–72 and 76–81). The loop 68–72 has a flipped conformation in mCRP compared with that of hCRP, resulting in distinct ligand-binding pocket geometries.

In addition, in the loop 76–81, a replacement of the threonine residue at position 76 with glycine in mCRP significantly alters the conformation of E81. G76 of mCRP would allow E81 to establish close contacts with the bound ligands. However, T76 of hCRP would push E81 away from the bound ligand. This result suggested that residues 71–91 mainly determine the functional difference between human and mouse CRPs. This notion was tested by constructing a human-mouse hybrid CRP - hCRP^m71-91, in which amino acids 71–91 of hCRP were replaced with those of mCRP. In contrast to the undetectable binding of hCRP to *Hib* capsule, hCRP^m71-91 gained the capsule-binding capability of mouse CRP (Fig. 7I). This result verified that the differences in amino acids 71–91 of human and mouse CRPs are primarily responsible for the mouse-specific binding to the *H. influenzae* capsule. The relatively lower affinity of hCRP^m71-91 to CPS-*Hib* ($EC_{50} = 7.98$ μg/ml) than that of mouse CRP ($EC_{50} = 0.56$ μg/ml) suggested the contribution of additional amino acids of mCRP beyond the 71–91 region to capsule binding, which is supported by the reduced binding of hCRP^m71-91 to CPS23F (Appendix Fig. S7C). Together, these data have revealed that human and mouse CRPs share the common structural principles of capsule recognition, with certain species-specific differences that may explain host-specific tropism and virulence of encapsulated bacteria.

## Discussion

CRP and a number of other plasma proteins have been shown to bind to pathogenic bacteria in vitro, but their precise roles in the immune clearance of blood-borne bacteria remain undefined. This study has demonstrated that circulating CRP binds to a wide range of bacterial capsules and thereby enables KCs to capture encapsulated Gram-positive and -negative bacteria. These molecular interactions enable the liver macrophages to swiftly capture and kill blood-borne pathogens in the liver sinusoids, and confer remarkable levels of protection against septic infections. CRP-deficient mice became hyper-susceptible to septic infections of the CRP-recognizable capsule types. Due to the evolutionary conservation of CRP (Pathak and Agrawal, 2019), the CRP-mediated

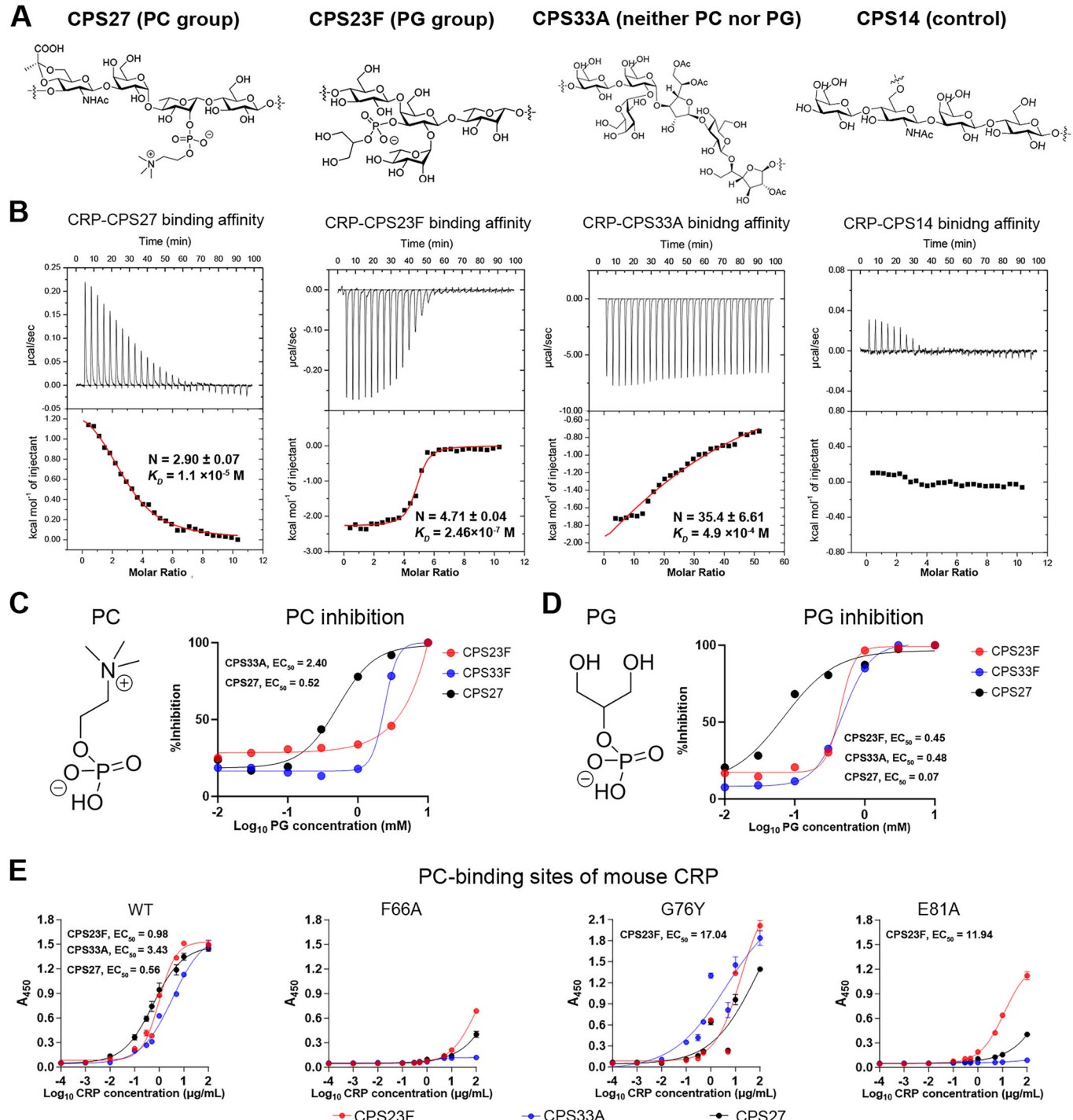

Figure 6. Molecular basis of capsule recognition by mouse CRP.

(A) The repeat unit structures of the PC-containing (CPS27), PG-containing (CPS23F), PC/PG-free (CPS33A) and negative control (CPS14) capsules of *S. pneumoniae*. (B) The binding affinity of CRP to CPS23F, CSP27, CP33A and CPS14 was measure by ITC. A repeating unit of the CPS will be regarded as a single molecule during ITC experiment, since bacterial CPSs are macromolecules with undefined number of repeating units. The dissociate constant ($K_D$), binding enthalpy ($\Delta H$) and stoichiometry (N) are shown after base-line integration and concentration normalization. CPS14 was used as a negative control. (C) Competitive inhibition of mCRP-capsule interactions by PC was tested by preincubating r-mCRP with various concentrations of PC (structure shown at the left panel) before being added to CPS-coated wells and detecting CPS-bound CRP (right panel). $n = 3$. (D) Competitive inhibition of mouse CRP-capsule interactions by free PG as in (C). $n = 3$. (E) Capsule binding of mCRP mutants was tested by ELISA using various concentrations of WT and mutant mCRP and plate wells coated with CPS23F, CPS33A, and CPS27 as in Fig. 1D. $A_{450\ nm}$ were presented as mean $\pm$ SEM of three biological replicates ($n = 3$). Source data are available online for this figure.

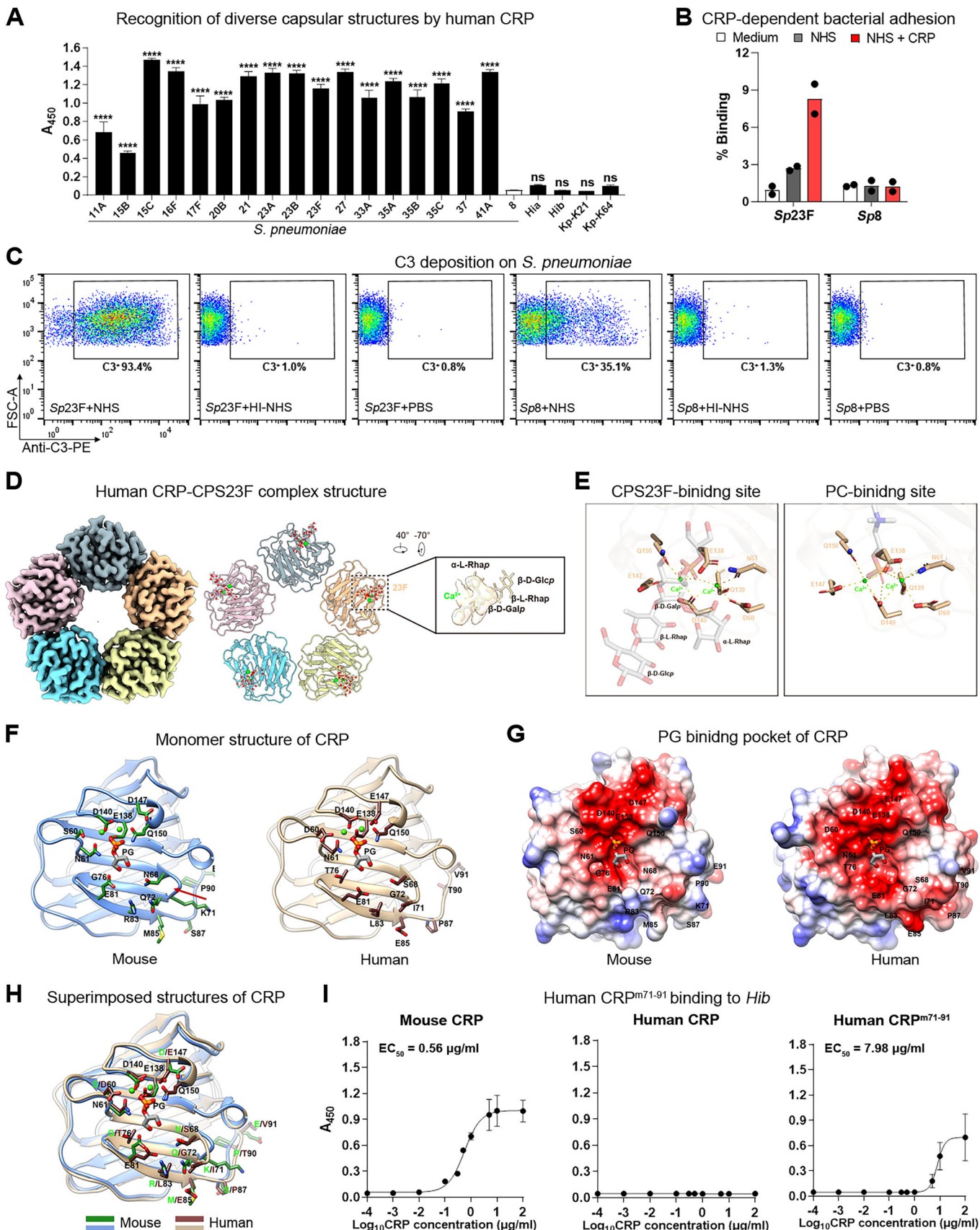

**A** Recognition of diverse capsular structures by human CRP

**B** CRP-dependent bacterial adhesion

**C** C3 deposition on *S. pneumoniae*

**D** Human CRP-CPS23F complex structure

**E** CPS23F-binidng site      PC-binidng site

**F** Monomer structure of CRP

**G** PG binidng pocket of CRP

**H** Superimposed structures of CRP

**I** Human CRP^m71-91 binding to *Hib*

**Figure 7.  Immune recognition of structurally diverse capsules by human CRP.**

(A) hCRP binding to the capsules of *S. pneumoniae*, *H. influenzae*, and *K. pneumoniae* was assessed by ELISA as in Fig. 1J. $A_{450\ nm}$ were presented as mean ± SEM of three biological replicates ($n = 3$). $P$ value was calculated by One-way ANOVA with Sidak's multiple comparisons vs *S. pneumoniae* 8 (****$P < 0.0001$, ns $P > 0.05$, no significance). $P$ value: *Sp*11A, 1.15e−10; *Sp*15B, 5.98e−006; *Sp*15C, 2.99e−023; *Sp*16F, 1.16e−021; *Sp*17F, 2.25e−016; *Sp*20B, 3.99e−017; *Sp*21, 7.10e−021; *Sp*23A, 1.75e−021; *Sp*23B, 2.48e−021; *Sp*23F, 5.44e−019; *Sp*27,1.53e−021; *Sp*33A, 1.88e−017; *Sp*35A, 3.81e−020; *Sp*35B, 1.34e−017; *Sp*35C, 7.20e−020; *Sp*37, 5.44e−015; *Sp*41A, 1.63e−021; *Hia*, 0.999978; *Hib*, 0.9999; *Kp*-K21, 0.9999; *Kp*-K64, 0.9999. (B) CRP-mediated bacterial binding to primary human primary KCs was tested in the presence of 10% normal human serum (NHS) alone or in combination with r-hCRP (NHS + CRP, 10 μg/ml). No serum (medium) served as a negative control. KC-bound bacteria in each well are present as the ratio with free bacteria. $n = 2$. (C) C3 deposition on the intact pneumococci. Serotype-23F and -8 bacteria ($10^6$ CFU) were incubated with normal human serum (NHS), heat-inactivated human serum (HI-NHS), or PBS at 37 °C for 30 min, stained with PE-conjugated anti-human C3 antibody, and analyzed by flow cytometry. (D) Cryo-EM density map (left panel, contoured at 3σ) and atomic model (middle panel) of the hCRP-CPS23F complex, with enlarged view of the EM density and atomic model for ligand (right panel, contoured at 3σ). Five protomers are indicated with different colors; the two bound calciums shown as green spheres. (E) Contact interface between human CRP and CPS23F. The left panel displays the molecular moieties of CPS23F (gray), key amino acids of hCRP (khaki) and two calciums (green) in the protein-polysaccharide complex. The previously published contact interface of the hCRP with PC (PDB: 1B09) is shown at the right panel as a reference. (F) Structural comparison of hCRP-PG (brown; PDB: 1b09) and mCRP-PG (dark green; predicted by AlphaFold3) complexes. The bound PG and residues involved in calcium binding and around the PG binding pocket are shown in sticks with the O, P and N atoms colored red, orange and blue, respectively. The C atoms in the residues are colored dark green in mCRP and brown in hCRP. The red arrows indicate the loop 68–72, in which residue 71 flips in mCRP compared to that in hCRP. The ligand PG in mCRP was generated by superimposition of the hCRP-PG complex onto the predicted mCRP structure. (G) Surface rendered representation showing the binding pocket of PG. The surface is colored according to the surface electrostatic potential. The bound PG is shown in stick. (H) The superimposed structures of mCRP and hCRP. The polymorphic residues in human and mouse CRPs are indicted by dark red and green letters, respectively. (I) *Hib* capsule binding to the hybrid human-mouse CRP (hCRP$^{m71-91}$) was determined and presented as in (A), with normal hCRP and mCRP as controls. $A_{450\ nm}$ were presented as mean ± SEM of three biological replicates ($n = 3$). Source data are available online for this figure.

pathogen recognition may represent a broad immune mechanism against encapsulated bacteria and other microbes with surface-exposed saccharide molecules.

## CRP drives a potent innate immunity against a broad spectrum of encapsulated bacteria

Many studies have attempted to define an anti-infection function of CRP in mice by passive administration or transgenic expression of human CRP, because it is believed that the low levels of endogenous CRP in mouse plasma (<2 μg/ml) are insufficient to manifest the function of the copious human CRP during bacterial infections (Bray et al, 2016; Ji et al, 2023; Ngwa and Agrawal, 2019). Human CRP is reported to protect mice against *S. pneumoniae* infection by binding to the cell wall PC (Mold et al, 1981; Ngwa et al, 2020a; Ngwa et al, 2020b; Szalai et al, 1995; Yother et al, 1982), whereas other lines of evidence have cast some doubt on the immunological significance of the CRP-PC interaction in host defense (An et al, 2025; Holzer et al, 1984). This work has shown that the natural level of CRP in mice confers a robust immunity against septic infections of CRP-recognizable encapsulated bacteria. This CRP-based immunity resembles what is induced by circulating natural antibodies recognizing lipopolysaccharides of *E. coli* (Zeng et al, 2018) and capsular polysaccharides of *S. pneumoniae* (Tian et al, 2025). These antibodies shuffle blood-borne bacteria to KCs through both the complement-dependent and independent mechanisms (Tian et al, 2025; Zeng et al, 2018). Likewise, the antibodies induced by pneumococcal capsular polysaccharide vaccine enable KCs to capture pneumococci via the activation of the complement system (Wang et al, 2023). In this regard, CRP acts like an anti-capsule antibody, with a much broader spectrum of pathogen recognition.

The potency of the CRP-mediated immunity varies among the capsule types. Among the 21 CRP-recognizable capsule types identified in this work, CRP is fully required for the early clearance of 13 types in the liver, and partially involved in the clearance of the other 7 types. This result indicates that CRP is the sole receptor for many capsule types. The partial clearance of certain CRP-binding

capsule types in CRP-deficient mice indicates that these capsules are also recognized by additional receptor(s). This is exemplified by the full dispensability of CRP in the hepatic clearance of serotype-23B *S. pneumoniae*. Our discovery of CRP as a plasma receptor for many bacterial capsules also explains why these types of bacteria show the LV phenotype in mouse sepsis models in our previous studies (An et al, 2022; Huang et al, 2022).

Our limited screening has identified 21 CRP-binding capsules from the 41 LV serotypes tested thus far. Except for serotype-23B *S. pneumoniae*, CRP-deficient mice showed a significant impairment in the hepatic clearance of all the CRP-binding serotypes. Mice lacking CRP were much more susceptible to septic infections of all the CRP-recognizable capsule types. While the precise spectrum of the CRP-mediated immunity remains to be defined, it is reasonable to expect that CRP recognizes additional capsule types beyond those discovered in this work. Despite the broad coverage of encapsulated bacteria, the CRP-mediated immunity is highly specific only to the CRP-binding capsule structures/serotypes, CRP-deficient mice still retained normal hepatic immunity against the non-CRP-binding serotypes.

## CRP engages Kupffer cells to clear blood-borne bacteria via multiple immune pathways

Consistent with our previous observations (An et al, 2022; Huang et al, 2022), IVM imaging showed that KCs of CRP-deficient mice completely lost the ability to capture the CRP-sensitive capsule types in the liver sinusoids. This finding is consistent with the dominant role of the liver resident macrophages in capturing blood-borne bacteria in the naïve (An et al, 2022; Tian et al, 2025; Zeng et al, 2018), and vaccinated (Wang et al, 2023) hosts. At the molecular level, our data with different infection doses have revealed that CRP accomplishes the hepatic trapping of circulating bacteria through the complement-dependent and independent mechanisms. This state is supported by a partial impairment of mice lacking C3 or C3 receptors in bacterial clearance at the relatively lower level of infection. Similar complement-dependent and independent modes of action have been described for the

natural antibody-mediated bacterial capture by KCs (Tian et al, 2025; Zeng et al, 2018), but the precise details are largely undefined.

The involvement of the complement system in CRP-driven immunity agrees with the existing literature that in vitro binding of human CRP to PC and other related substrates activates the complement system (Kaplan and Volanakis, 1974; Mortensen et al, 1976). The complement-dependent bacterial capture required the CRP-initiated activation of C3 on bacterial surface and subsequent C3 binding interactions with the C3 receptors on KCs. In agreement with the dominant expression of CRIg and CR3 by KCs (An et al, 2022), mice lacking CRIg and CR3 showed a similar level of deficiency as $C3^{-/-}$ mice in bacterial capture. A remaining question is how CRP activates C3 upon bindings to the capsules. Human CRP-PC complex has been shown to activate the complement classical pathway via binding to C1q (an essential factor of the classical pathway) under the in vitro conditions (Agrawal et al, 2001). Surprisingly, $C1q^{-/-}$ mice did not show obvious defect in the hepatic clearance of CRP-sensitive bacteria (Fig. 5). This phenotype is not due to any additional genetic changes in the $C1q^{-/-}$ mice because this mouse line showed the expected defect in the antibody-mediated activation of the classical pathway in our previous work (Tian et al, 2025). Likewise, mice lacking C4 (a key component of the classical pathway downstream of C1q) behaved like WT mice in bacterial clearance. These data strongly argue that CRP activates C3 in mice beyond the classical complement pathway.

CRP of horseshoe crab has been reported to activate the complement system on bacteria via the MASP-dependent lectin pathway by interacting with ficolin, a C-type lectin (Ng et al, 2007). Similarly, human serum CRP is reported to engage human L-ficolin, and activate the classical and lectin pathways, leading to bactericidal activity (Zhang et al, 2009). Our data showed that ficolin A (the major mouse plasma ficolin) is not required for the CRP-driven bacterial clearance. Likewise, the lack of MASP2 (an essential factor of the lectin pathway) or C4 (an essential component of the classical/lectin pathways) did not obviously impact the hepatic clearance of *Sp*23F. These discrepancies may reflect species specificity in the activities of CRP, ficolin and the complement system. Taken together, the complement system represents an important functional linkage between CRP and the hepatic anti-bacterial machinery (Fig. 8), but the precise mechanism(s) behind the action of C3 awaits for further investigation.

It remains to be determined how CRP activates the complement-independent mode of bacterial capture. Human CRP has been shown to bind to FcγRI- or FcγRIIA-transfected cells in vitro (Bharadwaj et al, 1999; Crowell et al, 1991). Surprisingly, none of the major Fcγ receptors are involved in the CRP-mediated immunity in mice since the animals lacking these Fcγ receptors kept the normal CRP-based immunity. It thus appears that an uncharacterized receptor(s) on KCs receives the CRP-opsonized bacteria. Alternatively, CRP may activate a C3-like adapter molecule in the plasma, which in turn engages KCs.

## Sequence variations between human and mouse CRPs influence the substrate specificities of the capsule receptor and host specificity of encapsulated bacteria

Consistent with the evolutionary conservation of CRP, the human and mouse orthologs commonly recognize all of the 17 pneumococcal serotypes identified in this work. Our literature search did not find strict correlation between CRP recognition and epidemiological prevalence of pneumococcal serotypes. As an example, the CRP-recognizable serotype 23F was one of the most prevalent pneumococcal serotypes in causing invasive pneumococcal disease of young children in the US before the introduction of the 7-valent pneumococcal conjugate vaccine (Belman et al, 2024). This discrepancy may be caused by individual-to-individual variations among human populations in the downstream molecular and cellular factors of CRP-mediated immunity, including the inflammation-dependent induction of CRP, state of the complement system and functional maturity of KCs. Seasonal respiratory co-infections represent an important external factor that may compromise the CRP-mediated immunity. Our recent study has shown that pre-infection of influenza virus compromises the CRP-mediated bactericidal immunity neutrophils in the lungs by impairing the expression of the complement receptor CR3 and thereby enhances the virulence levels of serotype-23F *S. pneumoniae* and type-b *H. influenzae* (Fang et al, 2024).

A major functional difference between human and mouse CRPs is their interactions with the four capsules of *H. influenzae* (types a and b) and *K. pneumoniae* (types K21 and K64). While mouse CRP recognizes these capsules, human CRP does not show obvious binding. When comparing human and mouse CRP, we observed significant differences in their binding capacities due to amino acid variations at key positions, such as the residue at position 76. In mouse CRP, the glycine at position 76 allows for the continued recognition of CPS23F, even after the G76Y mutation, indicating that this residue is less critical for binding. In contrast, in human CRP, the threonine at position 76 plays a crucial role in CPS23F recognition, and a mutation at this position (T76A) severely impairs the protein's ability to bind to the capsule. This difference highlights the role of specific residues, such as T76 in human CRP and G76 in mouse CRP, in modulating the protein's ability to recognize bacterial capsules. These findings underscore the importance of understanding the functional roles of equivalent residues in human and mouse CRP, as these structural differences may explain the species-specific differences in CRP-mediated immunity.

Our cryo-EM structural analysis has provided a molecular basis for CRP recognition of the PG-containing capsules. CRP binds to the PG and PC groups in a similar manner via electric charge-based interaction between the positive-charged calciums in each CRP protomer and the negative-charged phosphate group in PG. While it is difficult to explain how CRP recognizes the vast majority of the PG-free capsules, the charge-based molecular attraction may operate for certain capsule types with the phosphate groups in their repeat units. The repeat units of CRP-binding serotypes a and b of *H. influenzae* each possess a phosphate group, which may interact with the CRP-based calciums. However, the phosphate group alone in capsular polysaccharide is not sufficient for CRP binding because certain capsules contain the chemical group does not bind to CRP (e.g., serotypes c and f of *H. influenzae*). Further studies are warranted to fully understand the structural basis of the broad CRP-capsule interactions. In this regard, diverse CRP interactions with bacterial capsules can serve as informative models to elucidate the general molecular principles governing protein-polysaccharide binding recognition.

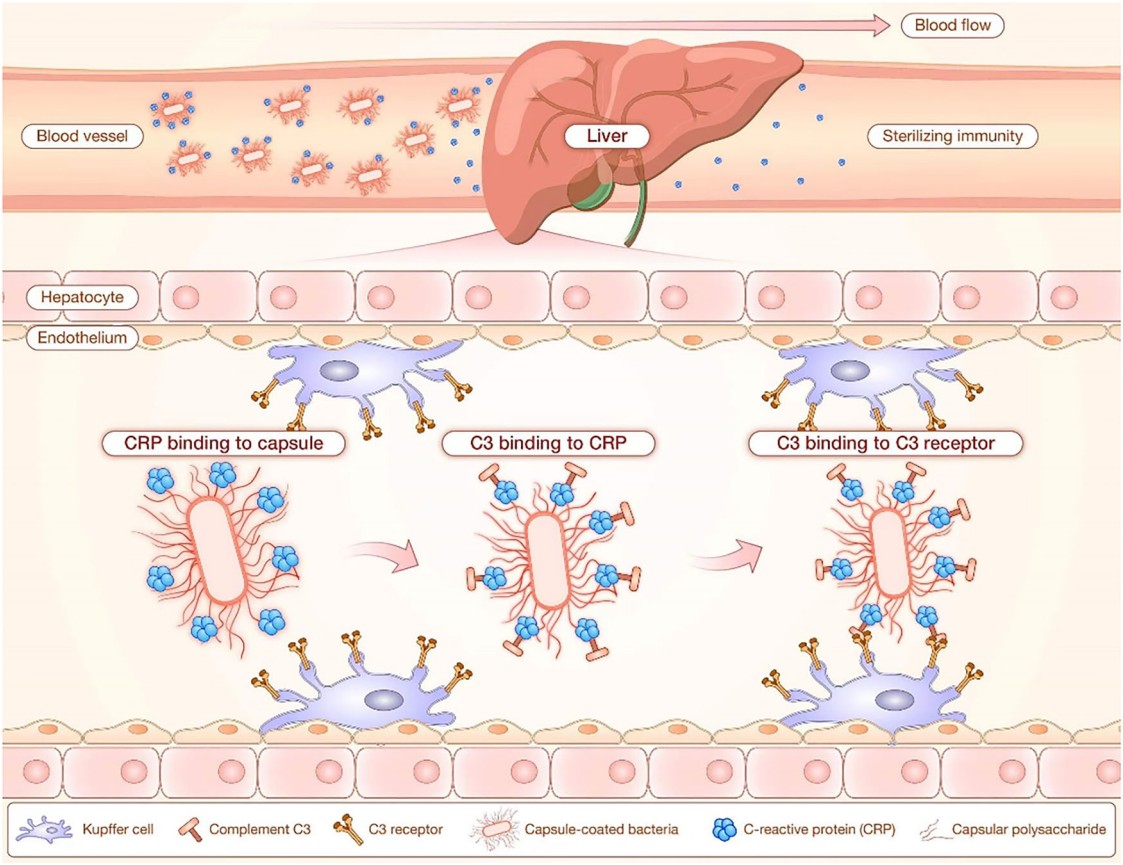

**Figure 8. Diagrammatic model for the CRP-based blood sterilizing immunity in the liver.**

In the blood vessels, CRP binds to the capsules of invading bacteria to achieve sterilizing immunity in the liver by the complement-dependent and independent mechanisms. The former is achieved by activating the binding interactions between complement protein C3 and its receptors (CR3 and CRIg) on Kupffer cells that are embedded in the endothelium of the liver sinusoids. Source data are available online for this figure.

## The perspectives of the CRP-mediated bacterial capture in understanding the molecular mechanisms of pathogen recognition and capture

The current literature almost exclusively focuses on human CRP, primarily due to the massive induction in response to infections and inflammation. As a result, passive administration or transgenic expression of human CRP has been used to study CRP function in mice (Ji et al, 2023; Singh et al, 2020). This study has demonstrated that the endogenous level of CRP in mice is sufficient to confer potent protection against septic infections by CRP-recognizable bacteria. Multiple lines of evidence strongly suggest that human CRP principally shares the anti-bacterial immunity. First, human CRP in normal human serum was identified as the specific receptor for the Sp23F capsule that is also recognized by mouse plasma CRP. Second, human and mouse CRPs commonly recognize many capsules. Lastly, the addition of recombinant hCRP in normal human serum significantly enhanced the adhesion of Sp23F to primary human KCs. In this context, massive increase in plasma CRP in humans during severe bacterial infections appears to represent an important host defense response to clear invasive bacteria from the bloodstream in the liver.

Our recent work has also shown that CRP plays a significant role in enabling neutrophils to clear CRP-sensitive bacteria in the lungs of mice (Fang et al, 2024). During the inflammatory stage of lung infection, CRP bound to the capsules of serotype 23F *S. pneumoniae* and type b *H. influenzae* activates the complement system, and shuffles the C3-opsonized bacteria to the CR3 and CRIg complement receptors on infiltrating neutrophils. This finding has demonstrated that CRP and other circulating capsule-binding receptors not only drive hepatic capture of blood-borne bacteria, but also promote bacterial clearance in the local infections.

Drug-resistant encapsulated bacteria are responsible for the majority of bacterium-associated human deaths globally (Naghavi and Collaborators, 2022). With the discovery of CRP and other capsule-binding receptors (An et al, 2022; Tian et al, 2025; Zeng et al, 2018), it is expected that new receptors for many other capsule types will be identified in the future. Based on their potency and broad spectrum in driving bacterial clearance, cocktails composed of multiple capsule-binding receptors may be an alternative for treating drug-resistant encapsulated bacteria. This possibility is supported by our success in enabling $Crp^{-/-}$ mice to clear serotype-23F from the bloodstream with recombinant CRP. Along this line, monoclonal antibodies have already been considered as the

therapeutic option for infections of drug-resistant bacteria (Simonis et al, 2023; Zurawski and McLendon, 2020).

## Limitations of the study

The molecular and cellular basis of anti-bacterial function of CRP is mostly characterized in mouse sepsis model. Because human and mouse CRPs showed a similar pattern of binding interactions with many capsule types, we expect that CRP serves an important immune function against invasive bacterial infections. This notion is supported by CRP-dependent binding of human primary KCs to mouse CRP-sensitive capsule types in vitro. However, human CRP clearly differs from the mouse counterpart in recognizing certain capsules, as manifested by the lack of binding interactions between hCRP and the CPSs of *H. influenzae* types a and b. In addition, it is known that human CRP binds to human C1q complement protein but not the mouse ortholog (Suresh et al, 2006). These lines of information indicate that CRPs of human and animals fulfill a common function in promoting the hepatic clearance of blood-borne pathogens, but the molecular details of their actions may vary among different hosts.

# Methods

### Reagents and tools table

| Reagent/Resource | Reference or Source | Identifier or Catalog Number |
|---|---|---|
| **Experimental models** | | |
| C57BL/6 *Crp* KO | Gempharmatech | Cat# T003305 |
| C57BL/6 *C1qa* KO | Gempharmatech | Cat# T015440 |
| C57BL/6 *C3* KO | Jackson Laboratory | Cat# 003641 |
| C57BL/6 *C4* KO | Congcong Zhang Laboratory | Cat# 003643 |
| C57BL/6 *Fcna* KO | Xulong Zhang Laboratory | N/A |
| C57BL/6 *Masp2* KO | Gempharmatech | Cat# T014510 |
| C57BL/6 CRIg (*Vsig4*) KO | Genentech | Cat# OM217373 |
| C57BL/6 CR3 KO | (Tian et al, 2025) | N/A |
| C57BL/6 CR3/CRIg DKO | (Tian et al, 2025) | N/A |
| C57BL/6 FcRα null | Dr. Jeffery V. Ravetch | N/A |
| Human liver sections | (An et al, 2022) | N/A |
| **Bacterial strains** | | All strains used in this work is listed in Table S5. |
| **Recombinant DNA** | | |
| pCMV-chikv-strepII | (Li et al, 2025) | Table S6 |
| pCMV-mCRP | This study | Table S6 |
| pCMV-hCRP | This study | Table S6 |
| pCMV-mCRP-F66A | This study | Table S6 |
| pCMV-mCRP-G76Y | This study | Table S6 |

| Reagent/Resource | Reference or Source | Identifier or Catalog Number |
|---|---|---|
| pCMV-mCRP-E81A | This study | Table S6 |
| pCMV-hCRP-F66A | This study | Table S6 |
| pCMV-hCRP-T76Y | This study | Table S6 |
| pCMV-hCRP-E81A | This study | Table S6 |
| PCMV-hCRP$^{m71-91}$ | This study | Table S6 |
| **Antibodies** | | |
| Strep-Tag mouse monoclonal IgG | Huaxingbio | Cat# HX1816 |
| HRP-conjugated goat anti-mouse IgG (H + L) | Huaxingbio | Cat# HX2032 |
| HRP-conjugated goat anti-mouse IgM (H + L) | Elabscience | Cat# E-AB-1008 |
| AF647-conjugated anti-mouse F4/80 IgG (clone: BM8) | BioLegend | Cat# 123121 |
| AF594-conjugated anti-mouse CD31 IG (clone: MEC13.3) | BioLegend | Cat# 102520 |
| rabbit anti-mCRP IgG | Invitrogen | Cat# PA5-81363 |
| HRP-conjugated Goat anti-mouse C3 IgG | Cappel | Cat# 55557 |
| AF647-conjugated goat anti-rabbit IgG | Invitrogen | Cat# A27040 |
| APC-Cy7 anti-mouse CD45 (Clone: 30-F11) | BioLegend | Cat# 103116 |
| APC anti-mouse CD31 (Clone: 390) | BioLegend | Cat# 102409 |
| FITC-conjugated anti-mouse F4/80 IgG (clone: BM8) | BioLegend | Cat# 123108 |
| Mouse anti-PC monoclonal IgG | (An et al, 2025) | N/A |
| Mouse anti-PC monoclonal IgM | (An et al, 2025) | N/A |
| PE anti-complement C3b/iC3b | BioLegend | Cat# 846104 |
| FITC-conjugated goat anti-mouse IgG | EASYBIO | Cat# BE0111 |
| **Oligonucleotides and other sequence-based reagents** | Synthesized by Synbio Tech | The primers are listed in Table S7. |
| **Chemicals, Enzymes and other reagents** | | |
| SMM 293-TII Expression Medium | Sino Biological | Cat# M293TII |
| Polyethylenimine (PEI) | Polysciences | Cat# 23966 |
| Strep-Tactin sepharose resin | IBA Lifesciences GmbH | Cat# 2-1201-010 |
| Desthiobiotin | IBA Lifesciences GmbH | Cat# 2-1000-002 |
| Cobra venom factor (CVF) | Quidel | Cat# A600 |
| TMB chromogenic substrate | Tiangen | Cat# PA107 |
| RPMI 1640 | Corning | Cat# 10-040-CV |

| Reagent/Resource | Reference or Source | Identifier or Catalog Number |
|---|---|---|
| Fetal bovine serum (FBS) | Biological Industries | Cat# 04-001-1ACS |
| Todd-Hewitt broth | Oxoid | Cat# CM0189 |
| Tryptic soy agar | BD Difco | Cat# 211047 |
| Tryptone | Oxoid | Cat# LP0042 |
| Yeast extract | Oxoid | Cat# LP0021 |
| Brain-heart infusion | BD Difco | Cat# 237500 |
| Hemin | Innochem | Cat# B95103 |
| Nicotinamide adenine dinucleotide (NAD) | Sigma-Aldrich | Cat# N7004 |
| Non-fat dry milk | BD Difco | Cat# 232100 |
| Fluorescein Isothiocyanate (FITC) | Sigma-Aldrich | Cat# F7250 |
| Collagenase IV | Sigma-Aldrich | Cat# C5138 |
| DNase I | Roche | Cat# 10104159001 |
| Hanks' Balanced Salt Solution (HBSS) | Corning | Cat# 21-022-CV |
| Phosphate-buffered saline (PBS) | FeiMoBio | Cat# FB14351 |
| RBC lysis buffer (10X) | BioLegend | Cat# 420301 |
| Tween 20 | Sigma-Aldrich | Cat# P9416 |
| Protein G Resin | GenScript | Cat# L00209 |
| Phosphocholine | Macklin | Cat# P912369 |
| β-Glycerophosphate disodium salt hydrate | Psaitong | Cat# G10340 |
| Maxima H Minus First Strand cDNA Synthesis Kit | Thermo Scientific | Cat# K1651 |
| Mem-PER Plus Membrane Protein Extraction Kit | Thermo Scientific | Cat# 89842 |
| MEGAshortscript Kit | Invitrogen | Cat# AM1354 |
| mMESSAGE mMACHINE T7 Ultra Kit | Invitrogen | Cat# AM1345 |
| ClonExpress Ultra One Step Cloning Kit | Vazyme | Cat# C115 |
| BCA Assay Kit | Solarbio | Cat# PC0020 |
| **Software** | | |
| GraphPad Prism v8.0.0 | GraphPad | https://www.graphpad.com |
| FlowJo v10.4 | Becton & Dickinson | https://www.flowjo.com |
| LAS X Core module v3.7.3 | Leica | https://www.leica-microsystems.com |
| Proteome Discovery v1.4 | Thermo Scientific | https://www.thermofisher.com |
| ClustalW v2.1 | Science Foundation Ireland | http://www.clustal.org |
| Microcal Origin v7.0 | OriginLab | https://microcal-origin.software.informer.com |
| MotionCor2 v1.2.4 | University of California San Francisco | https://emcore.ucsf.edu/ucsf-software |

| Reagent/Resource | Reference or Source | Identifier or Catalog Number |
|---|---|---|
| cryoSPARC v.4.1.0 | University of Toronto | http://www.cryosparc.com |
| AlphaFold3 | Alphabet | https://deepmind.google/science/alphafold |
| Origin software | MicroCal | https://www.originlab.com |
| **Others** | | |
| Carboxyl Latex Beads, 2 μm | Invitrogen | Cat# C37278 |
| 70 μm filter | Biologix | Cat# 15-1070 |
| 96-well plates | Jet Biofil | Cat# FEP-100-096 |
| 3 kDa Ultra-centrifugal Filter Unit | Merck Millipore | Cat# UFC800308 |
| 4-Chamber 35 mm glass bottom dish with 20 mm microwell | Cellvis | Cat# D35C4-20-1.5-N |
| Copper grids (R1.2/1.3 300 mesh) | Quantifoils | Cat# N-C14nCu30-01 |

## Methods and protocols

### Human subjects

Human Kupffer cells were isolated from liver sections that were freshly disposed after liver surgery or transplantation as previously described (An et al, 2022), with the approval by the Human Subject Study Committee of Tsinghua University (No. THU01-20240036).

## Bacterial strains and cultivation

All of the *S. pneumoniae*, *H. influenzae* and *K. pneumoniae* strains used in this study are described in Appendix Table S5. Pneumococci were cultured in Todd-Hewitt broth with 0.5% yeast extract (THY) or tryptic soy agar (TSA) plates with 3% defibrinated sheep blood at 37 °C and 5% $CO_2$ as described (Lu et al, 2006). *H. influenzae* strains were propagated under the same conditions in brain-heart infusion (BHI) broth or on BHI agar supplemented with 10 μg/ml hemin and 10 μg/ml NAD as described (Weiser et al, 1989). *K. pneumoniae* strains were grown in Luria-Bertani (LB) broth or on LB agar plates as described (Huang et al, 2022).

## Capsular polysaccharide (CPS) purification

CPSs were purified from broth cultures of *S. pneumoniae*, *H. influenzae* and *K. pneumoniae* strains and quantified as described (An et al, 2022; Huang et al, 2022).

## Isolation of liver non-parenchymal cells (NPCs)

Human and mouse liver non-parenchymal cells (NPCs) were isolated by a collagenase-DNase digestion procedure as described previously (An et al, 2022). The concentrations of membrane proteins were quantified with the BCA Assay kit.

## Screening for capsule-binding proteins

Mouse and human capsule-binding proteins were screened by an affinity pulldown approach as described (An et al, 2023). In brief, membrane proteins of mouse liver NPCs were enriched by the Mem-PER Plus Membrane Protein Extraction Kit according to the manufacturer's instructions. Capsule binding-proteins were enriched by co-incubating CPS23F-coated latex beads with mouse liver NPC membrane proteins in the presence of normal mouse serum 10% (v/v) at room temperature (RT) for 1 h. Beads with CPS8 from the HV serotype 8 was similarly processed as a negative control. Proteins bound to the CPS-coated beads were identified by mass spectrometry. Each sample containing 40 μg of total protein was separated by SDS-PAGE. The corresponding protein bands were excised and subjected to in-gel digestion. The resulting peptides were analyzed by liquid chromatography-tandem mass spectrometry (LC-MS/MS) using a Thermo-Dionex Ultimate 3000 HPLC system coupled with a Thermo Orbitrap Fusion mass spectrometer. For protein identification, the spectra from each LC-MS/MS run were processed using the Proteome Discoverer software (version 1.4) and searched against the mouse or human Unreviewed TrEMBL FASTA database (release 2020_05). Proteins identified by at least two unique peptides were included for quantification. Protein abundance was determined as the median intensity of all peptide spectra corresponding to the same protein. Protein abundance was compared between CPS23F- and CPS8-coated beads to identify CPS23-enriched proteins. Proteins with at least 2-fold enrichment by CPS23F were considered as CPS23-binding candidates. CPS23-binding proteins in human serum were identified in a similar manner by co-incubating CPS23- and CP8-coated beads with 10% normal human serum. The raw and processed proteomic data have been deposited in the iProX database under accession number IPX0012537001.

## Detection and depletion of C-polysaccharide/phosphocholine

To assess phosphorylcholine (PC) level on bacterial surface, $10^6$ CFU of bacteria were incubated with anti-PC IgG antibody (1:100) for 20 min at RT. After three washes with PBS, bacteria were further incubated with FITC-conjugated goat anti-mouse IgG antibody (1:2000) for 20 min at RT. FITC signals were detected using a flow cytometer (Bigfoot Spectral Cell Sorter, Thermo Fisher Scientific, USA).

The C-polysaccharide contaminants in CPS preparations were removed with the anti-PC IgG monoclonal antibody (An et al, 2025) as described (Paliwal et al, 2024). Free CPSs in PBS (100 μl, 10 μg/ml) were incubated with the anti-PC IgG at RT for 1 h, followed by adding 100 μl Protein G Resin, incubating at 4 °C for 1 h, and centrifugating at $300 \times g$ for 10 min. The supernatants (100 μl) were used to assess the effect of PC depletion or CRP binding by coating single wells of 96-well plates by ELISA. PC was detected with the anti-PC IgM monoclonal antibody (1:3000) (An et al, 2025) and HRP-conjugate goat IgG against mouse IgM (1:1000). PC-conjugated bovine serum albumin (BSA) was prepared as described (An et al, 2025), and used as a positive control. CRP binding to PC-depleted CPSs was tested in a similar manner as described in the ELISA method.

## Production of recombinant CRP

Mouse and human CRPs were expressed as Strep-tagged recombinant proteins in human HEK293F suspension culture cells as described (Cao et al, 2023). Briefly, the complementary DNAs (cDNAs) were generated with the total RNAs of mouse liver to amplify the coding sequence of mCRP with primers Pr19504 and Pr19505. The amplicon was cloned in the SalI/BamHI site of pCMV-chikv-strepII vector with a Strep-tag (WSHPQFEK) (Li et al, 2025) to generate pCMV-mCRP (pTH17157). The coding sequence of hCRP was chemically synthesized according to accession NM_000567.3, and cloned in the PstI/BamHI site of pCMV-chikv-strepII vector to generate pCMV-hCRP (pTH17158). The insert sequences were confirmed by DNA sequencing. Recombinant human CRP was expressed by transient transfection of HEK293F cells, isolated from culture supernatants by affinity chromatography with Strep-Tactin sepharose resin and eluted with 2.5 mM desthiobiotin in $Ca^{2+}$-supplemented Tris-buffered saline (TBS-$Ca^{2+}$) (20 mM Tris, 150 mM NaCl, 5 mM CaCl$_2$, pH 8.0). Except for a high concentration of NaCl (500 mM), the same conditions were used to purify r-mCRP. The fractions containing target proteins were pooled and proceeded for desthiobiotin removal using 3 kDa molecular weight cutoff (MWCO) ultrafiltration units. Protein concentration was quantified using the BCA Assay Kit.

pCMV-mCRP was used as template for the construction of F66A (TH17199), E81A (TH17198), and G76Y (TH17203) mutant mouse CRP. The upstream and downstream fragments of mutant mouse CRP were amplified and were fused to generate mutant mouse CRP DNA, which was then subcloned into SalI/BamHI sites of pCMV-chikv-strepII vector. The F66A (TH17201), E81A (TH17200), and T76Y (TH17202) mutant human CRP was similarly constructed. The recombinant mutant fragment was subcloned into PstI/BamHI sites of pCMV-chikv-strepII vector. Mutant plasmids of pCMV-hCRP[m71-91] (TH17388) were constructed by site-directed mutagenesis as described (Hemsley et al, 1989). Briefly, pCMV-hCRP served as the template, and the PCR products were circularized using ClonExpress Ultra One Step Cloning Kit. These mutant CRP were expressed and purified as described above. The strains and primers used in this study were, respectively, listed in Tables S6 and S7.

## Enzyme-linked immunosorbent assay (ELISA)

CRP binding to capsular polysaccharides (CPS) was assessed by ELISA principally as described (Agrawal et al, 2002). Briefly, 96-well plates were coated with 100 μl PBS containing 10 μg/ml free CPSs overnight at 4 °C, and blocked with 100 μl 5% non-fat milk in TBS-$Ca^{2+}$ containing 0.05% Tween-20 (TBST) for 1 h at RT. The CPS-coated wells were sequentially incubated at RT with 100 μl TBST buffer containing Strep-tagged recombinant CRP (5 μg/ml), Strep-Tag mouse monoclonal antibody to Strep-Tag (1:2000) or horseradish peroxidase (HRP)-conjugated goat anti-mouse IgG (1:2000). Each reaction was separated by a washing step (3 times with 200 μl TBST buffer). CRP-CPS binding was quantified by adding 100 μl TMB chromogenic substrate and measuring optical density (OD) at 450 nm. A coating concentration of 10 μg/ml CPS was chosen for all capsule types tested in this work, because it represented an overly saturating condition on the basis of our

preliminary experiment using CPS preparations purified from *S. pneumoniae* serotypes 5, 6A and 8 and mouse immune serum for the 13-valent pneumococcal conjugate vaccine (PCV13) generated in our previous study (Wang et al, 2023). In the trial, various concentrations of purified CPSs (0.01–10 μg/ml) were added to the wells of the ELISA plates before probing with the PCV13 immune serum. CPS of serotype 8 was used as negative control since PCV13 contained the CPSs of serotypes 5 and 6A but not 8. The reactivity of serotype-5 and -6A CPSs to the immune serum reached the plateau when the antigens were coated at a concentration of 1 μg/ml, with marginal signal to the serotype-8 CPS-coated wells (Fig. EV1A).

Competitive inhibition of free CPS23F, PC and PG (competitor) against the CRP-CPS binding interaction was carried out by adding the competitors and r-mCRP to CPS-coated wells simultaneously at the first reaction of ELISA. Titration curves were generated using sigmoid dose response of nonlinear fit in GraphPad to determine the effective median concentration of inhibition ($EC_{50}$).

The serum CRP level of mouse was similarly determined by ELISA. Serum samples were collected from mice by retro-orbital bleeding at various time points post i.v. infection with $10^6$ CFU of *Sp*23F. Serially diluted sera were incubated with CPS23F-coated plates at RT for 2 h. CRP was detected by a rabbit anti-mouse CRP antibody (1:2000) and quantified by an HRP-conjugated goat anti-rabbit IgG (1:2000).

## Detection of in vitro CRP binding to encapsulated bacteria

CRP binding to live bacteria was detected by immunofluorescent microscopy essentially as described (Lu et al, 2006). Briefly, bacteria ($10^5$ CFU) were sequentially treated in 100 μl each of 5% non-fat milk (w/v), 10 μg/ml r-mCRP, rabbit anti-mCRP antibody (1:200), and AF647-conjugated goat anti-rabbit IgG (1:200). Each incubation step was followed by centrifugation and resuspension in 100 μl PBS. The bacteria were resuspended in 20 μl PBS and visualized under a Leica TCS SP8 confocal microscope.

## Detection of in vitro C3 activation by the CRP-capsule interaction

The C3 activation by the mouse CRP-capsule interactions was determined by ELISA. Serum samples from WT or CRP-deficient mice were diluted to 10% (v/v) in PBS supplemented with 2 mM $CaCl_2$ and 2 mM $MgCl_2$ and incubated in CPS8- or CPS23F-coated wells for various times at 37 °C. C3 captured onto the immobilized CRPs was detected by HRP-conjugated goat anti-mouse C3 IgG (1:5000).

The C3 activation by the human CRP-capsule interactions was characterized by flow cytometry as described (Cunnion et al, 2004). Briefly, $10^6$ CFUs of *S. pneumoniae* were incubated with normal human serum, heat-inactivated NHS, or PBS at 37 °C for 30 min. Following three washes with PBS, bacteria were stained with PE-conjugated anti-human C3 IgG (1:40) for 20 min at RT. PE signals were measured by flow cytometry.

## Mouse infection

All infection experiments were conducted in C57BL/6 (6–8 weeks old) according to the animal protocols approved by the

Institutional Animal Care and Use Committee of Tsinghua University (Protocol No. 22-ZJR1). All of gene-deficient mice were maintained in the C57BL/6 background. $Crp^{-/-}$ mice, $C1qa^{-/-}$, and $Masp2^{-/-}$ mice were acquired from Gempharmatech (Nanjing, China). $C3^{-/-}$ mice were purchased from the Jackson Laboratory (Bar Harbor, Maine, USA). $C4^{-/-}$ mice were obtained from Congcong Zhang laboratory at Beijing Anzhen Hospital, Capital Medical University. $Fcna^{-/-}$ mice were obtained from Xulong Li laboratory at Capital Medical University. $CRIg^{-/-}$ mice (Helmy et al, 2006) were obtained from Genentech (CA, USA). Complement receptor 3 (CR3)-deficient mice were generated with CRISPR/Cas9 system as described (Wang et al, 2023). $CR3/CRIg^{-/-}$ mice were generated by crossing $CRIg^{-/-}$ mice with $CR3^{-/-}$ mice. FcRα null mice were generously provided by Jeffery Ravetch (Smith et al, 2012). All mice were kept under specific pathogen-free (SPF) conditions with free access to food and water.

Septic infections were carried out and analyzed as described (An et al, 2022). Briefly, bacteria in 100 μl of Ringer's solution were i.v. injected into tail vein. Bacteria in the blood were assessed by retro-orbital bleeding and CFU counting on blood agar plates. Bacterial 50% clearance time ($CT_{50}$) was calculated by nonlinear regression analysis of bacteremia kinetics using the formula $T = \ln\{(1-50/\text{Plateau})/(-K)\}$. Organ bacteria were similarly quantified with homogenized tissues. Total viable bacteria in each mouse were estimated as the sum of CFU values from blood and organ samples. Competitive inhibition of bacterial clearance was performed by i.v. administration of CPS 2–5 min prior to bacterial inoculation. For passive protection, bacteria were incubated with r-mCRP (10 μg/ml) in 100 μl phosphate-buffered saline (PBS) supplemented with 2 mM $CaCl_2$ at RT for 5 min before i.v. inoculation. Survival rate was determined by monitoring infected mice for 7 days post infection. $LD_{50}$, lethal dose 50%, was obtained on the basis of infection dose and corresponding survival rate of mice using the $LD_{50}$ calculator (Quest GraphTM, AAT Bioquest, Inc.).

## Intravital microscopy (IVM)

IVM imaging of mouse liver sinusoids was conducted as described (An et al, 2022). LSECs and KCs were labeled by i.v. injection of AF594 anti-CD31and AF647 anti-F4/80 antibodies, respectively before i.v. administration of FITC-labeled bacteria. Images of the liver vasculatures were acquired with Leica TCS-SP8 confocal microscope using 10×/0.45 NA and 20×/0.80 NA HC PL APO objectives 10–15 min post infection. Photomultiplier tubes (PMTs) and hybrid photo detectors (HyD) were used to detect fluorescence signals ($600 \times 600$ pixels for time-lapse series and $1024 \times 1024$ pixels for photographs). At least 5–10 random fields of view (FOV) were captured to calculate bacterial number per FOV. Leica Biosystems software was used for image and movie processing (LAS X Life Science).

## Bacterial binding to primary Kupffer cells

Bacterial binding to primary KCs in vitro was evaluated as described (An et al, 2022). In brief, KCs from human and mouse liver NPCs were isolated as abovementioned, and immobilized to the bottom of 48-well plates for 30 min at 37 °C before being infected with $1 \times 10^5$ CFU of bacteria (MOI = 1) in the presence or absence of 10% serum (v/v) for 30 min at 37 °C with 5% $CO_2$. The

CFUs in culture supernatants and KC lysates were enumerated as free and KC-bound bacteria. The in vitro visualization of bacterial binding to KCs was carried out by immunofluorescence microscopy as recently documented (Tian et al, 2025). Mouse liver NPCs were seeded in 4-chamber plates with 35 mm glass slip. After non-adherent cells were removed, KCs were stained with AF647 anti-F4/80 and infected with FITC-labeled bacteria (MOI = 10) in the presence of 20% mouse serum for 30 min at 37 °C with 5% $CO_2$. KCs and bacteria were visualized using the Leica TCS SP8 confocal microscope. Photographs with more than 100 bacteria were chosen for quantification.

## Isothermal titration calorimetry (ITC)

The binding characteristics of CRP to CPSs were measured at 25 °C using a VP-ITC Microcalorimeter (Malvern Instruments Ltd) as described (Lu et al, 2006). In principle, the titrant (CPS) and titrated molecules (CRP) were prepared in the TBS-$Ca^{2+}$ buffer. Each CPS was diluted to a 50-fold final concentration of recombinant CRP, except hCRP-27 (1:35) and hCRP-33A (1:200). CPS solutions were injected in the CRP-containing calorimeter cell with a titration volume of 10 μl and a spacing time of 210 s. Calorimetric data were recorded in real-time, and analyzed to acquire the stoichiometry (N, the molar ratio of binding), the association equilibrium constant ($Ka$), binding entropy ($\Delta S$), and binding enthalpy ($\Delta H$) with Origin software (Version 7, MicroCal). The binding affinity is presented as dissociation equilibrium constant ($Kd$) ($1/Ka$).

## Cryo-EM sample preparation and data collection

Human CRP was concentrated to 6 mg/ml, and incubated with CPS23F to a final concentration of 4.5 mM on ice for ~1 h. Tween-20 was added to the solution to a final concentration of 0.02% (w/v). Droplets (3.5 μl) of the complex were applied to glow-discharged carbon-coated copper grids (Quantifoils R1.2/1.3 300 mesh), which were blotted for 2 s before being plunge frozen in liquid ethane using Vitrobot Mark IV (Thermo Fisher Scientific). Gain normalized movies were collected on a Titan Krios microscope (ThermoFisher) operated at 300 keV and equipped with a Gatan K3 detector. Movies were acquired automatically using EPU software using a total dose of 50 e⁻/Å² with 32 frames, at ×105,000 magnification with a calibrated pixel size of 0.85 Å and a defocus range of −1.0 to −2.0 μm.

The movie frames were motion-corrected using MotionCor2 (v1.2.4) (Zheng et al, 2017), and the contrast transfer function (CTF) values of each micrograph were calculated using patch CTF estimation implemented in cryoSPARC (v.4.1.0) (Punjani et al, 2017). All imaging processes were performed using cryoSPARC unless mentioned elsewhere. The map was finally sharpened by DeepEMhancer (Sanchez-Garcia et al, 2021). We first picked ~600,000 particles from 1000 micrographs, and these particles were subjected to perform 2D-classification and generated a particles-dataset for Topaz training (Bepler et al, 2019). To strengthen the Topaz procedure's effect in particle picking, we retrained Topaz model using newly 2D-classification-filtering particles or heterogeneous refinement-filtering particles. Then, we applied well-trained Topaz model to pick up particles from the entire dataset,

and 854,515 particles were extracted and subjected to 2D-classification. After 2D-classification, a clean dataset containing 506,697 particles was selected to perform initial reconstruction and heterogeneous refinement. 207,101 particles were separated out, and a density map at 2.78 Å resolution estimated by the gold-standard FSC 0.143 criterion was obtained after CTF refinement and non-uniform refinement. The image-processing workflow is summarized in Appendix Table S4.

To build the pentameric model for hCRP and CPS23F complex, a monomeric subunit from a crystal structure of human CRP (PDB code 1B09) (Thompson et al, 1999) was placed into the cryoEM map as a rigid body using UCSF ChimeraX (Pettersen et al, 2021). The model was then manually adjusted using Coot (Emsley and Cowtan, 2004). The model of this monomer was duplicated into the remaining 4 sites as rigid bodies using ChimeraX. The resulting pentameric model was automatically refined using Phenix real-space refine with Ramachandran restraints, secondary structural restraints, and geometry restraints (Adams et al, 2010). The output from Phenix was manually inspected and refined for several rounds aided by stereochemical quality assessment using MolProbity (Chen et al, 2010). Besides, the structure of mouse CRP was predicted by AlphaFold (Jumper et al, 2021). Structural figures were prepared using PyMOL (www.pymol.org/) and UCSF ChimeraX.

## Sequence analysis

DNA and amino acid sequences were analyzed using the DNASTAR Lasergene version 15.0 for Macintosh (Madison, WI).

## Quantification and statistical analysis

Statistical analyses were performed using GraphPad Prism software version 8.0.0, and all data were expressed as Mean ± SEM unless otherwise stated. One- and two-way ANOVA multiple comparisons test was used to analyze data between multiple groups. Survival curves were analyzed by log-rank (Mantel-Cox) test. $P$ value < 0.05 was considered as significant (not significant, ns; *$P$ < 0.05; **$P$ < 0.01; ***$P$ < 0.001; ****$P$ < 0.0001).

# Data availability

The cryo-EM density map and atomic coordinate have been deposited to the Electron Microscopy Data Bank (EMDB ID: EMD-63844) and the Protein Data Bank (PDB ID: 9U4E), respectively. The mass spectrometry proteomics data have been deposited to the iProX database under the accession number IPX0012537001. The permanent URLs to access these data are as followed: Cryo-EM map: https://www.ebi.ac.uk/emdb/EMD-63844; Atomic coordinates: https://www.rcsb.org/structure/9U4E; Proteomics data: https://www.iprox.cn//page/subproject.html?id=IPX0012537001.
All data and materials used in this study will be made available under the conditions of material transfer agreements.

The source data of this paper are collected in the following database record: biostudies:S-SCDT-10_1038-S44318-025-00623-w.

# Peer review information

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

## Acknowledgements

We would be grateful to Yuanyuan Chen, Zhenwei Yang and Bingxue Zhou (Institute of Biophysics, Chinese Academy of Sciences) for technical help with ITC experiments. We thank the Tsinghua research platforms for assistance in animal experimentation (Laboratory Animal Research Center), flow cytometry (Medical Microbiology Core Facility), IVM imaging (Center for Cell Biology) and protein mass spectrometry (Center for Proteomics). This work was supported by the following grants: National Key Research and Development Program of China grants 2023YFC2306300 (J-RZ) and 2023YFC2308003 (HA), National Natural Science Foundation of China 82330071 (J-RZ) and 32200143 (YL), Tsinghua University Initiative Scientific Research Program 20243080033 (J-RZ), and Conducting Surveys, Research and Risk Assessment on Zoonotic Diseases 0733-24095348 (J-RZ).

## Author contributions

**Danyu Chen**: Conceptualization; Investigation; Visualization. **Jiao Hu**: Conceptualization; Investigation; Visualization. **Mengran Zhu**: Investigation. **Yufeng Xie**: Investigation. **Hantian Yao**: Investigation. **Haoran An**: Conceptualization; Funding acquisition; Visualization. **Yumin Meng**: Investigation. **Juanjuan Wang**: Investigation. **Xueting Huang**: Investigation. **Yanni Liu**: Funding acquisition; Investigation. **Zhujun Shao**: Methodology. **Ye Xiang**: Methodology; Writing—original draft. **Jianxun Qi**: Supervision; Funding acquisition; Writing—original draft; Project administration; Writing—review and editing. **George Fu Gao**: Funding acquisition; Writing—original draft; Project administration; Writing—review and editing. **Jing-Ren Zhang**: Conceptualization; Supervision; Funding acquisition; Writing—original draft; Project administration; Writing—review and editing.

Source data underlying figure panels in this paper may have individual authorship assigned. Where available, figure panel/source data authorship is listed in the following database record: biostudies:S-SCDT-10_1038-S44318-025-00623-w.

## Disclosure and competing interests statement

The authors declare that a patent application (application number 2024113454400) related to the use of CRP, based on the findings reported in this study, has been filed. This may constitute a potential competing interest. George F. Gao is a member of the Advisory Editorial Board of *The EMBO Journal*. This has no bearing on the editorial consideration of this article for publication.

# Expanded View Figures

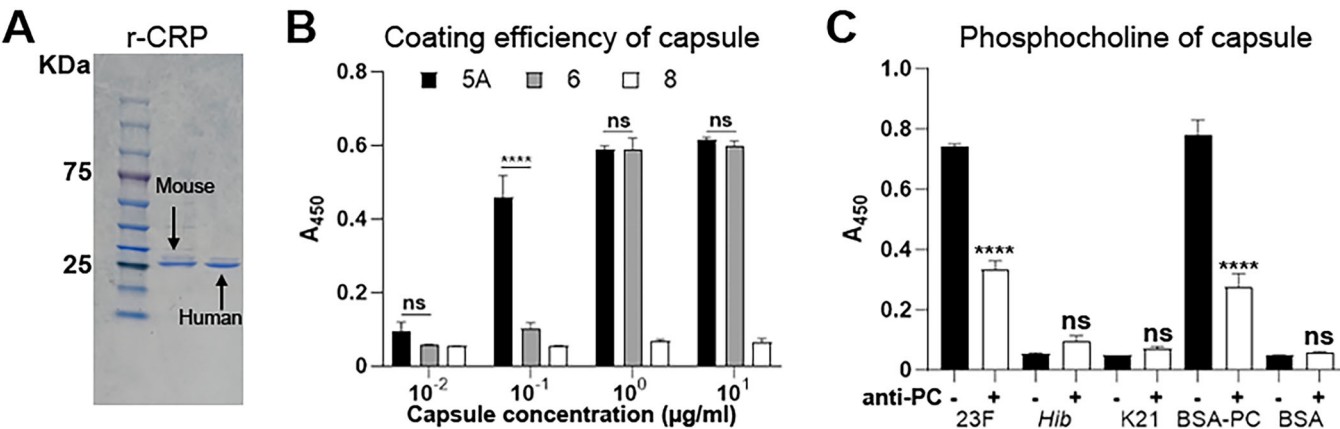

**Figure EV1. Characterization of capsule coating efficiency and phosphocholine levels by ELISA.**

(A) Construction of r-mCRP and r-hCRP. Purified r-mCRP and r-hCRP were detected by SDS-PAGE electrophoresis. $n = 2$. (B) Validation of capsule coating efficiency for ELISA. CPSs were coated onto plates at increasing concentrations (0.01, 0.1, 1, and 10 µg/ml, 100 µl/well) and detected using the serum from PCV13-immunized mice diluted at 1:300. Antibody binding was quantified by $OD_{450}$ measurements. $A_{450\ nm}$ were presented as mean ± SEM of three biological replicates ($n = 3$). $P$ value was calculated by Two-way ANOVA with Sidak's multiple comparisons (5 A vs 6) (****$P < 0.0001$, ns $P > 0.05$, no significance). $P$ value: $10^{-2}$, 0.5298; $10^{-1}$, 1.14e−010; $10^{0}$, 0.9999; $10^{1}$, 0.9291. (C) Detection of phosphorylcholine (PC) in capsule by ELISA. The black bars represent the level of PC detected in the capsule, while the white bars show the PC levels after treatment with anti-PC-IgG antibody. The capsules of *S. pneumoniae* serotype 23F, *H. influenzae* type b (*Hib*), and *K. pneumoniae* serotype K21, along with BSA-conjugated PC (BSA-PC) and BSA, were coated onto 96-well plates. PC were detected using a monoclonal anti-PC IgM antibody (1:3000), followed by an HRP-conjugated anti-mouse IgM secondary antibody (1:2000). Absorbance was measured at 450 nm ($OD_{450}$). $A_{450\ nm}$ were presented as mean ± SEM of three biological replicates ($n = 3$). $P$ value was calculated by Two-way ANOVA with Sidak's multiple comparisons (anti-PC+ vs anti-PC-) (****$P < 0.0001$, ns $P > 0.05$, no significance). $P$ value: 23F, 1.12e−009; *Hib*, 0.8068; K21, 0.9834; BSA-PC, 2.37e−011; BSA, 0.9997. Source data are available online for this figure.

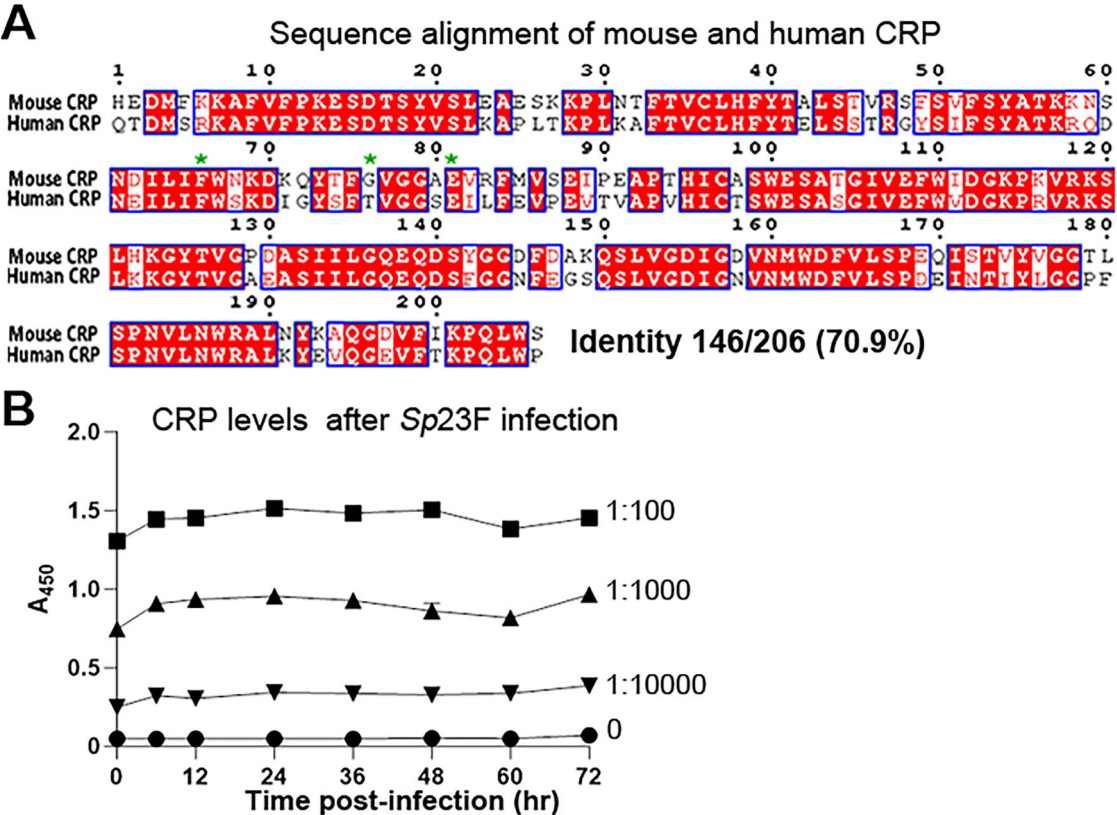

**Figure EV2. Comparative analysis of mouse and human CRP.**

(A) Sequence comparison between hCRP and mCRP. The amino acid sequences (without signal peptide) from mouse and human CRP were aligned by ClustalW software, and then the alignment results were imported into ENDscript/ESPript website to create the alignment figure. The conserved amino acids were indicted with red highlights. (B) ELISA measurement of serum CRP during the course of pneumococcal blood infection. Endogenous CRP levels in mouse serum were measured at the indicated time points post-*Sp*23F infection (0, 1, 6, 12, 24, 36, 48, 60, and 72 h). Serum samples were diluted at different concentrations (0, 1:100, 1:1000, 1:10,000) and analyzed using CRP-specific antibodies. $n = 3$. Source data are available online for this figure.

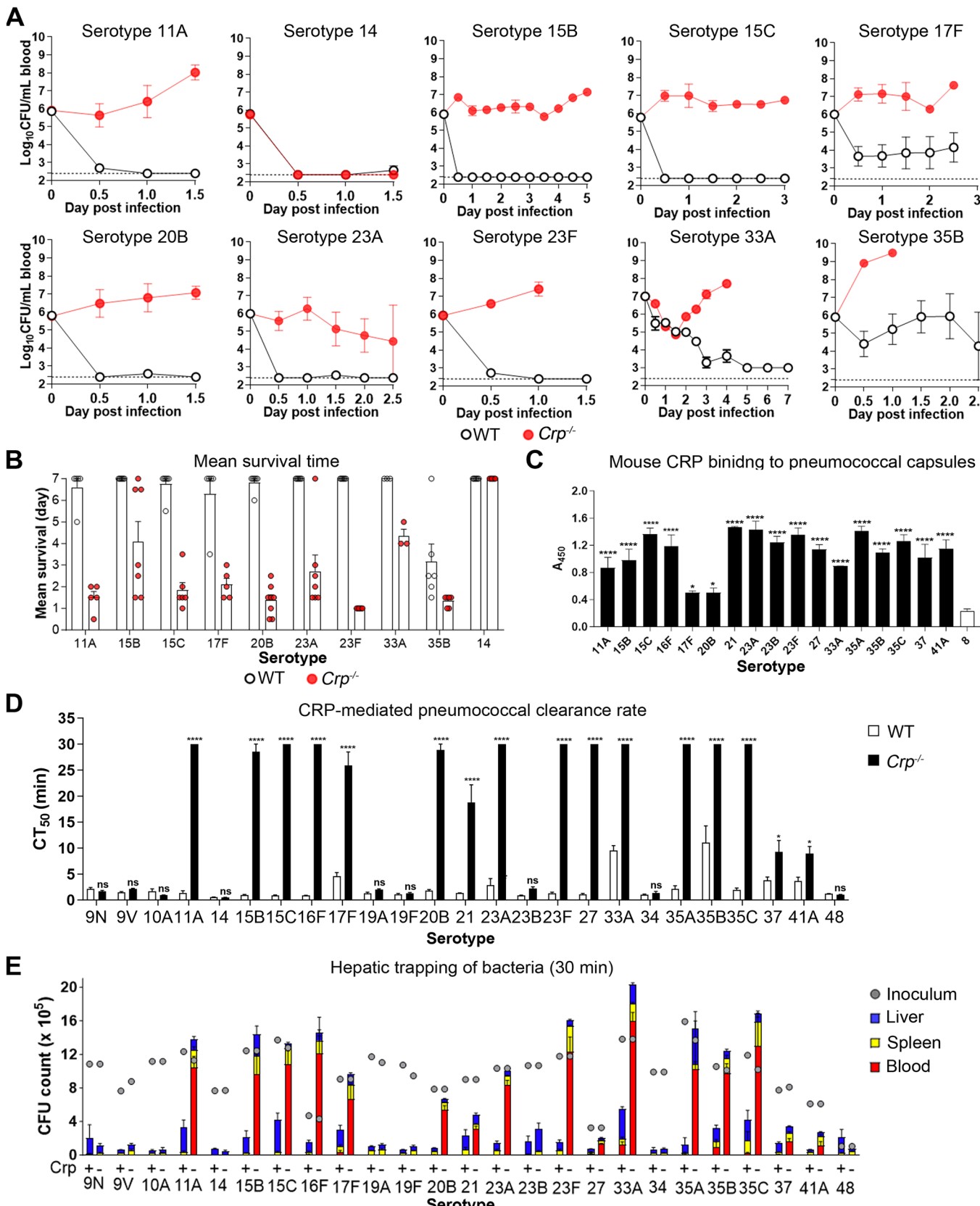

◄ **Figure EV3.  The requirement of CRP for broad serotype-specific shuffling of many pneumococcal serotypes from the blood circulation to the liver.**

(A) The bacteremia kinetics of WT and $Crp^{-/-}$ mice i.v. infected with 10 different serotypes. The blood CFU in individual mice were presented as mean ± SEM of three to eight biological replicates ($n$ = 3–8). (B) Broad protection of CRP against septic infection of CRP-sensitive serotypes. WT and $Crp^{-/-}$ mice were i.v. infected with $10^6$ CFU of $Sp$23F, $Sp$14, $Sp$11A, $Sp$15B, $Sp$15C, $Sp$17F, $Sp$20B, $Sp$23A, and $Sp$35B. Additionally, mice were i.v. infected with $10^7$ CFU of $Sp$33A. The survival of WT and $Crp^{-/-}$ mice were assessed. Survival time were presented as mean ± SEM of three to eight biological replicates ($n$ = 3–8). (C) mCRP binding to multiple pneumococcal capsules. The 96-well plates were individually pre-coated with purified pneumococcal capsules, and then incubated with 5 μg/ml r-mCRP. The CPS-bound CRP was then detected and represented by the absorbance at $A_{450\ nm}$ as in Fig. 1D. $A_{450\ nm}$ were presented as mean ± SEM of three to six biological replicates ($n$ = 3–6). $P$ value was calculated by One-way ANOVA with Sidak's multiple comparisons vs $S.\ pneumoniae$ 8 (****$P$ < 0.0001, *$P$ < 0.05). $P$ value: 11A, 3.36e−009; 15B, 3.27e−011; 15C, 2.06e−017; 16F, 3.75e−016; 17F, 0.0350; 20B, 0.04167; 21, 1.71e−017; 23A, 2.90e−018; 23F, 1.01e−018; 27, 7.66e−013; 33A, 1.10e−008; 35A, 1.01e−016; 35B, 3.41e−013; 35C, 1.04e−014; 37, 8.11e−011; 41 A, 5.34e−013. (D) The clearance rates of 25 low-virulence pneumococcal serotypes from the bloodstream in WT and $Crp^{-/-}$ mice infected i.v. with $10^6$ CFU of each serotype. $CT_{50}$ showed that $Crp^{-/-}$ mice significant delayed in clearing 16 of the 25 serotypes. $CT_{50}$ were presented as mean ± SEM of three to five biological replicates ($n$ = 3–5). $P$ value was calculated by Two-way ANOVA with Sidak's multiple comparisons (WT vs $Crp^{-/-}$) (****$P$ < 0.0001, ns $P$ > 0.05, no significance). $P$ value: 9N, 0.9999; 9V, 0.9999;10A, 0.9999; 11A, 1.98e−059; 14, 0.9999; 15B, 2.23e−050; 15C, 1.27e−055; 16F, 2.42e−040; 17F, 9.86e−038; 19A, 0.9999; 19F, 0.9999; 20B, 9.02e−044; 21, 2.52e−020; 23A, 1.65e−049; 23B, 0.9996; 23F, 1.75e−052; 27, 5.79e−040; 33 A, 4.01e−028; 34, 0.9999; 35A, 2.91e−038; 35B, 1.39e−032; 35C, 1.45e−038; 37, 0.0142; 41A, 0.02155; 48, 0.9999. (E) Serotype-specific distribution of viable bacteria in the blood, liver and spleen of WT and $Crp^{-/-}$ mice at 30 min post i.v. infection. The mice used to determine bacteremia kinetics in (A) were sacrificed at 30 min to quantify viable bacteria in the blood, liver and spleen by CFU plating. The inoculum of each group is indicated with a filled circle. Organ CFU in individual mice were presented as mean ± SEM of three to seven biological replicates ($n$ = 3–7). Source data are available online for this figure.

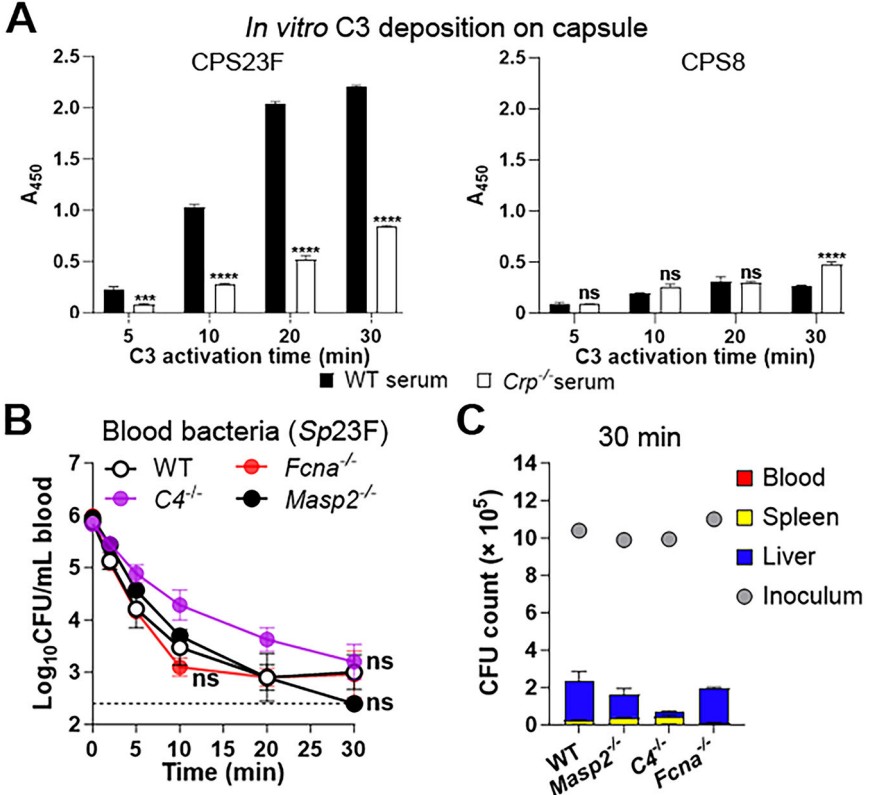

**Figure EV4. Evaluation of CRP-activated C3 activation.**

(A) CRP-activated C3 deposition on free capsular polysaccharide of serotype-23F *S. pneumoniae* was detected by ELISA. The 96-well plates were pre-coated with 10 µg/ml CPS23F or CPS8, and incubated with 100 µl of 10% serum at 37 °C for various durations. The abundance of C3 bound to capsular polysaccharides was detected with anti-C3 antibody. $A_{450\ nm}$ were presented as mean ± SEM of three to four biological replicates ($n = 4$). *P* value was calculated by Two-way ANOVA with Sidak's multiple comparisons (WT serum vs $Crp^{-/-}$ serum) (****$P < 0.0001$, ***$P < 0.001$, ns $P > 0.05$, no significance). *P* value of CPS23F: 5 min, 0.0009; 10 min, 1.95e−017; 20 min, 1.43e−024; 30 min, 2.00e−023. *P* value of CPS8: 5 min, 0.9985; 10 min, 0.1811; 20 min, 0.9988; 30 min, 6.95e−006. (B, C) The role of the C3 pathway in *Sp*23F infection. WT, $Fcna^{-/-}$, $MASP2^{-/-}$, and $C4^{-/-}$ mice were infected i.v. with *Sp*23F of $10^6$ CFU. The bacterial burden in the blood and major organs at 30 min was measured. The blood and organ CFU in individual mice were presented as mean ± SEM of three to six biological replicates ($n = 3$–6). *P* value was calculated by One-way ANOVA with Sidak's multiple comparisons vs WT (ns $P > 0.05$, no significance). *P* value: $C4^{-/-}$, 0.9015; $Masp2^{-/-}$, 0.9997; $Fcna^{-/-}$, 0.9996. Source data are available online for this figure.

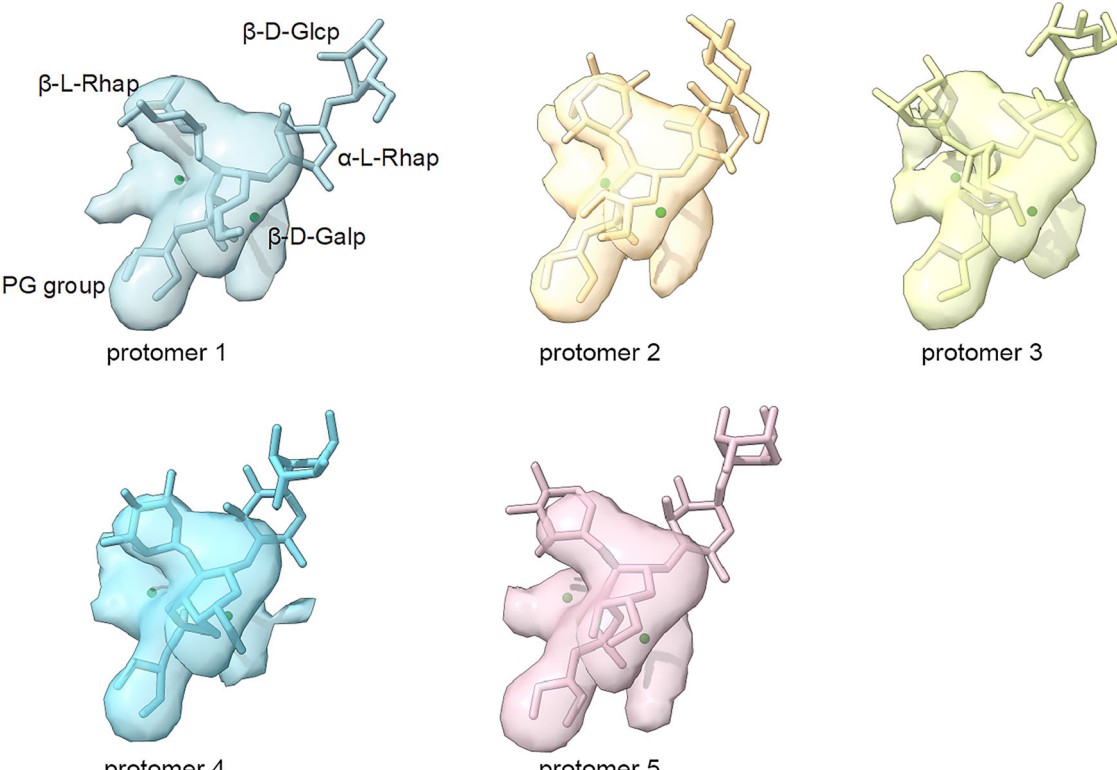

**Figure EV5.  Cryo-EM densities and atomic models of glycan ligand and associated calcium ions across five protomers in CRP.**

Cryo-EM densities (contoured at 3σ) are identical for protomer 1-5 due to imposed C5 symmetry. Atomic models reveal conserved binding features: two sugar rings (β-L-Rhap and β-D-Galp) and the PG group of a repeat unit fit well within the density, while two remaining sugar rings (α-L-Rhap and β-D-Glcp) extend beyond the density, consistent with the ligand's intrinsic flexibility. Source data are available online for this figure.

