## [Peer Review File · The EMBO Journal]

C-reactive protein has a broad-spectrum anti-bacterial function in the hepatic clearance of blood-borne bacteria

Danyu Chen, Jiao Hu, Mengran Zhu, Yufeng Xie, Hantian Yao, Haoran An, Yumin Meng, Juanjuan Wang, Xueting Huang, Yanni Liu, Zhujun Shao, Ye Xiang, Jianxun Qi, George Gao, and Jing-Ren Zhang

Corresponding author(s): Jing-Ren Zhang (zhanglab@tsinghua.edu.cn) , George Gao (gaof@im.ac.cn), Jianxun Qi (jxqi@im.ac.cn)

Review Timeline:

Submission Date:	22nd Feb 25
Editorial Decision:	16th Apr 25
Revision Received:	5th Aug 25
Editorial Decision:	6th Sep 25
Revision Received:	20th Sep 25
Accepted:	28th Sep 25

Editor: Ioannis Papaioannou

Transaction Report:

Dear Prof. Zhang,

Thank you again for submitting your manuscript EMBOJ-2025-120577 for consideration by The EMBO Journal, and for your patience during peer review. It has now been seen by three experts in the field, and we have received the full set of their detailed and informative reports, which you can find below.

As you will see, the referees indicate interest in the study and the findings, point out the high quality of the work, and explain that the manuscript makes an important contribution to the field. They also raise, however, a number of concerns that should be addressed in a revised version of the manuscript. Among others, there are concerns regarding sources of potential contamination for some of the experiments that must be addressed with the inclusion of appropriate controls, comments about remaining open questions that require clarification, and suggestions for more detailed descriptions of the used methods and improved presentation of the results and writing of the manuscript.

Given the referees' positive comments and recommendations, I would like to invite you to submit a revised version of your manuscript taking the referees' suggestions on board, along with a detailed point-by-point response addressing all referees' comments. I should add that it is The EMBO Journal policy to allow only a single round of major revision, and acceptance of your manuscript will therefore depend on the completeness of your responses in this revised version. Please let me know if you have any questions or comments that you would like to discuss with me. If there are any major points you do not agree with or cannot address during your revision, I would encourage you to share them with me as early as possible to discuss how to proceed further in the most efficient way.

We generally allow three months as standard revision time (July 16, 2025). As a matter of policy, competing manuscripts published during this period will not negatively impact our assessment of the conceptual advance presented by your study. However, we request that you contact us as soon as possible upon publication of any related work, to discuss how to proceed. Should you foresee a problem in meeting this three-month deadline, please let us know in advance and we will be able to grant an extension.

Thank you for the opportunity to consider your work for publication in The EMBO Journal. I look forward to your revision.

Best regards,

Ioannis

Instructions for preparing your revised manuscript

1. When you are ready to submit the revision, please upload:

- A Word file of the manuscript text (including legends of main Figures, EV Figures and Tables). Please make sure that changes are highlighted (or "tracked") to be clearly visible.

- Individual production-quality figure files (one file per figure). When assembling your figures, please refer to our figure preparation guidelines in order to ensure proper formatting and readability in print as well as on screen:

If the data shown in a figure are obtained from n {less than or equal to} 2, please use scatter plots showing the individual data points.

- i. the name of the statistical test used to generate error bars and P values
- ii. the number (n) of independent experiments (please specify technical or biological replicates) underlying each data point (discussion of statistical methodology can be reported in the Materials and Methods section, but figure legends should contain a basic description of n , P, and the test applied)
- iii. the nature of the bars and error bars (s.d., s.e.m.).

- A point-by-point response to the referees' comments, with a detailed description of the changes made (as a word file). All referees' concerns must be fully addressed and their suggestions taken on board. When preparing your letter of response to the referees' comments, please bear in mind that this will form part of the Review Process File and will therefore be available online to the community. Please note that you have the possibility to opt out of the transparent process at any stage prior to publication by letting the editorial office know (contact@embojournal.org); if you do opt out, the Review Process File link will point to the following statement: "No Review Process File is available with this article, as the authors have chosen not to make the review process public in this case.". For more details on our Transparent Editorial Process, please visit our website: <https://www.embopress.org/page/journal/14602075/authorguide#transparentprocess>

- Expanded View (EV) files (replacing Supplementary Information) that are collapsible/expandable online. A maximum of 5 EV Figures can be typeset. EV Figures should be cited as "Figure EV1, Figure EV2" etc. in the text, and their respective legends should be included in the manuscript file after the legends of regular figures. See detailed instructions regarding Expanded View files here: <https://www.embopress.org/page/journal/14602075/authorguide#expandedview>

- For the figures that you do NOT wish to display as Expanded View figures, they should be bundled together with their legends in a single PDF file called "Appendix", which should start with a short Table of Contents (including page numbers). Appendix figures should be referred to in the main text as: "Appendix Figure S1, Appendix Figure S2" etc. Please see detailed instructions here: <https://www.embopress.org/page/journal/14602075/authorguide#expandedview>

- A complete author checklist, which you can download from our author guidelines (<https://www.embopress.org/page/journal/14602075/authorguide>). Please note that the checklist will also be part of the Review Process File.

2. Please note that no statistics should be calculated and shown in Figures if $n=2$. Please also note that each p value should be reported as an exact value.

3. Before submitting your revision, primary datasets (and computer code, where appropriate) produced in this study need to be deposited in appropriate public databases (see <https://www.embopress.org/page/journal/14602075/authorguide#dataavailability>). In particular, we kindly request you to deposit all cryo-EM data reported in the manuscript. The accession numbers, database, and the specific URLs (links) should be listed in a formal "Data availability" section (placed after Methods), following the example below:

"The RNA-seq datasets produced in this study are available in the following database:
Gene Expression Omnibus GSE46843 (<https://www.ncbi.nlm.nih.gov/geo/query/acc.cgi?acc=GSE46843>)"

*** All links should resolve to a page where the data can be accessed. ***

*** Please remember to provide in the Data availability section of your revised manuscript reviewer passwords if the datasets are not yet public. ***

*** The Data Availability Section is restricted to new primary data that are part of this study. In case you have no data that require deposition in a public database, please state so instead of referring to the database: "Our study includes no data deposited in public repositories." under the heading "Data availability". ***

4. The materials and methods need to be described in the manuscript using our structured methods format, which is now required for all research articles. According to this format, the Methods section includes a single "Reagents and Tools Table" - listing key reagents, experimental models, software and relevant equipment including their sources and relevant identifiers - followed by a "Methods and Protocols" section describing the methods. Please download and fill our Reagents and Tools Table template (.docx), which you can find in our author guide:

<https://www.embopress.org/page/journal/14602075/authorguide#structuredmethods>. When submitting your revised manuscript, please do not include the Reagents and Tools Table in the Methods section of the manuscript but instead upload it as a separate file choosing the file type "Reagent Table".

5. Please check that the title and the abstract of the manuscript are brief, yet explicit, even to non-specialists. The length of the title should not exceed 100 characters, and the abstract should be a single paragraph not exceeding 175 words.

6. Please also note our reference format: <https://www.embopress.org/page/journal/14602075/authorguide#referencesformat>.

8. Please remember: digital image enhancement is acceptable practice, as long as it accurately represents the original data and

conforms to community standards. If a figure has been subjected to significant electronic manipulation, this must be noted in the figure legend or in the "Materials and Methods" section. The editors reserve the right to request original versions of figures and the original images that were used to assemble the figure.

9. Our journal encourages inclusion of data citations in the reference list to directly cite datasets that were obtained from public databases. Data citations in the article text are distinct from normal bibliographical citations and should directly link to the database records from which the data can be accessed. In the main text, data citations are formatted as follows: "Data ref: Smith et al, 2001" or "Data ref: NCBI Sequence Read Archive PRJNA342805, 2017". In the Reference list, data citations must be labeled with "[DATASET]". A data reference must provide the database name, accession number/identifiers, and a resolvable link to the landing page from which the data can be accessed at the end of the reference. Further instructions are available at: <https://www.embopress.org/page/journal/14602075/authorguide#referencesformat>.

10. We request authors to consider both actual and perceived competing interests. Please review our policy (<https://www.embopress.org/page/journal/14602075/authorguide#conflictofinterest>) and update your competing interests statement if necessary. Please name this section 'Disclosure and competing interests statement' and place it after the Acknowledgements section.

11. Please note that all corresponding authors are required to provide an ORCID ID upon submission of a revised manuscript (<https://orcid.org/>). Please find instructions on how to link your ORCID ID to your account in our manuscript tracking system in our Author guidelines (<https://www.embopress.org/page/journal/14602075/authorguide#authorshipguidelines>).

12. We use CRediT to specify the contributions of each author in the journal submission system. CRediT replaces the author contribution section, which should be removed from the manuscript. Please use the free text box to provide more detailed descriptions. See also guide to authors: <https://www.embopress.org/page/journal/14602075/authorguide#authorshipguidelines>.

14. We would also welcome the submission of cover suggestions or motifs to be used by our Graphics Illustrator in designing a cover.

15. Please use the link below to submit your revision:
<https://emboj.msubmit.net/cgi-bin/main.plex>

Referee #1:

Comments to the manuscript: "C-Reactive Protein Is A Broad-Spectrum Capsule-Binding Receptor For Hepatic Capture of Blood-Borne Bacteria", by Chen et al.

There are no line numbers in the submitted pdf file. The following is thus less precise, only identifying page numbers.

One major issue is that it is well known that preparations of pneumococcal capsular polysaccharides are contaminated with up to 10% (w/w) of the common antigen C-polysaccharide. This is discussed in, e.g., <https://www.sciencedirect.com/science/article/pii/S0022175924001194>, with references to other publications. Actually, in many pneumococcal strains, the C-polysaccharide is found to be covalently bound to the capsular polysaccharides. An exception is Type 3, where the C-polysaccharide is not found covalently bound to the capsular polysaccharide. Even polysaccharide preparations of vaccine quality are contaminated. Thus, when measuring pneumococcal antibodies by ELISA, it is routine to absorb antibodies directed against C-polysaccharide, as the analyses will otherwise be nonspecific. This reviewer is concerned that the ELISA tests are flawed by contamination. Since C-polysaccharide binds CRP, it is essential to have reagent controls in all experiments with pneumococci and pneumococcal polysaccharides. This could be, for example, to examine the amount of C-polysaccharide in the preparations using monoclonal antibody specific for PC (phosphoryl choline) or polyclonal rabbit antibody specific for C-polysaccharide. Both types of antibodies are available commercially. Since none of the structures of the pneumococcal capsular polysaccharides contain binding sites for these antibodies, ELISA measurements with these must be negative.

The same concerns apply to experiments with whole bacteria. Culturing is highly critical. It is well documented that cultures in the log phase contain large amounts of partially lysed bacterial cells, where cell wall antigens are exposed. Pneumococci lyse

very easily because they possess a highly active autolytic enzyme. In lysed bacteria, C-polysaccharide is exposed, which allows for CPR binding to the pneumococcal cells. As mentioned above, the authors must include relevant negative controls that document that their assays are specific for capsular polysaccharides and the CRP binding is not caused by structures such as C-polysaccharide.

The human pathogen *Haemophilus influenzae* adapts to different environments, e.g., culturing conditions, through variation of structures on its lipooligosaccharide (LPS), including phosphorylcholine. It is thus also important to establish whether the binding to capsular structures of *Haemophilus influenzae* is influenced by contaminations of phosphorylcholine.

Another issue is that when comparing binding intensities to the coated capsular structures (e.g., as in figures 1D and 1H and 6A and S2D), it is necessary to know how much is coated of each structure, presuming that they will not necessarily coat to an equal degree. Carbohydrate structures will not bind to, e.g., maxisorp wells - some scientists use activated wells to catch carbohydrate structures. It may be different for carbohydrate structures containing more complex structures.

The important ELISA method used for detecting the binding of CRP is not described in any detail in the given reference: Wang, J., H. An, M. Ding, Y. Liu, S. Wang, Q. Jin, Q. Wu, H. Dong, Q. Guo, X. Tian, J. Liu, J. Zhang, T. Zhu, J. Li, Z. Shao, D.E. Briles, J.W. Veening, H. Zheng, L. Zhang, and J.R. Zhang. 2023. Liver macrophages and sinusoidal endothelial cells execute vaccine-elicited capture of invasive bacteria. *Sci Transl Med* 15:eade0054.

Comments on the proteomic analysis.

The only description of the procedure I can find in the method section for this analysis is on page 19): "Proteins bound to the CPS-coated beads were identified by mass spectrometry."

To enable other scientists to reproduce data, it is very important that there is a detailed, correct description of the methods used. The authors need to explain the sample prep, measurement & data acquisition parameters (instrument, DDA, or DIA), and data analysis in detail. Depending on the software they used, there would also be more things to declare.

Page 5. A "membrane protein-enriched fraction of mouse liver nonparenchymal cells (NPCs) in the presence of 10% mouse serum" are exposed to beads exposing capsular structures. A control without serum would normally be required - i.e., how many proteins will bind with no serum being present?

The authors only label the 5 proteins on the bottom right in Fig. 1C - what about the other 5 down there? There is space to label those - and a correlation quantification is also missing.

Proteomic data should be presented as an Excel table with the log₂-transformed intensities and include additional data. The authors should also deposit the mouse data in public repositories like the PRIDE database.

I would have expected a supplementary table with the full extended list (e.g., # of identified unique peptides, MW, PSMs, quality parameters for identification, and log₂ or log₁₀-transformed intensities - nobody reports raw intensities in proteomics) and a short list with the top 10 of each category in the main text. The full list will be in a supplementary document.

Comments on the structure determinations.

The glycan ligands seem a bit different in the 5 copies, but in principle, they should be the same. The validation reports indicate that the 5 ligands are not fitted very precisely. Thus, it is important that the authors show for each of the five glycan ligands and the associated Ca²⁺ ions the density for the ligands at the same threshold level for all five instances.

Figure 6. A revised manuscript must offer panels inserted in Figure 6 (not in supporting information) that demonstrate the quality of the map for the CPS23F ligand and the Ca²⁺ ion. This will allow the reader to judge whether the inclusion of the ligand is justified; this is not documented in the current versions. Further panels documenting the quality of the density elsewhere in CRP can be placed in the supporting information. Table S6 must also include map-model correlation coefficients, including an explicit calculation for the CPS23F ligand.

Figure 6. panel C, to the left, is likely to show the experimental 3D map (please give the contour level in a revision), to the right of the cartoon representation of the atomic model built into the map. The difference should be made clear. The role of AF3 prediction in panel F is not obvious, and if the human structure presented is that from 1b09, this should be made clear in figure 6. Also, how is the AF3 prediction for the mouse complex of PG in panel F prepared? Was the apo state predicted and then PG superimposed by comparison with the experimental human structure? PG is not in the standard list of ligands for alphaFold3.

Minor issues.

Page 6. A sentence read: "Furthermore, passive i.v. administration of 1 µg r-mCRP fully enabled Crp^{-/-} mice to rapidly capture Sp23F by the liver and clear bacteria from the circulation (Fig. 2C)." It should be commented somewhere in the text that it seems that a very low amount of CRP is able to convert the phenotype - this 1 µg must very fast be diluted to quite low concentrations of CRP.

Page 9, top. "These data indicated that mouse CRP accommodates various structures of bacterial capsules with a highly dynamic platform."

I think this should be discussed in a bit more detail in the discussion section.

Page 16." Although human CRP-PC complex is known to activate the complement classical pathway via binding to C1q (Agrawal et al., 2001), C1q-deficient mice displayed normal level of CRP-mediated bacterial clearance. This finding indicates that CRP activates C3 via an C1q-independent mechanism."

The report <https://pubmed.ncbi.nlm.nih.gov/17581635/> shows that hCRP interacts with ficolin and that this interaction stabilizes CRP binding to bacteria and activates the lectin-mediated complement pathway. Thus, other ways to activate the complement system via CRP binding may exist.

Page 20. "All infection experiments were conducted in C57BL/6 (6-8 weeks old) according to the animal protocols approved by the Institutional Animal Care and Use Committee in Tsinghua University."

The protocol number must be given.

Page 23. "C3 deposition on *S. pneumoniae* was detected by immunoblotting as described (Zhang et al., 1997)."

The conditions for the western blots are not given in the given reference: Zhang, J.-R., J.M. Hardham, A.G. Barbour, and S.J. Norris. 1997. Antigenic variation in Lyme disease borreliae by promiscuous recombination of VMP-like sequence cassettes. *Cell* 89:275-285.

I agree that mouse models are very often used for testing the infectivity of pneumococci, i.e., and to test whether antibodies are protective - and that it is thus a justifiable animal model when compared with the literature. On the other hand, pneumococci are very seldom pathogenic for mice or other animals (e.g., for calves or hamsters).

Catalog numbers are missing in the methods section.

Throughout the text, there is missing a "the" in many places. Run the text through a grammar program.

Referee #2:

C-reactive protein (CRP) is known to be associated with infections. It was believed to bind to capsular polysaccharides that contain phosphocholine (PC) or phosphoglycerol (PG). CRP also binds to other molecules on the bacterial cell surface, such as teichoic acids and lipopolysaccharides (LPS). The simplest explanation for these binding events is the presence of PC or PG. However, this requirement appears to have exceptions. One example is the depyruvylated type-IV capsular polysaccharide derived from type 27 (PMID: 30863393). Additionally, CRP collaborates with plasma lectins, which may help recognize functional groups beyond PC and PG.

In this study, Chen et al. elegantly demonstrated that CRP is a central defense mechanism against pneumococcus and other capsule-producing pathogens. They illustrated that CRP recognizes capsular polysaccharides (CPS) at the surface of the bacterial cell, resulting in their clearance by Kupffer cells. Without this mechanism, these 'CRP-sensitive' bacteria could proliferate in the bloodstream, leading to sepsis and increased mortality. Overall, the work is of a high standard and generally well-written. In particular, the experiments were carefully controlled, with an impressive knock-out mice collection. The authors are congratulated for their excellent contribution to the field.

Suggestions

The work is respectable. Yet, its contribution mainly focuses on clarifying the downstream pathway of CRP-mediated immunity. Some of the claims in the discussion distract readers from this point and may be controversial. For example, "CRP-mediated immunity appears to 'operate in humans'" and "this study has thus discovered a potent and broad-spectrum anti-bacterial function for CRP" are not as good representations as the work deserves. A careful revision of the draft and another round of grammar checks may help.

There are no line numbers. I will describe the locations as best as I can.

p.17 The authors suggested that "we have demonstrated that the endogenous level of CRP in mice is sufficient to confer potent protection." Have they measured the level of CRP during the experimental timeframe to confirm there was no induction? If there is an induction, is it serotype-dependent?

p.17 The proposal to treat drug-resistant encapsulated bacteria with CRP may require reconsideration. As the authors pointed out, CRP levels can rise to 1 mg/ml during infection or inflammation. The benefit of injecting additional CRP remains unclear.

p.47 One concern is the Δ cps strain binds CRP (PMID: 10225891). If so, they are expected to be captured by Kupffer cells facilitated by this and other mechanisms. A plausible explanation for the absence of a phosphate group in the CRP-sensitive serotype is that the capsule is too thin. Consequently, the underlying antigens like the PC-containing wall teichoic acids may be exposed to the cell surface. A simple FACS with anti-phosphocholine antibodies will eliminate this possibility.

Other minor comments

p.3 Is the CRP gene polymorphism associated with bacteremia a serotype-specific phenomenon? Are the CRP-sensitive

serotypes less invasive or less prevalent?

p.12 Define flipped conformation. Change 'CSP23F' to CPS23F. The authors should discuss the equivalent residues of hCRP and mCRP, and correlate these variants on their ability to bind CPS23F.

p.12 The reason behind hCRP not binding to four CPSs recognized by mCRP was not discussed. Perhaps the structure solved can help. What does it mean by "mouse CRP accommodates various structures of bacterial capsules with a highly dynamic platform."?

p.13 Revise "While the ELISA result revealed significant binding of hCRP to all of the 17 mCRP-binding capsules of *S. pneumoniae* (Fig. 6A), and thus demonstrated the common characteristic of human and mouse CRPs in anti-pneumococcal immunity."

p.14 CRP is a hallmark. Delete 'thus far'.

p.15 'a' significant impairment; Change 'our limited screening has' to 'We'; Also, the notion that 'CRP acts like an anti-capsule antibody' needs to be more substantiated.

p.15 "Except for the PC moiety of serotype-27 capsule of *S. pneumoniae*, none of the remaining 20 CRP-binding capsules possess the PC group in their repeat units." When I looked into the structures, many contain PG. This should be mentioned such that the noteworthy exceptions stand out (e.g., serotype 27).

p.16 Revise this line: "Although human CRP-PC complex is known to activate the complement classical pathway via binding to C1q (Agrawal et al., 2001), C1q-deficient mice displayed [a] normal level of CRP-mediated bacterial clearance. This finding indicates that CRP activates C3 via a[n] C1q-independent mechanism." In mice, C1q is not involved. However, as the authors suggested in the limitations of this study, mCRP may behave differently from hCRP regarding C1q binding. Thus, it remains unclear whether CRP activates C3 via a C1q-independent mechanism.

p.16 The authors may want to use the first sentence as the subsection title. So, rephrase: "This study demonstrates that CRP engages liver macrophages to remove blood-borne bacteria."

p.17 Reword a few lines: "The current literature is overwhelming concentrated on human CRP."; "irresponsiveness of mouse CRP to microbial infections"; "there is a high level of high sequence similarity"

p.38 Fig. 6A showed the result of an end-point of an ELISA. It remains unclear whether

p.48 The negative control, i.e., the CPS14, should be shown in the main figure.

p.49 Fig. S7A. Are the residues important for binding the ligands conserved?

Referee #3:

The manuscript describes the exciting discovery of a function of C-reactive protein (CRP) in promoting clearance of bacteria (focusing on pneumococci but also addressing *H. influenzae* and *Klebsiella*) producing capsular serotypes that it recognizes. CRP likely promotes complement fixation on bacteria promoting their clearance by complement receptors on Kupffer cells. CRP was identified by pulldown from serum by bacteria of different serotypes and its subsequent analysis is comprehensive. The investigators inoculate a collection of bacteria with differing capsular structures into wild type mice or mice deficient in CRP, C3, or complement receptors mice, then monitor clearance, clinical course, and interaction with liver sinusoids by intravital microscopy. By using mice deficient in FcRa and C1q, they rule out a role for these receptors in CRP-mediated clearance. They also use fluorescent microscopy to assess CRP binding by whole bacteria, biochemical studies to measure binding affinities, cryo-EM to examine structure, mutagenesis to probe structure-function relationships, and in vitro studies with human CRP to extend the relevance of the study to human immunity. Although description of the structural insights gained could be more clearly articulated (see point 1 below), overall, the manuscript is very clearly written and the discovery that CRP functions as an antibody-like molecule that recognizes a broad range of bacterial capsules to promote phagocytic clearance and control bacteremia is exciting.

The authors should address the following:

1. Page 10, second paragraph and Fig. 5. (a) I think a more stringent conclusion of the results of panels A and B is that the presence of PC is not sufficient to confer high affinity binding. (b) Based on the panels A and B, it is not clear that "CRP recognizes bacterial capsules via distinct binding modes". Rather, this statement is better supported by the data in panel C; (c) In the absence of more detailed structural information, the results of panels C, D and E are open to a variety of interpretations. Correspondingly, the first sentence on page 11 and the last sentence of the following paragraph are vague. Some acknowledgement of the difficulty of drawing definitive conclusions from these experiments is warranted and would not detract

from this very nice overall study.

2. Fig. 6B. I think the authors are proposing that the increase in bacterial binding to Kupffer cells is due to complement fixation of bacteria. If so, (a) can complement be detected on bacteria in the presence of NHS? Is it increased by the addition of exogenous CRP? Does the boost in binding due to CRP depend on NHS? Can the binding be inhibited by exogenous capsule of a serotype human CRP binds but not by serotype 8?

3. The authors should provide more explication of statistical significance for some of the results. Examples:

a. Fig. 1D, E. How was significance determined? Do error bars represent SE?

b. Last line, page 7. How is "significantly" defined? More generally, what statistical analyses are performed on CFU counts at 5 and 30 min? In figure 2, the differences are stark, but in figure 3, some differences are moderate and could be evaluated statistically.

c. Fig. 2 vs. Fig. 4A and page 9, line 6. Statistically compare Ct50 to document the statement.

d. Fig. 4C and D. A statistical comparison of C3 KO vs. CR3/CR1g KO mice might provide evidence that one need not invoke any other receptors to explain defects. In addition, it is not clear from Figure/figure legend that C and D differ in inoculum.

e. Fig. 5. the KD of CRP for 23F is similar to that of 33F--is there a way to evaluate whether this difference is significant? If not, the text should be softened. In addition, the concentrations are given in $\mu\text{g/ml}$ whereas concentrations in μM are more intuitive. See also Fig. S3 and EC50.

f. Fig. 6. What is the significance of the open bar in 6A for serotype 8? what is considered "background" and how was that determined?

g. Page 9, line 10. the relationship between Fig. S1A and the dose chosen is not clear.

Minor points:

4. Fig. 2A and C. Consider including depicting structures of 23F and 14 in Fig. 2A to emphasize different structures. Indicate on Fig. 2C that infecting strain was 23F and that the mouse was CRP-deficient. (The 23F structure in Fig. 1A is not referred to and is not compared to other serotypes in Fig. 1 and could be removed.) Finally, the phrase "demonstrated that CRP is necessary and sufficient for serotype-specific hepatic capture and killing of serotype-23F pneumococci" might be more accurately read "demonstrated that the only defect of CRP-deficient mice in serotype-specific hepatic capture and killing of serotype-23F pneumococci is CRP".

5. P. 7. 5 lines from bottom. Might be clearer as "serotypes, i.e. all except serotype e, were..."

6. P. 11, line 14. Remove "While".

7. Page 9, last line of first paragraph. "relies" suggests that the C3-independent mechanism is the sole means of clearance; "requires" might better convey the authors' meaning.

8. Fig. 5. The figure would be easier for readers to digest if the authors made use of colors like their use in other figures, e.g., have 27, 23F, and 33F different colors and keep them consistent throughout all panels, and in Panel A, to make things obvious, authors could indicate the PC and PG groups on 27 and 23F, respectively. In addition, do the authors think that the stoichiometry of 5:1 for 23F is based on the pentameric structure of CRP? How would the authors explain the stoichiometry of 33F (2.9)?

Responses to Reviewer's Comments

Manuscript No: EMBOJ-2025-120577

Title: C-Reactive Protein Is A Broad-Spectrum Capsule-Binding Receptor For Hepatic Capture of Blood-Borne Bacteria

Date: July 16, 2025

We greatly appreciate the reviewer's insightful evaluation and suggestions of our manuscripts. The entire paper has been revised according to the comments by performing additional experiments, and modifying the text. The revised text sections are tracked with red characters, and the point-by-point responses to reviewers' comments are provided as below.

Referee #1

Major issues

1. This reviewer is concerned that the ELISA tests are flawed by contamination of cell wall polysaccharide (C-polysaccharide).

Response: We tested potential impact of C-polysaccharide on our ELISA readouts for CRP-capsule interactions using a monoclonal antibody to cell wall phosphocholine (PC). As described in the literature, a substantial level of PC contaminant was indeed detected in capsular polysaccharide purified from serotype-23F *S. pneumoniae* with the anti-PC antibodies. The further test showed that the PC contaminant was significantly removed from the capsular polysaccharide by the combination of anti-PC antibodies and protein G-beads (revised **Fig. EV1B**), but the same depleting did not obviously impact the level of CRP-capsule binding interactions as detected by ELISA (Fig. 1E). This information is added to the revised text (page 5, lines 128-134).

2. This reviewer has the same concerns on potential impact of cell wall phosphocholine on CRP-capsule binding at the surface of whole bacteria.

Response: We addressed this concern by comparing bacterial binding profiles between CRP and an anti-phosphocholine (PC) monoclonal antibody using flow cytometry. The anti-PC antibody showed a comparable level of binding to CRP-unrecognizable serotype-8 and CRP-recognizable serotype-23F pneumococci (revised **Fig. 1G**). In contrast, CRP bound to virtually all serotype-23F bacteria (98.5%), but the protein showed negligible binding to serotype-8 bacteria (1.5%) (revised **Fig. 1F**). This result indicates that the capsule is much more dominant ligand of CRP as compared with the C-polysaccharide. The text has been revised to reflect this (page 5, lines 135-138).

3. The human pathogen *H. influenzae* adapts to different environments, e.g., culturing conditions, through variation of structures on its lipooligosaccharide (LPS), including phosphorylcholine. It is thus also important to establish whether the binding to capsular structures of *H. influenzae* is influenced by contaminations of phosphorylcholine.

Response: We used flow cytometry to detect phosphorylcholine on the intact cells of type b *H. influenzae*. In contrast to 51.8% CRP-binding bacteria, less than 1% of the cells showed detectable binding to phosphocholine. This result showed that the *Hib* strain used in this work did not possess substantial level of phosphorylcholine.

This result has been added to revised **Fig. S1** and **Fig. S4** and text (page 9, lines 264-269).

4. Another issue is that when comparing binding intensities to the coated capsular structures (e.g., as in figures 1D and 1H and 6A and S2D), it is necessary to know how much is coated of each structure, presuming that they will not necessarily coat to an equal degree. Carbohydrate structures will not bind to, e.g., maxisorp wells - some scientists use activated wells to catch carbohydrate structures. It may be different for carbohydrate structures containing more complex structures.

Response: We evaluated the coating efficiency of capsular polysaccharides onto 96-well plates used in this study by adding various concentrations of purified capsular polysaccharides of *S. pneumoniae* serotypes 5, 6A and 8 A (0.01-10 µg/ml) before probing with sera from mice immunized with the 13-valent pneumococcal conjugate vaccine (PCV13). PCV13 contains serotypes 5 and 6A but not serotype 8. The ELISA results revealed an antigen dose-dependent reactivity of the immune serum to the wells coated with capsular polysaccharides of serotypes 5 and 6A, with marginal signal to the counterpart of serotype 8 (revised **Fig. EV1A**). The reactivity of both the vaccine-covered serotypes reached the plateau when the antigens were coated at a concentration of 1 µg/ml. Therefore, the concentration used to coat 96-well plates for CRP binding (10 µg/ml) represents a saturating level for the 96-well plates used in this work, and should not affect our conclusions on the serotype-specific capsule binding of CRP. In addition, we have cited a new reference for the ELISA method in the revision (Agrawal et al., 2002). We have added this information in the revised Results (page 5, lines 127-128) and Methods sections (page 24, lines 716-725).

5. Comments on the proteomic analysis.

1) The only description of the procedure I can find in the method section for this analysis is on page 19): "Proteins bound to the CPS-coated beads were identified by mass spectrometry." To enable other scientists to reproduce data, it is very important that there is a detailed, correct description of the methods used. The authors need to explain the sample prep, measurement & data acquisition parameters (instrument, DDA, or DIA), and data analysis in detail. Depending on the software they used, there would also be more things to declare.

Response: We have now revised the Methods section to include detailed descriptions of sample preparation, LC-MS/MS parameters, and database search process (pages 22, lines 645-653).

2) Page 5. A "membrane protein-enriched fraction of mouse liver nonparenchymal cells (NPCs) in the presence of 10% mouse serum" are exposed to beads exposing capsular structures. A control without serum would normally be required - i.e., how many proteins will bind with no serum being present?

Response: The data with the samples without serum has been added in the revised Table S2.

3) The authors only label the 5 proteins on the bottom right in Fig. 1C - what about the other 5 down there? There is space to label those - and a correlation quantification is also missing.

Response: We have added this information in **Fig. 1C and I** and text (page 5, lines 120-124).

4) Proteomic data should be presented as an Excel table with the log₂-transformed intensities and include additional data. The authors should also deposit the mouse data in public repositories like the PRIDE database.

Response: We have presented the experimental procedures and the lists of CPS23F- and CPS8-enriched proteins are described in Tables S8 (mouse serum plus KC), 9 (mouse KC), and 10 (human serum). The raw and processed proteomic data have been deposited in the iProX database under accession number IPX0012537001. This information is added in the Methods section (page 22, lines 657-660).

5) I would have expected a supplementary table with the full extended list (e.g., # of identified unique peptides, MW, PSMs, quality parameters for identification, and log₂ or log₁₀-transformed intensities - nobody reports raw intensities in proteomics) and a short list with the top 10 of each category in the main text. The full list will be in a supplementary document.

Response: The proteomic data are presented in three layers in the revision: 1) Tables S1-3 present the simplified lists of capsule-enriched proteins; 2) Tables S8-10 provide the full lists of proteins and the experimental procedures of capsule-based pulldown; 3) the raw data are deposited in the iProX database under accession number IPX0012537001.

The top 10 hits of each category are listed in the revised text (page 5, lines 122-124 for mouse data; page 6, lines 147-148 for human data).

6. Comments on the structure determinations.

The glycan ligands seem a bit different in the 5 copies, but in principle, they should be the same. The validation reports indicate that the 5 ligands are not fitted very precisely. Thus, it is important that the authors show for each of the five glycan ligands and the associated Ca²⁺ ions the density for the ligands at the same threshold level for all five instances.

Response: Figure 6. Macromolecules often exhibit considerable conformational flexibility, particularly at their active sites. Accordingly, when CRP binds to 23F, it is expected to display a certain degree of dynamic behavior—an intrinsic property of such molecules. Under crystal packing conditions, ligand density typically appears more uniform. In contrast, cryo-EM captures subtle variations in ligand conformation arising from molecular dynamics, which in fact reflect the inherent flexibility of the biomacromolecule. We have now included five additional panels in **Fig. S10**, each showing map (contoured at 3σ, consistent across all copies), glycan ligand and associated Ca²⁺ ions density. These panels demonstrate the electron density for each individual glycan ligand (CPS23F) in all five CRP monomers, allowing direct comparison of fitting quality. In the single repeating unit of CPS23F, two sugar rings were well-resolved within the density map, while two others extended beyond the interpretable density, and we described it in the main text (page 13, lines 390-397).

Figure 6. A revised manuscript must offer panels inserted in Figure 6 (not in supporting information) that demonstrate the quality of the map for the CPS23F ligand and the Ca²⁺ ion. This will allow the reader to judge whether the inclusion of the ligand is justified; this is not documented in the current versions. Further panels documenting the quality of the density elsewhere in CRP can be placed in the supporting information. Table S6 must also include map-model correlation

coefficients, including an explicit calculation for the CPS23F ligand.

Response: We have revised Figure 7 to include the following additions:

As stated in the response above, we have now included five additional panels in **Fig. EV5**. Each panel presents the electron potential (EP) map contoured at 3σ (consistently applied across all protomers), along with the corresponding densities for the glycan ligand and associated Ca^{2+} ions. Since these panels primarily highlight the EP maps and atomic models of the glycan ligand and Ca^{2+} ions across the five protomers, it is more appropriate to include them in the Supplementary Information rather than in the main figure.

1. New panels displaying the electron density maps (contoured at 3σ) for the CPS23F ligand and Ca^{2+} ion (**Fig. 7D**).

2. Further panels documenting the quality of the density elsewhere in CRP are now placed in **Fig. S10**.

3. Map-model correlation coefficients (CC) for these regions, now explicitly listed in **Table S4**, including a dedicated calculation for the CPS23F ligand.

Figure 6, panel C, to the left, is likely to show the experimental 3D map (please give the contour level in a revision), to the right of the cartoon representation of the atomic model built into the map. The difference should be made clear. The role of AF3 prediction in panel F is not obvious, and if the human structure presented is that from 1b09, this should be made clear in figure 6. Also, how is the AF3 prediction for the mouse complex of PG in panel F prepared? Was the apo state predicted and then PG superimposed by comparison with the experimental human structure?

Response: We have carefully revised the figure and its legend. The legend is revised to clearly distinguish between the experimental electron density maps and atomic model representations, and to specify the contour level of the density maps.

The AF3 prediction in Figure 7F was used to validate structural conservation between human and mouse complexes. The human structure is from 1b09, as explicitly stated in the revised legend. The mouse CRP AF3 prediction was generated in the apo state. The bound PG in mouse CRP (mCRP) was generated by superimposition of the hCRP-PG structure (1b09, conserved binding pocket, key residues). The predicted mCRP structure and its comparison with hCRP show differences in the loop 68-72.

Minor issues

1. Page 6. A sentence read: "Furthermore, passive i.v. administration of $1\ \mu\text{g}$ r-mCRP fully enabled *Crp*^{-/-} mice to rapidly capture *Sp23F* by the liver and clear bacteria from the circulation (Fig. 2C)." It should be commented somewhere in the text that it seems that a very low amount of CRP is able to convert the phenotype - this $1\ \mu\text{g}$ must very fast be diluted to quite low concentrations of CRP.

Response: We apologize for the errors in the **Fig. 2G** legend although the experimental setting was correctly described in the Methods section. *Sp23F* pneumococci (10^6 CFU) were pre-treated with $100\ \mu\text{l}$ r-mCRP ($10\ \mu\text{g}/\text{ml}$) for 5 min at room temperature before i.v. inoculation. The legend has been corrected.

2. Page 9, top. "These data indicated that mouse CRP accommodates various structures of bacterial capsules with a highly dynamic platform." I think this should be discussed in a bit more detail in the discussion section.

Response: This information is added in the Discussion section.

3. Page 16." Although human CRP-PC complex is known to activate the complement classical pathway via binding to C1q (Agrawal et al., 2001), C1q-deficient mice displayed normal level of CRP-mediated bacterial clearance. This finding indicates that CRP activates C3 via an C1q-independent mechanism." The report <https://pubmed.ncbi.nlm.nih.gov/17581635/> shows that hCRP interacts with ficolin and that this interaction stabilizes CRP binding to bacteria and activates the lectin-mediated complement pathway. Thus, other ways to activate the complement system via CRP binding may exist.

Response: We tested multiple possibilities that may functionally link CRP and C3 in promoting hepatic trapping of blood-borne pneumococci by assessing the early clearance of serotype-23F *S. pneumoniae* in mice lacking ficolin A, MASP2 (a key component of the lectin complement activation pathway) or C4 (a key factor in the classical and lectin complement activation pathways). The results showed that CRP-mediated bacterial clearance remained intact in all of the three mouse lines (revised **Fig. EV4**), thus excluding the functional involvement of ficolin A and the classical/lectin pathways. We have added this information in the revised Results (page 11, lines 307-315) and Discussion (page 17, lines 510-520).

4. Page 20. "All infection experiments were conducted in C57BL/6 (6-8 weeks old) according to the animal protocols approved by the Institutional Animal Care and Use Committee in Tsinghua University." The protocol number must be given.

Response: We have added the protocol number (No. 22-ZJR1) in the revision (page 25, line 760).

5. Page 23. "C3 deposition on *S. pneumoniae* was detected by immunoblotting as described (Zhang et al., 1997)." The conditions for the western blots are not given in the given reference: Zhang, J.-R., J.M. Hardham, A.G. Barbour, and S.J. Norris. 1997. Antigenic variation in Lyme disease borreliae by promiscuous recombination of VMP-like sequence cassettes. *Cell* 89:275-285.

Response: We have replaced the western blotting of C3 on capsular polysaccharide with new flow cytometry data showing C3 deposition on intact bacteria in the revision (shown in revised **Fig. 7C**). The text has been accordingly modified (page 13, lines 375-379).

6. I agree that mouse models are very often used for testing the infectivity of pneumococci, i.e., and to test whether antibodies are protective - and that it is thus a justifiable animal model when compared with the literature. On the other hand, pneumococci are very seldom pathogenic for mice or other animals (e.g., for calves or hamsters).

Response: We revised the Discussion section to reflect the limitation of our data.

7. Catalog numbers are missing in the methods section.

Response: This and other lines of the reagent information have been placed in the Reagents and tools table.

8. Throughout the text, there is missing a "the" in many places. Run the text through a grammar program.

Response: We have revised the full text.

Referee #2

Major issues

1. The work is respectable. Yet, its contribution mainly focuses on clarifying the downstream pathway of CRP-mediated immunity. Some of the claims in the discussion distract readers from this point and may be controversial. For example, "CRP-mediated immunity appears to 'operate in humans'" and "this study has thus discovered a potent and broad-spectrum anti-bacterial function for CRP" are not as good representations as the work deserves. A careful revision of the draft and another round of grammar checks may help.

Response: We have thoroughly revised the Discussion section.

2. The authors suggested that "we have demonstrated that the endogenous level of CRP in mice is sufficient to confer potent protection." Have they measured the level of CRP during the experimental timeframe to confirm there was no induction? If there is an induction, is it serotype-dependent?

Response: We measured serum CRP levels of mice at multiple time points following intravenous infection with serotype-23F *S. pneumoniae*. The results have confirmed the information in the literature that CRP is not significantly induced in mice during systemic bacterial infection. The result is placed in **Fig. EV2C**, and described in the text (page 7, lines 174-178).

3. p.17 The proposal to treat drug-resistant encapsulated bacteria with CRP may require reconsideration. As the authors pointed out, CRP levels can rise to 1 mg/ml during infection or inflammation. The benefit of injecting additional CRP remains unclear.

Response: We have revised the Discussion section to address this valid point.

4. p.47 One concern is the Δcps strain binds CRP (PMID: 10225891). If so, they are expected to be captured by Kupffer cells facilitated by this and other mechanisms. A plausible explanation for the absence of a phosphate group in the CRP-sensitive serotype is that the capsule is too thin. Consequently, the underlying antigens like the PC-containing wall teichoic acids may be exposed to the cell surface. A simple FACS with anti-phosphocholine antibodies will eliminate this possibility.

Response: This issue is related to Referee #1 comments on points 1-3. To ascertain the impact of the well-known CRP binding to the C-polysaccharide on the host-pathogen interaction, we tested the clearance of an isogenic acapsular strain, which maximally expose the C-polysaccharide at the surface. The Δcps mutant was effectively cleared from the circulation of WT and *Crp*^{-/-} mice in a similar manner. We have added this information in **Fig. 2C** and text (page 6, lines 168-173).

Minor issues

1. p.3 Is the CRP gene polymorphism associated with bacteremia a serotype-specific phenomenon?

Response: Many single nucleotide polymorphisms in the CRP gene locus, but none of these change the amino acid sequence of CRP. Polymorphisms in the promoter region of human CRP gene is associated with the increased mortality in pneumococcal bacteremia patients. This information is added in the revised

Introduction (page 3, lines 49-53).

2. Are the CRP-sensitive serotypes less invasive or less prevalent?

Response: Our literature search did not find strict correlation between CRP recognition and epidemiological prevalence of encapsulated bacteria. We discussed the possibilities in the revised Discussion (pages 17-18, lines 531-543).

3. The authors should discuss the equivalent residues of hCRP and mCRP, and correlate these variants on their ability to bind CPS23F.

Response: Elaborated in the revised Discussion.

4. p.12 The reason behind hCRP not binding to four CPSs recognized by mCRP was not discussed. Perhaps the structure solved can help. What does it mean by "mouse CRP accommodates various structures of bacterial capsules with a highly dynamic platform."?

Response: Elaborated in the revised Discussion.

5. p.13 Revise "While the ELISA result revealed significant binding of hCRP to all of the 17 mCRP-binding capsules of *S. pneumoniae* (Fig. 6A), and thus demonstrated the common characteristic of human and mouse CRPs in anti-pneumococcal immunity."

Response: Revised (page 12, lines 366-369).

6. p.14 CRP is a hallmark. Delete 'thus far'.

Response: Revised.

7. p.15 'a' significant impairment; Change 'our limited screening has' to 'We'; Also, the notion that 'CRP acts like an anti-capsule antibody' needs to be more substantiated.

Response: Revised.

8. p.15 "Except for the PC moiety of serotype-27 capsule of *S. pneumoniae*, none of the remaining 20 CRP-binding capsules possess the PC group in their repeat units." When I looked into the structures, many contain PG. This should be mentioned such that the noteworthy exceptions stand out (e.g., serotype 27).

Response: Revised.

9. p.16 Revise this line: "Although human CRP-PC complex is known to activate the complement classical pathway via binding to C1q (Agrawal et al., 2001), C1q-deficient mice displayed [a] normal level of CRP-mediated bacterial clearance. This finding indicates that CRP activates C3 via a[n] C1q-independent mechanism." In mice, C1q is not involved. However, as the authors suggested in the limitations of this study, mCRP may behave differently from hCRP regarding C1q binding. Thus, it remains unclear whether CRP activates C3 via a C1q-independent mechanism.

Response: This information is added in the revised Discussion (page 20, lines 607-611).

10. p.16 The authors may want to use the first sentence as the subsection title. So, rephrase: "This study demonstrates that CRP engages liver macrophages to remove blood-borne bacteria."

Response: We have revised in the text (page 19, lines 570-571).

11. p.17 Reword a few lines: "The current literature is overwhelming concentrated on human CRP."; "irresponsiveness of mouse CRP to microbial infections"; "there is a high level of high sequence similarity"

Response: We have revised in the text.

12. p.48 The negative control, i.e., the CPS14, should be shown in the main figure.

Response: We have added the information in **Fig. 6B** and the text (page 11, line 330).

13. p.49 Fig. S7A. Are the residues important for binding the ligands conserved?

Response: Most of these residues are highly conserved across species, but there are some differences between human and mouse CRPs. Specifically, the residues at positions 60 and 147 differ between human and mouse CRP. Additionally, at positions 66, 76, and 81, we observed species differences, with position 76 being a notable variation.

Referee #3

Major issues

1. Page 10, second paragraph and Fig. 5. (a) I think a more stringent conclusion of the results of panels A and B is that the presence of PC is not sufficient to confer high affinity binding. (b) Based on the panels A and B, it is not clear that "CRP recognizes bacterial capsules via distinct binding modes". Rather, this statement is better supported by the data in panel C; (c) In the absence of more detailed structural information, the results of panels C, D and E are open to a variety of interpretations. Correspondingly, the first sentence on page 11 and the last sentence of the following paragraph are vague. Some acknowledgement of the difficulty of drawing definitive conclusions from these experiments is warranted and would not detract from this very nice overall study.

Response: We have modified these statements in the revision (page 11, line 335; pages 12, lines 336-337; page 12, lines 349-351).

2. Fig. 6B. I think the authors are proposing that the increase in bacterial binding to Kupffer cells is due to complement fixation of bacteria.

a. If so, (a) can complement be detected on bacteria in the presence of NHS?

Response: To address this question, we incubated 10^6 CFU of *S. pneumoniae* serotype 23F or serotype 8 with normal human serum (NHS), heat-inactivated serum (HI-NHS), or PBS at 37°C for 30 min, followed by flow cytometric analysis of C3 deposition. The results showed that serotype 23F strongly activated complement, with 93.4% C3-positive signal observed in the presence of NHS, but not with HI-NHS (1.0%) or PBS (0.8%). In contrast, serotype 8 exhibited minimal C3 deposition (35.1%). These findings demonstrate that serotype 23F, but not serotype 8, can effectively activate complement when incubated with NHS. These data have been included in the revised manuscript as Figure 7C to address this concern. The main text has also been accordingly revised (page 13, lines 375-377).

b. Is it increased by the addition of exogenous CRP? Does the boost in binding due to CRP depend on NHS?

Response: In response to the reviewer's question regarding whether complement deposition is increased by the addition of exogenous CRP, we performed flow cytometric analysis of C3 deposition on *S. pneumoniae* serotype 23F in the presence of NHS. C3 deposition was readily detectable with NHS alone. To further test the effect of CRP, we supplemented the system with 1 µg, or 10 µg of purified human CRP. However, no significant enhancement of C3 deposition was observed, indicating that exogenous CRP did not markedly promote complement deposition under these conditions. These data have been included in the revised manuscript as Figure S7 to address this concern. The main text has also been accordingly revised (page 13, lines 377-379).

c. Can the binding be inhibited by exogenous capsule of a serotype human CRP binds but not by serotype 8?

Response: To address whether bacterial binding can be inhibited by exogenous capsule of a serotype that human CRP binds (e.g., serotype 23F), but not by a serotype it does not bind (e.g., serotype 8), we performed a C3 deposition assay. When *S. pneumoniae* serotype 23F was incubated with human serum, robust C3 deposition was observed. Addition of increasing amounts of purified CPS23F (100 µg, 200 µg) led to a dose-dependent reduction in C3 deposition (74.9% and 52.8%, respectively), indicating that CPS23F competitively binds serum CRP and inhibits complement activation. In contrast, addition of CPS8 at the same concentrations (100 µg, 200 µg) did not significantly reduce C3 deposition (71.7%, and 75.4%, respectively), consistent with the inability of CRP to bind serotype 8. These results confirm that the inhibitory effect on complement deposition is specific to capsules that can bind CRP. These data have been included in the revised manuscript as Figure S7 to address this concern. The main text has also been accordingly revised (page 13, lines 375-379).

3. The authors should provide more explication of statistical significance for some of the results. Examples:

a. Fig. 1D, E. How was significance determined? Do error bars represent SE?

Response: In these panels, significance was determined using Two-way ANOVA with Tukey's (A, D and J) or Sidak's (E) multiple comparisons were performed. The *P*-values are indicated as follows: **, *P* < 0.01; ***, *P* < 0.001; ****, *P* < 0.0001; and ns, not significant. Regarding the error bars, they represent the standard error of the mean (SEM).

b. Last line, page 7. How is "significantly" defined? More generally, what statistical analyses are performed on CFU counts at 5 and 30 min? In figure 2, the differences are stark, but in figure 3, some differences are moderate and could be evaluated statistically.

Response: In the last line of page 7, the term "significantly" refers to statistical significance determined by Two-way ANOVA with Tukey's multiple comparisons test, which was used throughout the manuscript to analyze differences between groups. For the statistical analysis, a *P*-value of < 0.05 is considered significant, with the following thresholds: **, *P* < 0.01; ***, *P* < 0.001; ****, *P* < 0.0001; ns, not significant. We will update the manuscript to clarify this in the relevant sections. For the CFU counts at 5 and 30 min, we did not perform statistical analyses as these data were primarily presented for descriptive purposes. In Figure 3, some differences are more moderate but were still evaluated statistically, and the results were determined to

be significant. We will ensure that the statistical analyses are explicitly noted in the figure legends and manuscript to clarify that these differences in Figure 3 were statistically evaluated.

c. Fig. 2 vs. Fig. 4A and page 9, line 6. Statistically compare Ct50 to document the statement.

Response: We have calculated the Ct50 values (Fig. 5A) and have accordingly revised the text (page 10, lines 281-282).

d. Fig. 4C and D. A statistical comparison of C3 KO vs. CR3/CR1g KO mice might provide evidence that one need not invoke any other receptors to explain defects. In addition, it is not clear from Figure/figure legend that C and D differ in inoculum.

Response: The degree of bacterial clearance in C3 KO and CR3/CR1g KO mice was found to be similar, with Ct50 values of 2.07 and 2.09 min, respectively. These results suggest that CRP-mediated bacterial clearance occurs downstream of C3, without the involvement of additional receptors. In addition, we have already indicated the difference in inoculum between panels C and D in the figure and figure legend.

e. Fig. 5. the KD of CRP for 23F is similar to that of 33F--is there a way to evaluate whether this difference is significant? If not, the text should be softened. In addition, the concentrations are given in $\mu\text{g/ml}$ whereas concentrations in μM are more intuitive. See also Fig. S3 and EC50.

Response: According to your suggestion, we have revised the text. In Figure 5C and D, we used μM because both phosphocholine and phosphoglycerol have well-defined molecular weights. However, for CRP, we used $\mu\text{g/ml}$. Therefore, we have revised Figure 6C and D accordingly.

f. Fig. 6. What is the significance of the open bar in 6A for serotype 8? what is considered "background" and how was that determined?

Response: Base on the mass spectrometry results and the binding of human Kupffer cells to *Sp8*, serotype 8 can be as a negative control. The significance of differences between serotype 8 and the other groups was determined using one-way ANOVA with multiple comparisons, confirming that the differences observed were statistically significant relative to the negative control. We have revised Figure 7A accordingly.

g. Page 9, line 10. the relationship between Fig. S1A and the dose chosen is not clear.

Response: The dose was selected based on Fig. S1A, which shows that WT mice can survive at this dose, making it suitable for evaluating bacterial clearance in $C3^{-/-}$ mice. This dose provides a sufficient challenge to the immune system without being lethal to the WT mice, allowing for a meaningful comparison between the two groups. The main text has also been accordingly revised (page 10, line 285).

Minor issues.

1. Fig. 2A and C. Consider including depicting structures of 23F and 14 in Fig. 2A to emphasize different structures. Indicate on Fig. 2C that infecting strain was 23F and that the mouse was CRP-deficient. (The 23F structure in Fig. 1A is not referred to and is not compared to other serotypes in Fig. 1 and could be removed.) Finally, the

phrase "demonstrated that CRP is necessary and sufficient for serotype-specific hepatic capture and killing of serotype-23F pneumococci" might be more accurately read "demonstrated that the only defect of CRP-deficient mice in serotype-specific hepatic capture and killing of serotype-23F pneumococci is CRP".

Response: Thank you for your insightful comments and suggestions. We appreciate your suggestion to include the structures of 23F and 14 serotypes in **Fig. 2A** to emphasize their structural differences. Additionally, we have added a note in **Fig. 2G** to clarify that the infecting strain was 23F and that the mouse was CRP-deficient to provide more context for the experimental setup. We have revised the text (page 7, lines 199-202).

2. P. 7. 5 lines from bottom. Might be clearer as "serotypes, i.e. all except serotype e, were..."

Response: Revised (page 8, lines 233-235).

4. P. 11, line 14. Remove "While".

Response: Revised.

5. Page 9, last line of first paragraph. "relies" suggests that the C3-independent mechanism is the sole means of clearance; "requires" might better convey the authors' meaning.

Response: Revised.

6. Fig. 5. The figure would be easier for readers to digest if the authors made use of colors like their use in other figures, e.g., have 27, 23F, and 33F different colors and keep them consistent throughout all panels, and in Panel A, to make things obvious, authors could indicate the PC and PG groups on 27 and 23F, respectively. In addition, do the authors think that the stoichiometry of 5:1 for 23F is based on the pentameric structure of CRP? How would the authors explain the stoichiometry of 33F (2.9)?

Response: Thank you for your thoughtful suggestions regarding Fig. 5.

a. We agree with your recommendation to use distinct colors for CPS27, CPS23F, and CPS33F in Fig. 5 to improve clarity and readability. We have updated the figure to assign unique colors to each serotype and ensure that these colors are used consistently across all panels. This will make it easier for readers to differentiate between the different serotypes and follow the data more easily.

b. We appreciate your suggestion to explicitly indicate the PC and PG groups on CPS27 and CPS23F, respectively, in Panel A. We will add clear annotations to this panel, marking PC for CPS27 and PG for CPS23F, to make the distinction between these groups more obvious for readers.

c. Stoichiometry of 5:1 for 23F and Pentameric Structure of CRP: Regarding the stoichiometry of CPS23F, the observed 5:1 ratio is indeed consistent with the pentameric structure of CRP. We hypothesize that the five binding sites on CRP's pentameric structure contribute to this specific stoichiometry.

d. The observed stoichiometry for CPS33A is 35.4. This is due to the weak affinity between CRP and CPS33A, which results in the titration curve not displaying an S-shape, meaning that the N value is inaccurate.

Dear Jing-Ren,

Thank you again for the submission of your revised manuscript (EMBOJ-2025-120577R) to The EMBO Journal for our consideration, and for your patience during peer review. Your manuscript has been sent back to the three original referees who had previously assessed the first version of the work, and we have now received the complete set of their comments, which are included below.

As you will see, the referees are satisfied with the revision, mention that the work is significant, novel, and relevant, point out that all initially raised concerns have been adequately addressed in a significantly strengthened revised manuscript, and now support publication of the manuscript without any further comments. In light of this input, I am pleased to inform you that your manuscript has been accepted in principle for publication in The EMBO Journal. Congratulations on an excellent work!

There are only a few changes from the editorial side that we kindly request you to address in a final version of your manuscript, before we can move forward with its formal acceptance and publication:

- We noticed that five co-authors have been designated as "co-first" authors, which is rather unusual for our journal. We kindly request that you consider again the actual contributions of all co-authors and reduce the number of co-first authors to no more than three, if possible. Please also note that we use the CRediT author contribution taxonomy system for specifying in detail the nature of each co-author's contribution (see below for more information), while contributions at the Figure panel level can also be specified (please see our guide to authors for more information: <https://www.embopress.org/page/journal/14602075/authorguide>).
- Please include the funding information in the "Acknowledgements" section of the revised manuscript.
- The maximum number of keywords that can be listed after the Abstract is 5 (9 keywords are currently listed).
- Please rename heading "DATA AND MATERIALS AVAILABILITY" to "Data availability".
- Please make sure that all deposited cryo-EM and mass spectrometry proteomics datasets will be publicly available at the time of publication. Please also add in the Data availability statement of the revised manuscript the permanent and specific URLs to each deposited dataset.
- Heading "COMPETING INTERESTS" should be renamed to "Disclosure and competing interests statement".
- Please include the following information in your "Disclosure and competing interests statement": "George F. Gao is a member of the Advisory Editorial Board of The EMBO Journal. This has no bearing on the editorial consideration of this article for publication."
- The author contributions statement should be removed from the manuscript file. Instead, we use CRediT to specify the contributions of each author in the journal submission system. Please feel free to use the free text box to provide more detailed descriptions during submission. See also our guide to authors for more information: <https://www.embopress.org/page/journal/14602075/authorguide#authorshipguidelines>.
- All Figure panel callouts should be listed sequentially.
- Callouts for Appendix Figures and Tables should be corrected to "Appendix Figure S1-S11" and "Appendix Table S1-S10".
- We noticed that there are callouts for Fig. 3E and 3F, but no panels E-F in Fig. 3.
- Callout(s) for Fig. 8 are missing.
- Please remove Movie legends from the Appendix PDF file (and its Table of Contents on the title page); each Movie file should be provided in a ZIP folder containing also its corresponding legend; Serial No. and Format are not required in the Table of Contents of the Appendix file, only the Figure/Table name and the corresponding page number are necessary.
- Please note that EMBO press papers are accompanied online by:
 - A) a short (2 sentences) summary of the findings and their significance,
 - B) 2-5 short bullet points highlighting the key results, and
 - C) a synopsis image in .jpg or .png format that is exactly 550 pixels wide and 300-600 pixels high (the height is variable). Please note that all text needs to be legible at the final size.Please upload this information along with your revised manuscript (the text for A and B should be provided in a separate Word file).

- During our standard pre-publication data checks, our data editors raised the following queries about data, Figures, and their legends. Please make sure that they are completely addressed in the final version of your manuscript (and highlight all changes to the Figure legends).

1. When $n=2$, no statistics (such as SD error bars) can be calculated and shown; please only show the individual data points instead.
2. Please make sure that the error bars are defined in the legends of Figures 1A D, E, J; 2A-G; 3A-C; 4A-F; 5A-H; 6E, 7A, B, I; EV1 A, B; EV3 A-E; EV4 A-C.
3. Please define the annotated p-values ****/****/**/ and provide the exact p-values for the same in the legends of Figures 1A, D E, J; 2A, D E, F, G; 3A, C; 4A-F; 5A-H; 7A, EV1 B; EV3 C as appropriate.
4. Please indicate the statistical test used for data analysis in the legends of Figures 1A, D E, J; 2A, C, D, E, F, G; 3A, C; 4A-F; 5A-H; 7A; EV1 B; EV3 C, EV4 B.

- Please add heading "EV Figure Legends" before the legends of EV Figures.

- The Movie files should be renamed to "Movie EV1-EV2", and their corresponding callouts be updated accordingly; the Movie legends should be removed from the Appendix PDF file and zipped instead together with each Movie file, as also explained above.

- The order of manuscript sections must be corrected as follows: Title page - Abstract and Keywords - Introduction - Results - Discussion - Methods - Data Availability - Acknowledgements - Disclosure and Competing Interests Statement - References - Figure Legends - main Tables (if applicable) - Expanded View Figure Legends.

Please also note that as part of the EMBO publications' Transparent Editorial Process, The EMBO Journal publishes online a Peer Review File along with each accepted manuscript. This File will be published in conjunction with your paper and will include the referee reports, your point-by-point response and all pertinent correspondence relating to the manuscript. You can opt out of this by letting the editorial office know (contact@embojournal.org). If you do opt out, the Peer Review File link will point to the following statement: "No Peer Review File is available with this article, as the authors have chosen not to make the review process public in this case."

We look forward to seeing a final version of your manuscript as soon as possible. Please let us know if you have any questions and use this link to submit your revision: <https://emboj.msubmit.net/cgi-bin/main.plex>.

Best regards,

Ioannis

Referee #1:

General summary and opinion

This manuscript addresses the role of C-reactive protein (CRP) as a broad-spectrum capsule-binding plasma protein that facilitates hepatic clearance of blood-borne bacteria. The work is comprehensive, spanning biochemical assays, structural analysis, proteomics, and in vivo experiments in mice. The study provides new insights into the mechanisms of CRP-mediated immunity, suggesting that CRP functions as a capsule-binding molecule that promotes complement deposition and bacterial clearance via liver macrophages. The revisions have strengthened the manuscript by addressing key methodological and interpretational concerns raised previously. The study is of high significance as it revises our understanding of CRP, shifting from a primarily phosphocholine-binding role to a broader one in capsule recognition, with implications for antibacterial immunity. Minor concern.

The corrections to the discussion section of the manuscript have not been highlighted in red in the revised manuscript. I believe that the corrections were made.

Referee #2:

The authors have thoroughly addressed all my previous concerns and have incorporated all of my suggestions. I found the revised manuscript a pleasure to read, and I believe it is now suitable for publication in The EMBO Journal. Congratulations to the authors for this seminal contribution.

Referee #3:

All of concerns raised by my initial review have been addressed.

All editorial and formatting issues were resolved by the authors.

Dear Jing-Ren,

Congratulations on an excellent manuscript! I am very pleased to inform you that it has been accepted for publication in The EMBO Journal. Thank you for comprehensively addressing the initially raised referee criticisms and the editorial requests for corrections and changes.

If you have any questions, please do not hesitate to contact the Editorial Office. Thank you for your contribution to The EMBO Journal. Working with you has been a pleasure!

Best regards,

Ioannis
